# A simple model for daily basin-wide thermodynamic sea ice thickness growth retrieval

James Anheuser[1], Yinghui Liu[2], and Jeff R. Key[2]

[1]Department of Atmospheric and Oceanic Sciences, University of Wisconsin-Madison, Madison, Wisconsin
[2]Center for Satellite Applications and Research, NOAA/NESDIS, Madison, Wisconsin

**Correspondence:** James Anheuser (anheuser@wisc.edu)

**Abstract.** As changes to Earth's polar climate accelerate, the need for robust, long–term sea ice thickness observation datasets for monitoring those changes and for verification of global climate models is clear. By linking an algorithm for retrieving snow–ice interface temperature from passive microwave satellite data to a thermodynamic sea ice energy balance relation known as Stefan's Law, we have developed a retrieval method for estimating thermodynamic sea ice thickness growth from space: Stefan's Law Integrated Conducted Energy (SLICE). With an initial condition at the beginning of the sea ice growth season, the method can model basin-wide absolute sea ice thickness by combining the one-dimensional SLICE retrieval with an ice motion dataset. The advantages of the SLICE retrieval method include daily basin-wide coverage, lack of atmospheric reanalysis product input requirement, and a potential for use beginning in 1987. Validation of the retrieval against measurements from ten ice mass balance buoys show a mean correlation of 0.89 and a mean bias of 0.06 m over the course of an entire sea ice growth season. Despite its simplifications and assumptions relative to models like the Pan-Arctic Ice–Ocean Modeling and Assimilation System (PIOMAS), basin-wide SLICE performs nearly as well as PIOMAS when compared against CryoSat-2 and Operation IceBridge using a linear correlation between collocated points.

## 1 Introduction

Observing sea ice concentration and areal extent from satellites is a well established practice (Liu et al., 2016; Meier et al., 2017; Markus and Cavalieri, 2000; Markus and Cavalieri, 2009; Comiso, 2009; Lavergne et al., 2019). There are methods based on data in the visual, infrared and microwave wavelength bands and climate data records produced from these methods are commonly cited as polar climate indicators (Stroeve et al., 2012; Screen and Simmonds, 2010; Liu et al., 2009).

While sea ice concentration is more readily observed, sea ice thickness provides a more complete characterization of the state of the climate system because it allows for calculation of sea ice volume and latent heat release. Recent literature has made clear that reliable long-term observations of basin wide sea ice thickness are needed in order to constrain the representations of sea ice in global climate models (Mayer et al., 2019). Sea ice thickness based observations of sea ice volume can be used along with other observations to refine the large range of projected sea ice area and volume across coupled global climate models (Docquier and Koenigk, 2021). Indeed, the lack of reliable long term sea ice thickness observation constraints is the primary barrier to reducing the uncertainty in future sea ice area and volume projections (Massonnet et al., 2018).

Sea ice thickness derived from space-based altimetry data collected by satellites like CryoSat-2 and the Ice, Cloud, and land Elevation Satellite (ICESat)-2 stand as the current state of the art (Connor et al., 2009; Kwok and Cunningham, 2008; Markus et al., 2017, Wingham et al., 2006; Laxon et al., 2013). However, their low spatial coverage and orbit characteristics make for incomplete and infrequent sampling across the Arctic basin while uncertainties in snow depth and ice density lead to high uncertainties in sea ice thickness estimations from these instruments (Wang et al., 2016). A method of combining data

from CryoSat-2 and ICESat-2 has been shown to alleviate some of these concerns (Kwok et al., 2020). Additional satellite based methods include estimating the thickness of thin sea ice using low-frequency passive microwave satellite data (Tian-Kunze et al., 2014) and combining low frequency passive microwave data with altimetry data in order to take advantage their complementing data and spatial coverage (Ricker et al., 2017b; Zhou et al., 2018; Shi et al., 2020). Other strategies for retrieving sea ice thickness include a one-dimensional surface energy balance model driven by satellite products (Key et al.,

2016) and a coupled ocean–sea ice model with assimilated observational data called Pan-Arctic Ice–Ocean Modeling and Assimilation System (PIOMAS; Zhang and Rothrock, 2003). Another approach involves correlating sea ice thickness with sea ice age (Liu et al., 2020). The various available products are discussed and compared against one another both qualitatively and quantitatively in Wang et al. (2016) and against upward looking sonar (ULS) in Sallila et al. (2019).

     Recent efforts to retrieve temperature at the boundary between snow and sea ice, referred to as the snow–ice interface

temperature, have opened a new door in polar climate observation (Kilic et al., 2019; Lee and Sohn, 2015; Lee et al., 2018). These methods take advantage of radiances from the Advanced Microwave Scanning Radiometer (AMSR)-Earth Observing System (-E), AMSR2, the Special Sensor for Microwave Imager (SSM/I) and Special Sensor Microwave Imager/Sounder (SSMIS) passive microwave instruments using channels whose wavelengths carry information from the snow–ice interface. Kang et al. (2021) demonstrated the utility of these snow–ice interface temperature data by using them to nudge a sea ice

model, improving the model's results. By coupling this newly available snow–ice interface temperature data with Stefan's Law governing the thermodynamics of sea ice growth (Stefan, 1891; Lepparanta, 1993), we introduce a new method of retrieving thermodynamic sea ice thickness growth called Stefan's Law Integrated Conducted Energy (SLICE).

     As sea ice accretes on the underside of the ice layer, the latent heat of fusion conducts up through the ice to the snow–ice interface. In Stefan's Law, that conducted heat and therefore rate of accretion is calculated using a heat conduction equation

with the snow–ice interface temperature as the upper boundary condition and the local freezing temperature of sea water set as the lower boundary condition (Stefan, 1891; Lepparanta, 1993). By using the satellite retrieved snow–ice interface temperature and an assumed initial ice thickness value in this relationship, SLICE is able to retrieve daily rates of ice accretion and model sea ice thickness on a basin-wide scale during the sea ice growth season by integrating these retrieved growth rates from an initial sea ice thickness condition.

## 55  2   Data

The SLICE retrieval method described here utilizes passive microwave brightness temperatures and a passive microwave based sea ice concentration dataset. Modelling basin-wide sea ice thickness requires motion vectors and a satellite based initial

condition. A preliminary validation of the retrieval method references sea ice thickness from ice mass balance buoy data, satellite and airborne altimeter based sea ice thickness data, and sea ice models.

## 2.1 Passive microwave brightness temperatures and sea ice concentration

The AMSR-E and AMSR2 brightness temperatures available from the National Snow and Ice Data Center (NSIDC) were used in this study (Cavalieri et al., 2014; Markus et al., 2018). The AMSR-E data is available for June 2002 through October 2011 and the AMSR2 data is available for July 2012 to the present. The AMSR2 data has been intercalibrated with the AMSR-E data and the brightness temperatures between these two instruments are treated here as a continuous dataset (Markus et al., 2018). The data is provided on a 25 km polar stereographic grid, but when needed on a basin-wide scale for use with the sea ice thickness retrieval method described here, the data were linearly interpolated to a 25 km Equal-Area Scalable Earth (EASE)-Grid 2.0.

The NASA Team 2 algorithm is a passive microwave brightness temperature sea ice concentration algorithm (Markus and Cavalieri, 2000). It is an enhancement to the original NASA Team algorithm (Cavalieri et al., 1984; Gloersen and Cavalieri, 1986) in that it adds 85 GHz frequency brightness temperatures to the original algorithm, which used only 19 GHz and 37 GHz data, in order to better account for interference from surface effects. The algorithm utilizes open ocean and 100% ice concentration tie points in polarization and spectral gradient ratios to determine sea ice concentration. While originally developed for use with SSM/I data (Markus and Cavalieri, 2000), the algorithm was planned to be and is now in use with AMSR-E and AMSR2 data. Here we use this AMSR-E and AMSR2 sea ice concentration data, which is available from the NSIDC as a part of the same dataset that contains the brightness temperatures used to calculate snow–ice interface temperature (Cavalieri et al., 2014; Markus et al., 2018).

## 2.2 Sea ice motion vectors

Sea ice is not static, but rather a dynamic collection of variably sized parcels that are each in constant horizontal motion under the effects of wind, ocean currents and internal stress. In order to treat sea ice in a Lagrangian sense, the motion of these parcels must be understood. Here we use the Polar Pathfinder Daily 25 km EASE-Grid Sea Ice Motion Vectors, Version 4 available from the NSIDC (Tschudi et al., 2019; Tschudi et al., 2020). The product is available from 1978 to present at daily and weekly temporal resolution, each with basin-wide coverage and each on the 25 km EASE-Grid. Using an optimal interpolation scheme, ice motion vectors are created from cross correlated satellite brightness temperature data from AMSR-E, Advanced Very High Resolution Radiometer (AVHRR), Scanning Multichannel Microwave Radiometer (SMMR), SSM/I and SSMIS, along with International Arctic Buoy Program (IABP) buoy locations and a National Centers for Environmental Protection (NCEP)/National Center for Atmospheric Research (NCAR) wind reanalysis data derived free drift estimate. Each satellite and buoy dataset are included in the optimal estimation scheme only when available within the life span of the ice motion product. This means input data sources vary throughout the record. DeRepentigny et al. (2016) found the weekly sea ice motion vectors to have a 7% median error when compared against IABP buoys between 1988 and 2011.

## 2.3 Airborne and satellite based sea ice thickness products

CryoSat-2 carries the SAR/Interferometric Radar Altimeter-2 (SIRAL-2) instrument (Wingham et al., 2006; Laxon et al., 2013) and was launched by the European Space Agency (ESA) in 2010. Similar to other satellite altimeters, ice thickness is determined from CryoSat-2 data by first calculating the thickness of the sea ice above sea level—known as the freeboard—by determining the distance traveled by the radar signal between the satellite and the the ice surface and subtracting that distance from the satellite orbit altitude above sea level. An assumed snow loading provides a correction for the reduced propagation speed of the radar signal through snow. Then, the assumed snow loading and a hydrostatic balance is used to determine sea ice mass which in turn is converted to thickness using an assumed density (Laxon et al., 2013). Gridded ice thickness products derived from ESA CryoSat-2 Level 1b data are provided by numerous sources (Tilling et al., 2018; Kurtz et al., 2014a; Ricker et al., 2014; Hendricks and Ricker, 2020; Ricker et al., 2017a; Kwok and Cunningham, 2015; Hendricks et al., 2018; Guerreiro et al., 2017). The primary differences between these datasets relate to averaging period, grid sizing and radar response waveform retracking procedure.

The ESA Soil Moisture and Ocean Salinity (SMOS) satellite carries the Microwave Imaging Radiometer using Aperture Synthesis (MIRAS) instrument which measures 1.4 GHz passive microwave brightness temperatures at 35 to 50+ km resolution (Mecklenburg et al., 2012). While originally intended for measuring soil moisture and ocean salinity, the high penetration depth of the 1.4 GHz channel into sea ice allows for retrieval of an ice temperature that when incorporated into a radiative transfer model yields a sea ice thickness estimate (Tian-Kunze et al., 2014). This approach has associated uncertainties in sea ice below 0.5 m thick that are lower than those of satellite altimeters.

Sea ice thickness observations from SMOS and CryoSat-2 have complementing uncertainties. SMOS has high uncertainties when measuring thick ice and CryoSat-2 has high uncertainties when measuring thin ice (Ricker et al., 2017b). This creates an opportunity for synergy between the instruments. The AWI CS2SMOS dataset takes advantage of this synergy. By combining the datasets through a weighted averaging scheme, root mean squared errors are reduced from 76 cm with CryoSat-2 alone to 66 cm and the squared correlation coefficient is increased from 0.47 with CryoSat-2 to 0.61 when compared against NASA Operation Ice Bridge data (Ricker et al., 2017b). The AWI CS2SMOS dataset is available at a weekly time resolution and on a 25 km EASE-Grid 2.0 and was used with the method demonstrated here due to the high spatial coverage.

The NASA Operation IceBridge (OIB) mission is an airborne campaign comprising a series of flights covering the years 2009-2020, bridging the gap between the NASA Ice, Cloud and land Elevation Satellite (ICESat) and NASA ICESat-2 laser altimeter satellite missions (Kurtz et al., 2013). Among the instruments aboard each OIB flight, of primary importance to this study are the Airborne Topographic Mapper (ATM) instrument (Krabill et al., 1995) and snow radar (Panzer et al., 2013). The ATM is a laser altimeter whose return signal is used along with an aerial photography based sea ice lead (fracture) discrimination algorithm to retrieve sea ice total freeboard height—i.e., freeboard plus snow depth—at 40 m spatial resolution. The snow radar return signal is used to determine snow depth. Sea ice freeboard and snow depth are used in conjunction with a hydrostatic balance to determine sea ice thickness at the sea ice freeboard resolution of 40 m. We use OIB sea ice thickness from the IceBridge L4 Sea Ice Freeboard, Snow Depth, and Thickness, Version 1 and its Quick Look counterpart as provided

by the NSIDC (Kurtz et al., 2015; Kurtz et al., 2016). The data from 2014-2019, covering all of the data used here except those form 2013, are only available in the Quick Look format and may have increased uncertainties due to a roughly 5 cm underestimation of snow depths by the snow radar waveform retracking method (Kwok et al., 2017). Figure 1 shows OIB flight paths used in this study.

## 2.4 Ice Mass Balance Buoys

In order to statistically characterize the sea ice thickness retrieval method described herein, ice mass balance buoy data served as the reference. The ice mass balance buoys were deployed and maintained by the United States Army Corps of Engineers Cold Regions Research and Engineering Laboratory (CRREL) (Perovich et al., 2021). Undeformed ice floes are chosen for buoy sites to ensure the buoy is representative of the surrounding ice (Polashenski et al., 2011).

Data fields used from the buoys were sea ice thickness and geolocation in latitude and longitude. Ice thickness is observed using two acoustic rangefinder sounders, one positioned above and one positioned below the ice. Winter sea ice growth is derived from the under–ice sounder. Each sounder has an accuracy of 0.005 m (Richter-Menge et al., 2006). An Argos antenna mounted on the buoy transmits the geolocation and other observations at minimum twice per day (Richter-Menge et al., 2006). For this study, all data fields were resampled to 1 d resolution by calculating daily mean values. All buoys from the years 2003 to 2016 showing an entire season of sea ice thickness growth were used for comparison with the exception of buoys installed in landfast ice and those that show obvious ice deformation effects, which often lead to the end of data acquisition. As such, sea ice thickness growth observed by the buoys is taken to be caused strictly by thermodynamic processes. Table 1 provides details pertaining to the buoys used. Buoy 2013F spanned two winter seasons and as such has been divided into two buoy numbers, 2013F and 2013Fb, with 2013Fb covering the second winter season during which the buoy was deployed. As such, a deployment date is not listed for 2013Fb. Drift tracks from 1 November to 1 April for the buoys are shown in Fig. 1. The buoys are concentrated in the Central Arctic and Beaufort Sea.

Efforts to compare satellite based records of sea ice thickness with ground truth are hampered by the scale of the question. Ground truth measurements of sea ice are necessarily taken from a single point while satellites observe sea ice thickness on the scale of kilometers. The variability of sea ice across those kilometers leads to uncertainty in the comparison. It has been shown, however, that while variability in absolute ice thickness may be significant on the scale of a satellite observation, sea ice growth and melt are relatively uniform on the satellite length scale (Polashenski et al., 2011). Therefore, while absolute comparisons of sea ice thickness between a ground truth and satellite observation may be tenuous, comparisons of growth over a winter season between single point ground truth and satellite based observations are more robust.

## 2.5 Sea ice models

PIOMAS is a numerical model reanalysis product that couples the Parallel Ocean Program (POP) model developed at Los Alamos National Laboratory with a thickness and enthalpy distribution (TED) model (Zhang and Rothrock, 2003). The TED model includes a viscous–plastic sea ice rheology (Hibler, 1979) and a sea ice thickness distribution scheme that accounts for redistribution due to ridging (Thorndike et al., 1975). The model utilizes a generalized orthogonal curvilinear coordinate

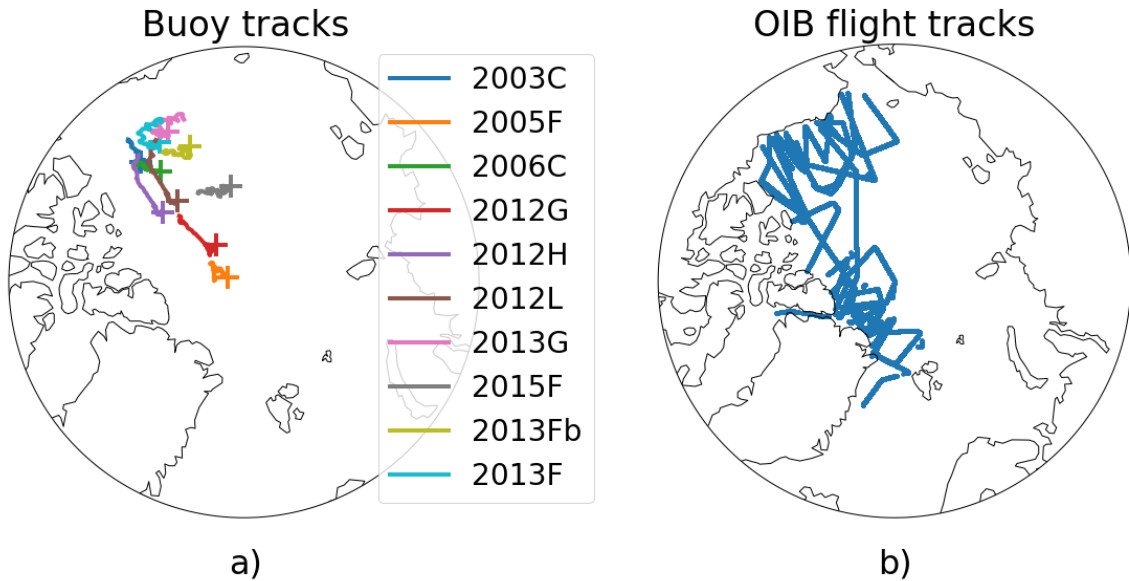

**Figure 1.** Tracks traveled by a) ice mass balance buoys and b) Operation IceBridge flights used in this study. Initial buoy location is signified with a plus symbol and all buoy tracks from 1 November to 1 April.

**Table 1.** A listing of United States Army Corps of Engineers Cold Regions Research and Engineering Laboratory (CRREL) Ice Mass Balance buoys used in this work. All buoys from 2003 to 2016 containing a full season of sea ice thickness growth are included, excluding those in landfast ice or showing obvious dynamic effects.

| Buoy | Region | Ice Type | Deployment date | Final Acquisition Date |
|---|---|---|---|---|
| 2003C | Beaufort Sea | Multi-year | 2002/8/31 | 2004/7/12 |
| 2005F | Central Arctic | Multi-year | 2005/9/3 | 2007/3/22 |
| 2006C | Beaufort Sea | Multi-year | 2006/9/4 | 2009/8/22 |
| 2012G | Central Arctic | First year | 2012/10/1 | 2015/5/20 |
| 2012H | Beaufort Sea | First year | 2012/9/10 | 2014/1/16 |
| 2012L | Beaufort Sea | Multi-year | 2012/8/27 | 2013/9/25 |
| 2013F | Beaufort Sea | Multi-year | 2013/8/25 | - |
| 2013Fb | Beaufort Sea | Multi-year | - | 2015/8/27 |
| 2013G | Beaufort Sea | Multi-year | 2013/9/4 | 2014/5/5 |
| 2015F | Central Arctic | Multi-year | 2015/8/13 | 2016/8/2 |

(GOCC) grid. The northern grid pole is shifted to be over Greenland where there is no ocean or sea ice, avoiding the need to deal with converging meridians and grid cell areas that approach zero. This shift also allows the highest grid resolutions to occur in the Canadian Archipelago, Baffin Bay and the Greenland Sea where the geography is intricate. The model is driven by

daily surface forcing and sea surface temperatures (SSTs) provided by NCEP/NCAR and NSIDC sea ice concentration in order to produce daily sea ice thickness distributions from 1978 to present (Schweiger et al., 2011). Here we use the daily effective sea ice thickness output which converts sea ice thickness distribution into a single effective thickness value for each grid cell.

Additionally, the data output from the sea ice model described in Kang et al. (2021), henceforth K21, was used to provide additional context. As briefly described in Sect. 1, K21 nudged a one-dimensional sea ice thermodynamic model with satellite retrieved snow–ice interface temperatures. The model itself is based on Maykut and Untersteiner (1971) and is a multi-layer numerical approximation for the snow and sea ice system. While the model is driven by European Centre for Medium-Range Weather Forecasts (ECMWF) ERA-Interim reanalysis data, temperatures at the snow–ice interface are nudged using the snow–ice interface temperature retrieval described by Lee and Sohn (2015) in a non-physical correction term. This method was extrapolated to basin-wide results through the use of Lagrangian tracking of individual sea ice parcels. The SLICE model is similar but greatly simplified and driven directly by the snow–ice interface temperature retrieval.

## 3 Methodology

Sea ice grows thicker through two primary physical mechanisms: thermodynamic phase change and dynamic changes due to the relative motion of the ice pack. The governing equation for a Eulerian sea ice thickness field can be written as

$$\frac{\partial H}{\partial t} = f(t, H, \mathbf{x}) - \nabla \cdot (\mathbf{u}H), \tag{1}$$

where $H$ is plane slab sea ice thickness, $t$ is time, $f$ is a function of time, thickness and position vector $\mathbf{x}$ describing thermodynamic sea ice thickness increase, and $\mathbf{u}$ is the ice motion vector. This equation is analogous to Eq. (3) in Thorndike et al. (1975), but does not include the redistribution term because here we use a plane slab thickness $H$ rather than a thickness distribution. The second term on the right hand side of Eq. (1) captures dynamic thickness changes, including both advection and deformation of sea ice. Through the following methodology called Stefan's Law Integrated Conducted Energy (SLICE), we retrieve the first term on the right hand side of Eq. (1) and use a parcel tracking approach to approximate the sea ice advection component of the second term on the right in order to model basin-wide thickness.

### 3.1 Snow–Ice Interface Temperature

In order to retrieve snow–ice interface temperature from passive microwave brightness temperatures, we used a multi-linear regression algorithm described by Kilic et al. (2019). The algorithm is developed on the premise that low frequency passive microwave brightness temperatures are well correlated with snow–ice interface temperature as demonstrated using a radiative transfer model by Tonboe et al. (2011).

The algorithm first determines an estimated snow depth with an algorithm developed using multi-linear regression of ground truth snow depth to AMSR2 passive microwave brightness temperatures in vertical polarization at 6.9 GHz, 18.7 GHz, and 36.5 GHz (6V, 18V, and 36V). The ground truth snow depth training dataset is taken from the 2012G, 2012H, 2012J, and 2012L CRREL IMBs. Another multi-linear regression was performed between ground truth snow–ice interface temperature

from the same CRREL IMBs, the estimated snow depth and AMSR2 passive microwave brightness temperatures from the 6.9 GHz vertical polarization channel, which was found to have the highest correlation to the buoy snow–ice interface temperature. The resulting algorithm is as follows:

$$D_s = 1.7701 + 0.0175T_{B,6V} - 0.0280T_{B,18V} + 0.0041T_{B,36V} \tag{2}$$

$$T_{si} = 1.086T_{B,6V} + 3.98log(D_s) - 10.70, \tag{3}$$

where $D_s$ is snow depth and $T_{B,6V}$, $T_{B,18V}$, and $T_{B,36V}$, are observed 6V, 18V and 36V channel brightness temperatures, respectively. When evaluated against a set of IMBs consisting of 2013F, 2013G, 2014F and 2014I, the regression shows a root mean square error (RMSE) of 2.9 K.

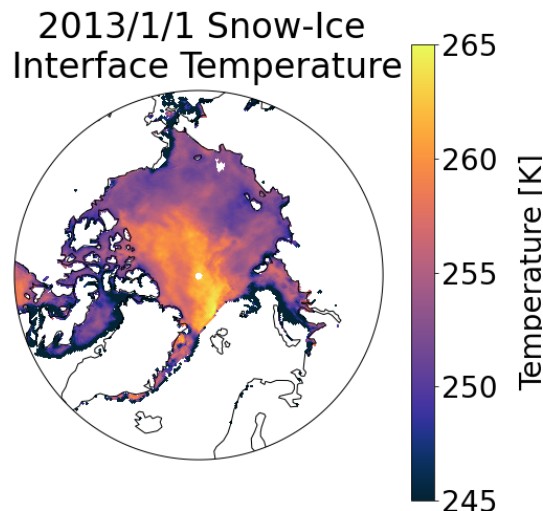

**Figure 2.** Snow–ice interface temperatures on 1 January 2013 derived from AMSR2 radiances.

Liquid water at the emitting layer in the form of open ocean or melt ponds interferes with the snow–ice interface temperature algorithm. As such, the snow–ice interface is only calculated here in grid cells with greater than 95% sea ice concentration. While the algorithm was developed for use with AMSR2, we applied it to brightness temperatures from both AMSR2 and AMSR-E using the continuous and intercalibrated dataset provided by NSIDC (Markus et al., 2018). Figure 2 shows snow-ice interface temperatures on January 1, 2013 derived from AMSR2 data.

### 3.2 Thermodynamic sea ice thickness growth

A simple model of one-dimensional thermodynamic sea ice thickness growth is a balance of heat fluxes where phase change is occurring, i.e., at the interface between the solid sea ice and liquid sea. These heat fluxes are latent heat released during the

phase change of liquid sea water to solid sea ice, $F_l$, basal sensible heat flux from the liquid sea to the solid ice, $F_w$, and heat conducted through the ice and away from the phase change interface, $F_c$. These three fluxes are balanced at the phase change interface as follows:

$$F_l + F_w = F_c. \tag{4}$$

When the snow–ice interface temperature drops below the temperature at the bottom of the ice, heat provided by basal sensible flux and the latent heat of freezing is pulled to the snow–ice interface. In the method described here, a satellite observation of snow–ice interface temperature (Kilic et al., 2019) drives this energy balance in order to determine sea ice thickness growth.

### 3.2.1 Stefan's Law

By balancing the conductive heat equation with a latent heat of freezing and basal sensible heat flux term, Stefan's Law relates the rate of thermodynamic sea ice thickness increase to the temperature difference between the snow–ice interface and bottom of the ice layer, the later of which is at or very near to the freezing temperature of sea ice (Stefan, 1891; Lepparanta, 1993).

Conduction is the transfer of heat across a solid medium and is always accompanied by a temperature difference across that medium. The equation governing one-dimensional, steady state conduction through sea ice is

$$F_c = \frac{\kappa_{eff}}{H} \left( T_f - T_{si} \right), \tag{5}$$

where $\kappa_{eff}$ is the thermal conductivity of sea ice, $H$ is sea ice thickness, $T_f$ is the freezing point of sea water and $T_{si}$ is the snow–ice interface temperature.

A change in the phase of a material must either release or accept energy as the molecular bonds and motion within the material change. In the case of a phase change from liquid to solid, energy is released as the molecular motion is reduced with the introduction of molecular bonds. The equation describing the one-dimensional, latent heat release as sea water changes phase from liquid to solid is:

$$F_l = \rho_i L \frac{dH}{dt}, \tag{6}$$

where $\rho_i$ is the density of the solid phase of the material, $L$ is the latent heat of fusion and $\frac{\partial H}{\partial t}$ is the change in sea ice thickness per unit time.

In Stefan's Law, Eqs. (5) and (6) are substituted into Eq. (4) to form

$$\rho_i L \frac{\partial H}{\partial t} = \frac{\kappa_{eff}}{H} \left( T_f - T_{si} \right) - F_w. \tag{7}$$

Isolating for sea ice thermodynamic growth rate, we have

$$\frac{\partial H}{\partial t} = \frac{\kappa_{eff}}{\rho_i L H} \left( T_f - T_{si} \right) - \frac{F_w}{\rho_i L}. \tag{8}$$

Equation (8) defines the thermodynamic growth function, $f$, found in Eq. (1) and is equivalent to Eq. (1) when dynamic growth is neglected. There are three assumptions inherent to this relationship (Lepparanta, 1993). First, heat conduction in the

horizontal direction is assumed to be negligible. Second, it is assumed that there is no thermal inertia present in the ice. This means that the local derivative of temperature with respect to sea ice depth is constant throughout the sea ice layer—i.e., the temperature profile is linear—and the system is in equilibrium. The spatial derivative of temperature found in a typical heat equation reduces to the temperature difference between the snow–ice interface temperature and the freezing point of water due to these first two assumptions. Third, it is assumed that there is no internal heat source, such as the absorption of short wave radiation. The second and third assumptions are only valid during polar winter and times of the year when solar incidence angles are very shallow.

Eqs. (7) and (8) are differential equations with the following solution:

$$H = \sqrt{H_0^2 + \delta t \frac{2\kappa_{eff}}{\rho_i L}(T_f - T_{si})} - \delta t \frac{F_w}{\rho_i L}, \tag{9}$$

where $H_0$ is the initial sea ice thickness. The time interval $\delta t$ chosen for the results shown herein is one day based on the daily availability of snow–ice interface temperature and both snow–ice interface temperature and basal sensible flux are assumed to be constant during each day.

At each time step, SLICE determines sea ice thickness by solving Eq. (9) for $H$ given an $H_0$ using the snow–ice interface temperature calculated at the nearest AMSR-E or AMSR2 grid cell. The change in sea ice thickness at each time step is dependent on initial sea ice thickness. This necessitates SLICE be applied in a Lagrangian sense as the sea ice thickness must be tracked and stored in order to accurately calculate the change at the next time step. In Eq. (8), thicker sea ice grows slower than thinner sea ice with a given snow–ice interface temperature. This means that in the presence of only thermodynamic effects, a SLICE sea ice thickness profile that is biased relative to ground truth will correct towards the unbiased SLICE thickness profile. This relationship replicates the phenomenon described in Bitz and Roe (2004), whereby thick ice recovers from climate related perturbations slower than thin ice and has experienced greater thinning on a decadal time scale.

### 3.2.2 Basal heat flux

Observation of basal sensible heat flux from liquid sea water to solid sea ice is inherently difficult. Typically, basal sensible heat flux is calculated as a residual of other more readily observed quantities in the heat budget of a one-dimensional sea ice profile, typically from a drifting station or buoy. Using this methodology, McPhee and Untersteiner (1982) observed March through May basal sensible heat fluxes of less than 2 W m$^{-2}$ using data from the FRAM I drift station in the Arctic Ocean, Perovich and Elder (2002) report oceanic sensible heat flux values of just a few W m$^{-2}$ from November to May during the Surface HEat Budget of the Arctic Ocean (SHEBA) field experiment and Lei et al. (2014) examined Chinese National Arctic Research Expedition (CHINARE) buoy data to discover relatively high basal sensible heat fluxes of greater than 10 W m$^{-2}$ through December that gradually decreased to near 0 by mid-February.

Maykut and Untersteiner (1971) completed a sensitivity analysis using their thermodynamic sea ice model, investigating the equilibrium mean annual sea ice thickness corresponding to basal sensible heat flux values ranging from 0 to 8 W m$^{-2}$. Realistic mean annual sea ice thickness resulted when basal sensible heat flux was set between 1.3 W m$^{-2}$ and 2.6 W m$^{-2}$. They chose a constant basal sensible heat flux value of 2 W m$^{-2}$ for their model based on this analysis and available observational data.

The coupled ocean–sea ice model PIOMAS supplies oceanic sensible heat fluxes to the sea ice model component as modeled by the ocean model component. Zhang and Rothrock (2003) show these modeled ocean sensible heat fluxes in most of the Arctic basin to be near 2 W m$^{-2}$. K21 also employed a constant basal sensible heat flux of 2 W m$^{-2}$.

In keeping with these studies, we apply a constant basal sensible heat flux $F_w = 2$ W m$^{-2}$. The effect of basal sensible heat flux on thermodynamic sea ice growth is independent of thickness and can be easily quantified as the last term of Eq. (9). For a given snow—ice interface temperature, the reduction of sea ice thickness growth by inclusion of a basal sensible flux is linearly related to the flux value by a factor of $1/\rho_i L$. With a density of 917 kg m$^{-2}$ and a latent heat of fusion of 3.32 x 10$^5$ J kg$^{-1}$, each 1 W m$^{-2}$ of basal sensible heat flux from the liquid sea water to solid sea ice decreases sea ice thickness growth by $2.84 \times 10^{-4}$ m d$^{-1}$. Removal of the 2 W m$^{-2}$ basal sensible heat flux would increase sea ice growth by $5.67 \times 10^{-4}$ m d$^{-1}$ and an increase from 2 W m$^{-2}$ to 10 W m$^{-2}$ would decrease thermodynamic sea ice thickness growth by $2.27 \times 10^{-3}$ m d$^{-1}$. This corresponds to a 0.0857 m increase and a 0.343 m decrease, respectively, when summed from 1 November to 1 April.

### 3.2.3 Multi-phase properties of sea ice

Sea ice is best described not as a homogeneous solid medium but rather as a heterogeneous, multi-phase material including solid ice and pockets of liquid brine whose size and salinity change with varying ice temperatures. In turn, these brine pocket changes significantly affect the bulk thermodynamic properties of the ice layer (Feltham et al., 2006). Equation (9) includes effective thermal conductivity, $\kappa_{eff}$, a property that is subject to this effect. As such, we adopt the parameterization of effective conductivity described in Feltham et al. (2006).

We begin by defining a constant ocean salinity, S, of 33 ppt. Next, we will assume that the ice is in thermal equilibrium relative to phase change between liquid brine and solid ice and calculate freezing point temperature (in °C), $T_f$, as a function of salinity (in ppt) per Notz (2005):

$$T_f(S) = -0.0592S - 9.37 \times 10^{-6}S^2 - 5.33 \times 10^{-7}S^3. \tag{10}$$

The latent heat of fusion for liquid to solid phase change is defined as the difference between the enthalpies of the two states. In this case, we will use a latent heat of fusion (in J kg$^{-1}$), $L$ as calculated as a function of temperature (in °C) by Notz (2005):

$$L(T_f) = 333700 + 762.7T_f - 7.929T_f^2. \tag{11}$$

We then use Eq. (15) from Feltham et al. (2006) to define effective thermal conductivity (in W m$^{-1}$ K$^{-1}$) as function of sea ice temperature, $T_i$ and sea ice salinity, $S_i$:

$$\kappa_{eff} = \kappa_{bi} - (\kappa_{bi} - \kappa_b)\frac{(T_f(0) - T_f(S_i))}{(T_f(0) - T_i)}, \tag{12}$$

where $\kappa_{bi}$ is the thermal conductivity of bubbly ice and $\kappa_b$ is the thermal conductivity of liquid brine and are defined per Schwerdtfecer (1963), Batrak et al. (2018) and Bailey et al. (2010) (all in W m$^{-1}$ K$^{-1}$):

$$\kappa_{bi} = \kappa_i(2\kappa_i + \kappa_a - 2V_a(\kappa_i - \kappa_a))/(2\kappa_i + \kappa_a + V_a(\kappa_i - \kappa_a)) \tag{13}$$

$$\kappa_b = 1.162(0.45 - 1.08 \times 10^{-2}T + 5.04 \times 10^{-5}T^2) \tag{14}$$

$$\kappa_i = 1.162(1.905 - 8.66 \times 10^{-3}T + 2.97 \times 10^{-5}T^2) \tag{15}$$

$$\kappa_a = 0.03 \tag{16}$$

$$V_a = 0.025\,, \tag{17}$$

where $\kappa_i$ is the thermal conductivity of pure ice, $\kappa_a$ is the thermal conductivity of air and $V_a$ is the fractional volume of air in the sea ice. We will use effective conductivity calculated with surface conditions for SLICE which is similar to the approach adopted by Cox and Weeks (1988) who also used conductivity calculated from the surface to determine conductive flux through the ice layer.

A first-year sea ice (FYI) density of 917 kg m$^{-3}$ and multi-year sea ice (MYI) density of 882 kg m$^{-3}$ was reported by Alexandrov et al. (2010) and these values have seen use in the sea ice thickness calculations from CryoSat-2 data (Laxon et al., 2013; Tilling et al., 2018; Hendricks and Ricker, 2020; Kwok and Cunningham, 2015). A sea ice density of 915 kg m$^{-3}$ is also in use with altimeter data (Kurtz et al., 2014b; Petty et al., 2020) and a sea ice density of 925 kg m$^{-3}$ has been used with IceSat data (Kwok and Rothrock, 2009). Choice of sea ice density is a significant source of uncertainty in altimeter-based estimates of sea ice thickness. Kurtz et al. (2014b) report that the range of densities from 882 kg m$^{-3}$ to 925 kg m$^{-3}$ yields a 1.1 m range in sea ice thickness estimates from a 60 cm snow—ice freeboard with 35 cm of snow. The SLICE system of equations uses a FYI sea ice density of 917 kg m$^{-3}$. For a given snow—ice interface temperature, basal sensible heat flux and sea ice thickness, a change to 915 kg m$^{-3}$ would increase sea ice thickness growth rate by at most only 0.2% and a change to 925 kg m$^{-3}$ would decrease sea ice thickness growth by at most only 0.8%.

### 3.3 Parcel tracking of advection

The divergence term on the right hand side of Eq. (1) represents sea ice dynamics and includes the effects of both advection and deformation on local sea ice thickness. Advection moves sea ice parcels horizontally and deformation redistributes sea ice volume vertically through ridging or lead formation.

The advection effects contained within the divergence term can be approximated using a parcel tracking approach, allowing for the modelling of basin-wide results. This parcel tracking methodology is initialized with the CS2SMOS data from the first week of November and each 25 km x 25 km grid cell is divided into 5 km x 5 km parcels, which are advected daily throughout the winter using the Polar Pathfinder motion vectors interpolated to their position. This parcel tracking methodology is similar to the Sea Ice Tracking Utility (SITU, http://icemotion.labs.nsidc.org/SITU/) described by DeRepentigny et al. (2016). Each parcel adds sea ice thickness thermodynamically using the SLICE thermodynamic model. At any given time step, the parcels can be gridded to the EASE-Grid 2.0 by taking the mean thickness of parcels within each grid cell. This approach leaves out

deformation effects. While taking the sum of all parcel volume within each grid cell and dividing by grid cell area would be a
more physically sound methodology for this regridding, the results from such a method proved unrealistic. Figure A1 found in
the Appendix shows a comparison between the chosen method using grid cell mean and the method using grid cell sum. New
parcels are initiated at any grid cell not containing a sea ice parcel but showing 95% or greater sea ice concentration. New ice
parcels are initialized at a thickness of 0.05 m. Any parcel that is located in a region with less than 95% sea ice concentration at
any time step is removed. The result is a daily gridded Arctic basin-wide sea ice thickness dataset representing thermodynamic
sea ice growth in the 95% sea ice concentration ice pack.

## 4 Results

The SLICE sea ice thickness retrieval methodology can be applied on a single one-dimensional profile basis or across a large
area. Here we present results comparing one-dimensional SLICE data to ice mass balance buoy thicknesses and Arctic basin-
wide results compared to OIB and AWI CS2SMOS data.

### 4.1 Ice mass balance buoy comparison

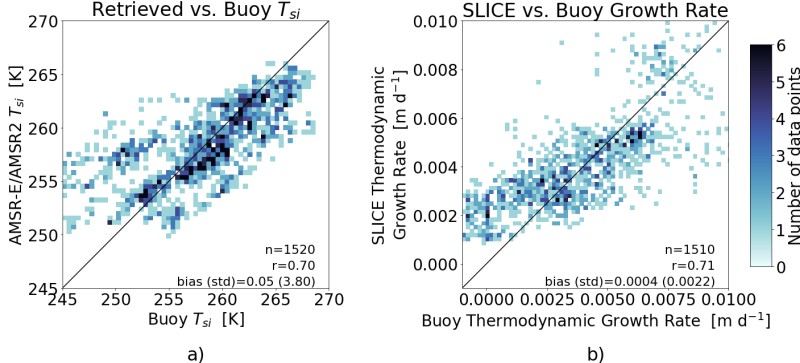

**Figure 3.** An instantaneous comparison of (a) AMSR-E/AMSR2 retrieved snow–ice interface temperature to buoy snow–ice interface tem-
perature and (b) SLICE retrieved thermodynamic growth rate using buoy thickness to buoy thermodynamic growth rate. Linear correlations
are 0.70 and 0.71, respectively.

The SLICE retrieval method results were compared to data from ice mass balance buoys. First, the SLICE results and
buoy data were compared from an instantaneous perspective. The input snow–ice interface temperature retrieval results were
compared to buoy snow–ice interface temperature in Fig. 3a. The snow–ice interface temperature retrieval shows a mean bias
of 0.05 K with a standard deviation of 3.80 K. The linear correlation value between retrieved snow–ice interface temperature
and buoy snow–ice interface temperature is 0.70. This analysis does include the 2012G, 2012H, and 2012L buoys that were
also included in the training dataset used in creation of the multilinear regression snow–ice interface temperature retrieval
algorithm, because we don't intend to validate this retrieval, rather investigate sources of error in the SLICE results. Figure 3b

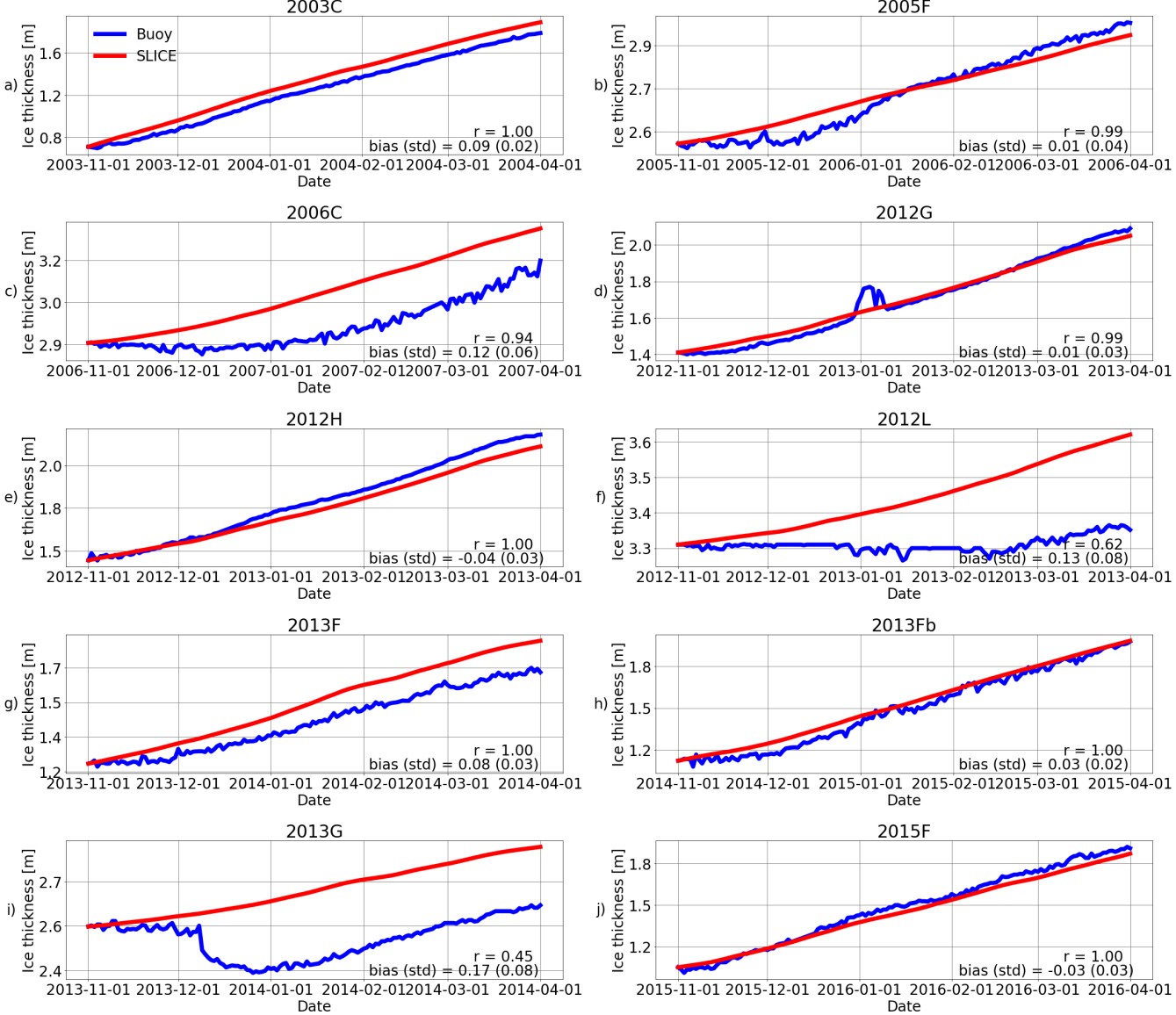

**Figure 4.** Ice thickness observations from ice mass balance buoys (blue) and SLICE (red) for buoys (a) 2003C, (b) 2005F, (c) 2006C, (d) 2012G, (e) 2012H, (f) 2012L, (g) 2013F, (h) 2013Fb, (i) 2013G and (j) 2015F. Linear correlation (r) and bias values are listed. Across all buoys, the r values have a mean of 0.89 and the biases have a mean of 0.06 m.

shows daily instantaneous SLICE thermodynamic growth rate retrieved via Eq. 8 using buoy thickness versus buoy provided daily thermodynamic growth rate. Because the CRREL IMB thickness data is provided to only the hundredths of a meter and daily variations in thickness are typically much small than this, a 14 day rolling average is applied to the buoy data. The linear

350

correlation of this comparison is 0.71 and SLICE shows a mean bias of 4 x $10^{-4}$ md$^{-1}$ with a standard deviation of 2.2 x $10^{-3}$ md$^{-1}$.

Next, SLICE was compared one-dimensionally to buoy data from a seasonal perspective. The retrieval method was initialized with the buoy observed sea ice thickness on 1 November and integrated through 1 April using Eq. 9. The results here are dependent only on the satellite based snow–ice interface temperature. The snow–ice interface temperature used on a given day is taken from the nearest AMSR-E or AMSR2 grid cell to the buoy location. The resultant sea ice thickness time series are plotted with buoy sea ice thickness in Fig. 4. It is clear from Fig. 4 that the SLICE profiles agree well with the buoy sea ice thickness when initialized with an accurate initial ice thickness. The correlation coefficients ranges from 0.45 to 0.999 with a mean of 0.89 and standard deviation of 0.21 across all buoys. The bias, calculated by taking the mean over the entire profile length of the retrieval method result minus the buoy thickness, ranges from -0.04 m to 0.17 m with a mean of 0.06 m and standard deviation of 0.07 m across the buoys.

## 4.2 Arctic Basin-wide Comparisons

Next, the SLICE retrieval method was utilized to model sea ice thickness on a Arctic basin-wide scale. Using the AWI CS2SMOS data for the first week of November as the initial state, the retrieval method was applied daily to the entire Arctic basin from 1 November to 1 April for the growth seasons beginning in 2012 through 2019. November first was chosen to ensure most ice was below the freezing point and there were limited melt ponds to interfere with the snow ice interface temperature observation. At each time step, the parcel tracking methodology described in Sect. 3.3 is applied. The 1 April results are regridded to the 25 km EASE-Grid 2.0 using the procedure also described in Sect. 3.3. Monthly basin-wide sea ice thickness plots for the sea ice growth season beginning in fall 2012 using AWI CS2SMOS as the initial state are shown in Fig. 5. The sea ice thickness data from SLICE is available daily but only the first of every month is plotted. The sea ice thickness is increasing and advection is relocating the volume horizontally throughout the Arctic basin.

In order to compare SLICE with PIOMAS and K21 data, all three datasets were compared with OIB sea ice thickness observations. OIB data from the month of March for the years 2013 through 2018 (including NSIDC OIB quick looks data) was first binned by 25 km EASE 2.0 grid cell and averaged across each bin to create an OIB dataset collocated with SLICE. Both PIOMAS and the K21 data were also interpolated to the 25 km EASE 2.0 grid. A comparison between SLICE, PIOMAS and K21 was created and shown in Fig. 6. Linear correlation and bias statistics were calculated from this data. All three datasets show very similar linear correlations of 0.64, 0.67 and 0.68 for SLICE, PIOMAS and K21, respectively. The best mean bias is PIOMAS at 0.10 m, followed closely by SLICE at -0.12 and K21 at -0.24 m. The standard deviation of the bias is 0.70 for SLICE, 0.68 for PIOMAS and 0.69 for K21.

SLICE was also compared to PIOMAS using AWI CS2SMOS as the reference dataset. Figure 7 shows the differences in sea ice thickness between SLICE and AWI CS2SMOS and the differences between PIOMAS and AWI CS2SMOS on 1 April for the years 2013 through 2020. Figure A2 in the Appendix shows plots of sea ice thickness used in this comparison from all three datasets. The differences are almost all between -1.5 m and 1.5 m and in most cases are near zero. The difference plots for SLICE and PIOMAS show similar patterns, though PIOMAS overestimates thickness in more areas than SLICE, which

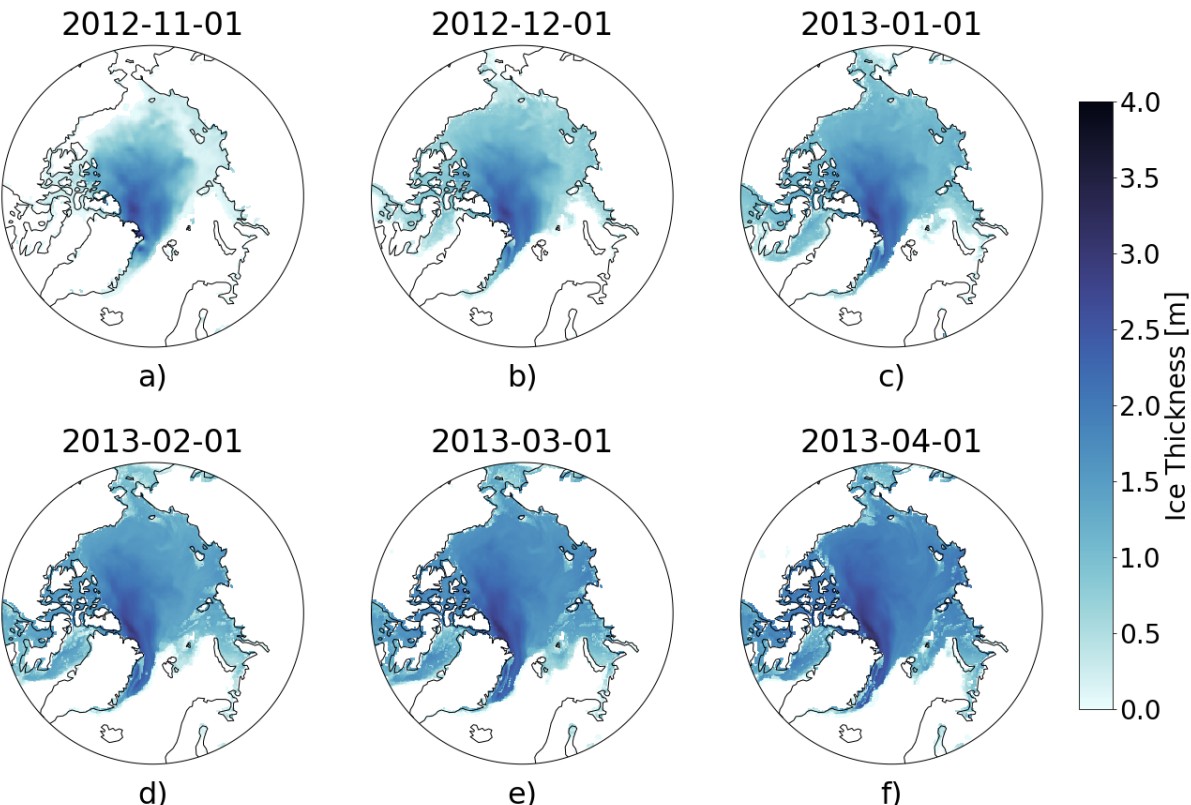

**Figure 5.** Sea ice thickness on a) 2 November 2012, b) 1 December 2012, c) 1 January 2013, d) 1 February 2012, d) 1 March 2013, e) 1 April 2013, f) 30 April 2013 created using SLICE with the 1 November 2012 AWI CS2SMOS as an initial state. The changes from month to month represent thermodynamic growth and advection.

underestimates sea ice thickness in the central Arctic in almost all cases. The differences between SLICE and CS2SMOS are likely to be due to a lack of deformation effects. Figure 8 shows a scatter plot of AWI CS2SMOS sea ice thickness to sea ice thickness from SLICE and from PIOMAS. Both AWI CS2SMOS and SLICE are on a 25 km EASE-Grid 2.0 but in order to compare the AWI CS2SMOS data to PIOMAS, it is interpolated to each PIOMAS grid point. Linear correlation and bias statistics were calculated from this data. SLICE and PIOMAS have similar linear correlations with AWI CS2SMOS of 0.76 and

0.77, respectively. SLICE shows a lower mean bias at 0.09 m compared to PIOMAS at 0.10 m. SLICE standard deviation of the difference is 0.63 m compared to 0.67 m for PIOMAS. Figure 9 shows total daily Arctic sea ice volume from SLICE, PIOMAS and AWI CS2SMOS during the winters from late 2012 to early 2020. The AWI CS2SMOS is taken from the the weekly data centered on each day. Both SLICE and PIOMAS follow the AWI CS2SMOS volume profile well. PIOMAS overestimates end of season volume in all years and underestimates initial volume in all cases except 2012-2013 and 2016-2017 leading to an

overestimation of sea ice volume growth in all years. SLICE begins each season at the same volume as AWI CS2SMOS and

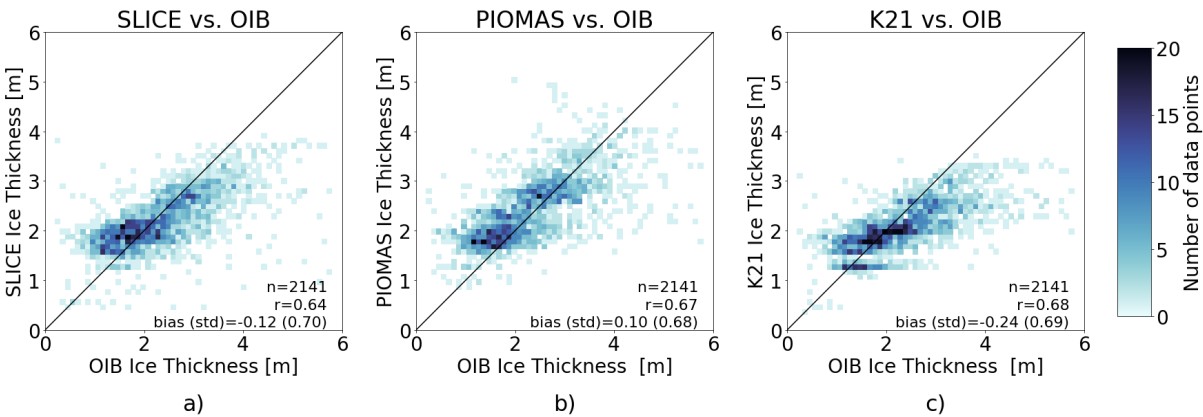

**Figure 6.** OIB thickness versus a) SLICE, b) PIOMAS and c) Kang et al., 2021 data including number of data points, linear correlations and bias (standard deviation). All three have similar linear correlations.

ends all seasons closer than PIOMAS to the AWI CS2SMOS volume except 2014-2015 and 2015-2016. While there are not enough data points for a strong statistical comparison, SLICE is certainly a viable indicator of sea ice volume growth.

## 5 Discussion

SLICE uses a retrieval of snow–ice interface temperature (Kilic et al., 2019) to drive a very simple one-dimensional model of sea ice thermodynamics in order to retrieve thermodynamic sea ice thickness growth. By applying SLICE to individual parcels whose location throughout the Arctic basin is determined using a sea ice motion product (Tschudi et al., 2020), SLICE is able to capture sea ice advection and produce basin-wide results. In doing so, SLICE functions similarly to much more intricate sea ice models such as PIOMAS (Zhang and Rothrock, 2003) and a sea ice model that is nudged with retrieved snow–ice interface temperature (Kang et al., 2021). While SLICE is capable of capturing thermodynamic sea ice growth and advection, it is unable to detect deformation effects—i.e., thickness changes due ridging or lead formation.

Figure 3 demonstrates SLICE's utility as a thermodynamic growth retrieval. The comparison between the input snow–ice interface retrieval and buoy snow–ice interface temperature shows that the retrieval is effective. The instantaneous SLICE thermodynamic growth rate calculated using buoy thickness shows a linear correlation of 0.71. The mean bias is $4 \times 10^{-4}$ $md^{-1}$ which is roughly 10% of the mean growth during the period. This bias is likely influenced by portions of the time series from buoys 2006C, 2012L and 2013G during which SLICE overestimates growth.

Figure 4 shows a comparison between ice mass balance buoy sea ice thickness measurements and the retrieval method initialized with the buoy data and integrated along an entire season for 10 buoys within the years 2003–2016. The mean correlation coefficient of 0.89 between the buoy measurements and the method is high. The bias values are also very encouraging with a mean of 0.06 m. Buoys 2006C, 2012L and 2013G show a SLICE profile that produces greater sea ice thickness than the buoys. There are likely two mechanisms causing this error. For buoys 2006C and 2012L, the initial thicknesses are the two highest of

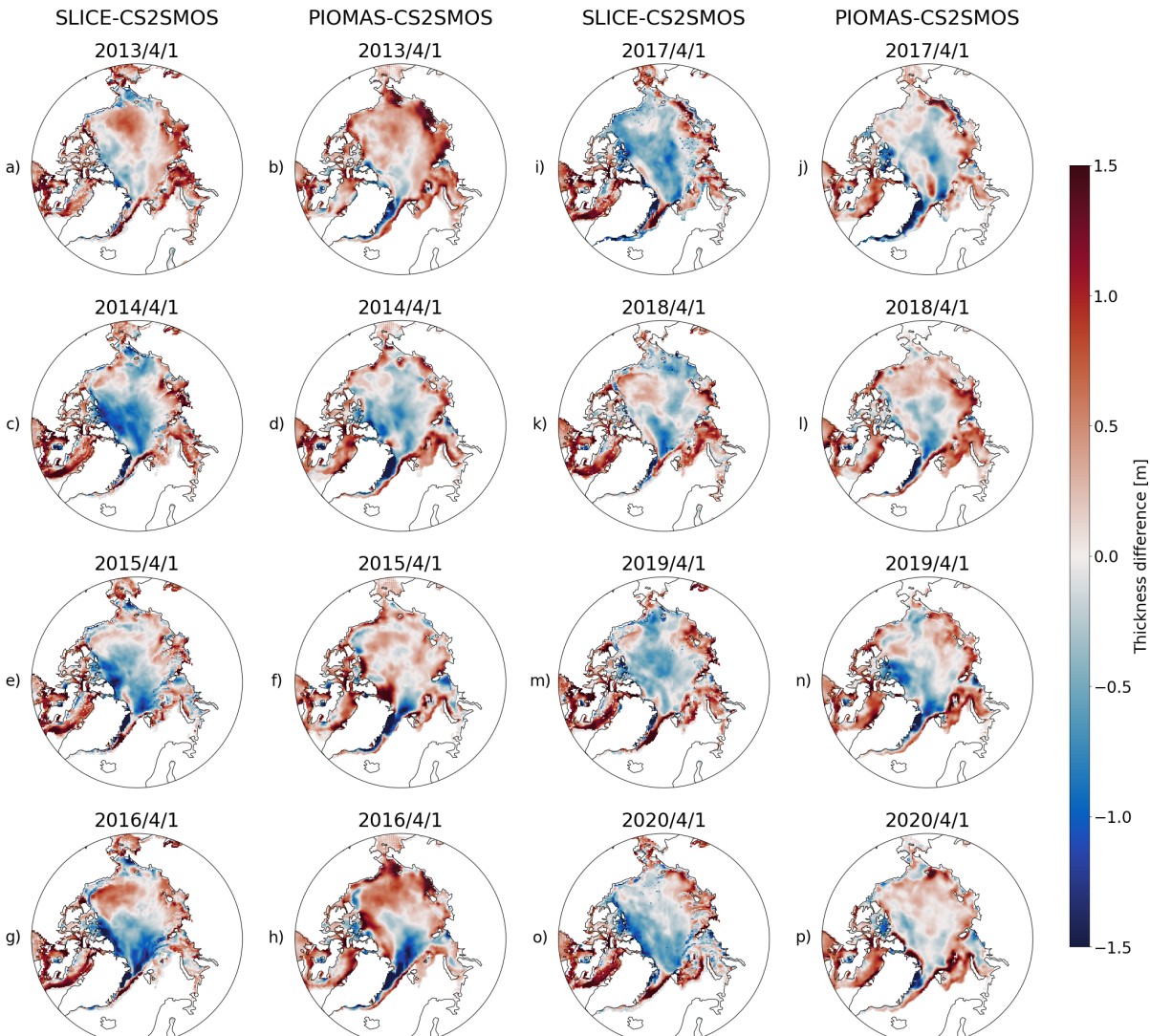

**Figure 7.** For the sea ice growth seasons ending in a–b) 2013, c–d) 2014, e–f) 2015, g–h) 2016, i–j) 2017, k–l) 2018, m–n) 2019 and o–p) 2020, a, c, e, g, i, k, m, o) SLICE sea ice thickness - AWI CS2SMOS sea ice thickness from 1 April and b, d, f, h, j, l, n, p) PIOMAS sea ice thickness - AWI CS2SMOS sea ice thickness from 1 April. The SLICE and PIOMAS differences show similarities in their overall pattern.

the set and are near 3 m. In these cases, the cold atmospheric temperatures of the growth season have not yet reached the base of the ice, which must be below the freezing point in order for thickness to increase. In other words, the heat stored in the ice from summer has not yet escaped due to the higher thickness and greater heat storing capacity. This means the ice does not yet have a linear temperature profile with the sea ice base at the freezing point, a condition that SLICE assumes but is not met in reality by buoys 2006C and 2012L until after 1 November. In the case of 2013G, a melt event, which SLICE is unable to capture, occurs in December. Both of these phenomena cause SLICE to overestimate sea ice thickness. When sea ice is indeed

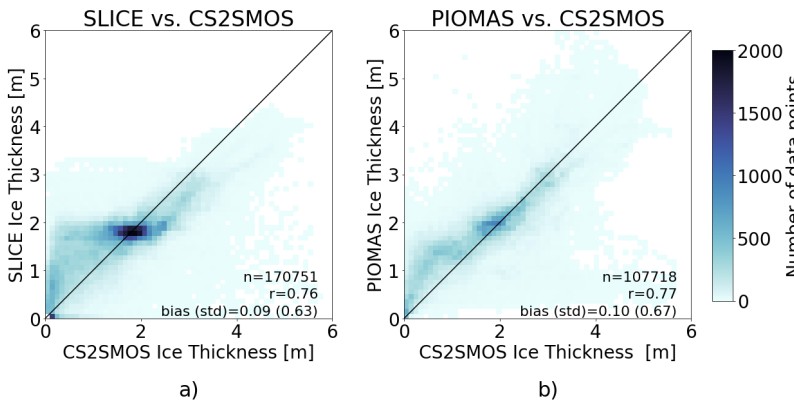

**Figure 8.** AWI CS2SMOS sea ice thickness versus a) SLICE sea ice thickness and b) PIOMAS sea ice thickness including number of data points, linear correlations and bias (standard deviation). SLICE and PIOMAS have nearly equal mean bias and linear correlation values.

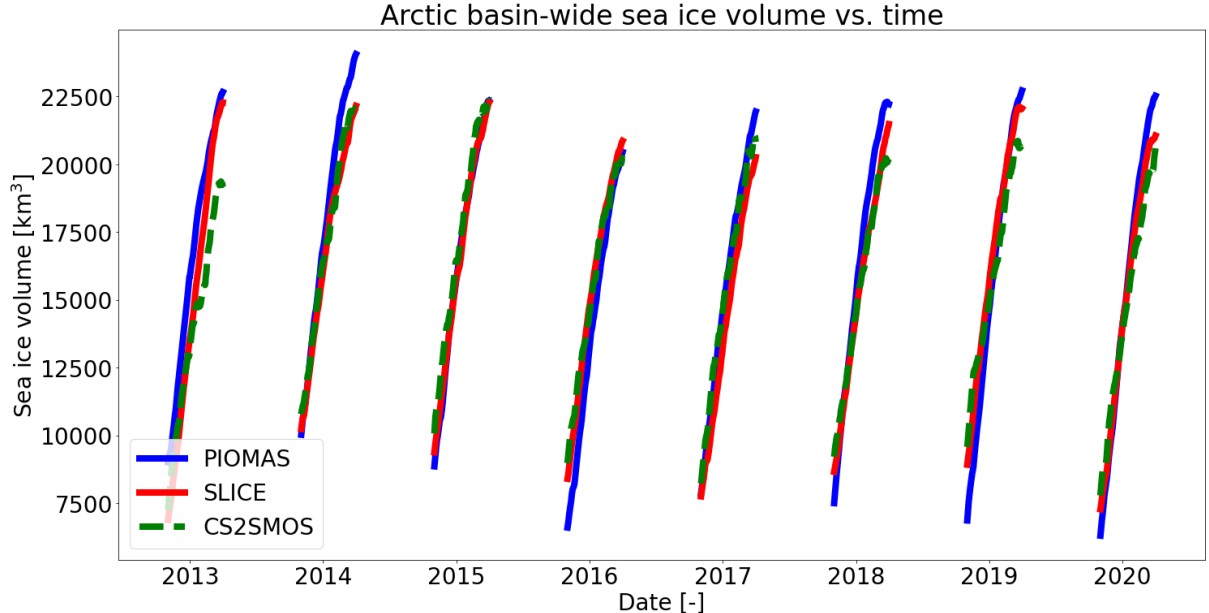

**Figure 9.** Wintertime sea ice volume versus time for SLICE, PIOMAS and AWI CS2SMOS.

increasing via thermodynamics, SLICE captures the growth well. Additionally, SLICE has a self-correcting quality by nature of Eq. (8) whereby sea ice thicknesses that are biased in either direction approach the unbiased SLICE sea ice thickness over time. These points suggest the retrieval method is viable as a basis for modelling sea ice thickness but is highly dependent on an initial condition, as it calculates thermodynamic sea ice thickness increase rather than absolute thickness.

There are a number of assumptions inherent to Stefan's Law (Lepparanta, 1993) that must be considered in relation to SLICE as introduced in Sect. 3.2. In order to characterize conduction through the ice layer with only the snow–ice interface

temperature and an assumed freezing point temperature at the bottom of the ice layer, it must be assumed that heat conduction in the horizontal is negligible and that the local vertical derivative of temperature throughout the ice layer is constant. These

assumptions are reasonable. The remaining two assumptions are more salient. The first is that there is no internal heat source. This is untrue when there is significant short wave radiation absorbed within the sea ice. The final assumption is that heat exchange between the sea ice and the ocean is constant in space and time, which is likely to be invalid in some regions. Impacts of this assumption on the sea ice growth are investigated in Sect. 3.2.2.

Another source of uncertainty in SLICE ice thickness is the constraint that it is limited to areas with sea ice concentration

greater than 95%. There is significant growth in areas where the sea ice concentration is low, such as the marginal ice zone (MIZ). This constraint would likely cause underestimated sea ice growth over those areas. Further validation of SLICE, particularly in regions other than the Beaufort Sea and Central Arctic, where all buoys and OIB flights used here were located and where the snow–ice interface temperature retrieval multi-linear regression was performed, as well as investigation of the impacts of these assumptions and full characterization of uncertainties is warranted.

The Stefan's Law energy balance relationship and attendant assumptions amounts to a simplification the multi-layer thermodynamic model based on Maykut and Untersteiner (1971) that makes up the foundations of PIOMAS and K21. These assumptions remove the need for multiple layers. Additionally, whereas the thermodynamics in PIOMAS and K21 are driven by an atmospheric reanalysis product and nudged by snow–ice interface temperature in the case of K21, SLICE is driven by satellite observed snow–ice interface temperature and not reliant on an atmospheric reanalysis product. These factors allow for

the retrieval of instantaneous thermodynamic thickness growth rate. Whereas PIOMAS models sea ice motion again using atmospheric reanalysis, SLICE uses a sea ice motion satellite product when used to model basin-wide sea ice thickness. This sea ice motion product and the snow–ice interface temperature product mean SLICE is heavily observationally constrained. The comparison between SLICE, PIOMAS and K21 show that these assumptions and simplifications do not significantly degrade resultant sea ice thickness values when SLICE thermodynamic growth rate is used to model absolute thickness on a basin-

wide scale. Indeed, thermodynamic growth rate calculated using average thickness over a 25 km x 25 km grid cell likely does not completely accurately describe thermodynamic growth rate over the entirety that grid cell. Additionally, SLICE does not include the effects of deformation processes, which recently were shown to have contributed roughly 30% of total thickness growth during the Multidisciplinary drifting Observatory for the Study of Arctic Climate (MOSAiC; Nicolaus et al., 2022) field campaign (von Albedyll et al., 2022; Koo et al., 2021).

Figure 9 is encouraging for the capability of SLICE to capture volumetric sea ice changes on a basin-wide scale. Per the model described by Eq. (1), sea ice volume is only added through thermodynamic processes—dynamic processes only serve to rearrange the volume already present. Though this statement does invoke the false assumption that dynamic processes do not change the density of the ice, it seems to be a factor in explaining the volumetric results. Though dynamic processes do not directly change sea ice volume within a given time step, their changing of the thickness of ice at a given location does

impact thermodynamic processes at later time steps by virtue of $f$ being a function of thickness, $H$, in Eq. (1). Inspection of Eq. (7) indeed shows that $H$ impacts $\frac{\partial H}{\partial t}$. In regions where deformation increases sea ice thickness, SLICE will overestimate sea ice thickness increase and in regions where deformation decreases sea ice thickness, it will underestimate sea ice thickness

increase. These phenomena, along with any phenomena inherent to either reference dataset, may explain volumetric differences between SLICE and the reference datasets. The 95% or greater criteria for SLICE may also contribute to differences, as the other datasets are not limited by this threshold.

## 6 Conclusions

New methods for observing snow–ice interface temperature (Kilic et al., 2019) have made possible a new strategy for observing thermodynamic sea ice thickness growth from space during the winter growth season: Stefan's Law Integrated Conducted Energy (SLICE). The new strategy involves linking observed satellite retrieved snow–ice interface temperature with Stefan's Law (Stefan, 1891; Lepparanta, 1993). In the Stefan's Law relationship, latent heat of fusion is conducted from the bottom of the ice layer where new ice forms to the snow–ice interface and this rate of conduction and accretion is calculated using the snow–ice interface temperature and a parameterized freezing point temperature at the bottom of the ice layer. The snow–ice interface temperature retrieval algorithm used to drive the sea ice thickness growth equation uses passive microwave brightness temperatures from the AMSR-E and AMSR2 instruments (Kilic et al., 2019). Gridded brightness temperature data from these instruments are available at daily temporal resolution in the polar regions (Cavalieri et al., 2014; Markus et al., 2018), meaning modelled daily sea ice thickness growth is available basin-wide. Lee et al. (2018) provides a method for retrieving snow–ice interface temperatures using passive microwave brightness temperatures from the SSM/I and SSMIS instruments, allowing for the application of SLICE to sea ice growth seasons beginning in 1987. SLICE requires an initial sea ice thickness value is required and does not capture melting.

When compared from an instantaneous perspective, SLICE is an effective retrieval of thermodynamic growth rate with a bias of 4 x $10^{-4}$ md$^{-1}$ relative to an ice mass balance buoy dataset when buoy thickness is used a priori at each time step. When SLICE is initialized with the ice mass balance buoy thickness on 1 November and integrated for an entire growth season, the retrieval method compares well with the buoy observed sea ice thickness growth. Using ten buoys from 2003 to 2016, the mean linear correlation value is 0.89 and the mean bias is 0.06 m. SLICE can be used to model basin-wide sea ice thickness by applying the thermodynamic growth retrieval to individual sea ice parcels and advected the parcels across the basin using a sea ice motion product. This basin-wide methodology was applied to the winters between late 2012 and early 2020 using AWI CS2SMOS basin-wide sea ice thickness as the initial state. Results show that SLICE performs comparably to PIOMAS and K21, despite the assumptions and simplification that allow for the direct retrieval of instantaneous thermodynamic growth rate.

Current state of the art sea ice thickness observations from space, though capable of observing sea ice growth whether from thermodynamic or dynamic effects, are not capable of this spatial and temporal coverage. They also do not discriminate between dynamic and thermodynamic effects. For these reasons, a sea ice thickness dataset based on SLICE will be especially qualified for investigating thermodynamic and dynamic sea ice phenomena that are small scale in space and time. SLICE need not be initialized at the beginning of the growth season and applied for an entire growth season but can be initialized at any time during the growth season and applied to any interval of time, allowing for use with case studies or other small time and

495 space scale events. Additionally, the high temporal resolution retrieval of thermodynamic effects will allow for creation of useful datasets of surface energy flux from latent heat of fusion.

*Code and data availability.* Data used in creation of all figures is available at https://doi.org/10.5281/zenodo.6985505. Code for creation of data and figures is available at https://doi.org/10.5281/zenodo.7143799 and https://github.com/janheuser/SLICE/releases/tag/3.0.0. The following auxilliary datasets were used and are available at these locations: AMSR-E and AMSR2 brightness temperatures, https://doi.org/10.
5067/AMSR-E/AE_SI25.003 and https://doi.org/10.5067/TRUIAL3WPAUP; AMSR-E and AMSR2 SIC, https://doi.org/10.5067/AMSR-E/AE_SI25.003 and https://doi.org/10.5067/TRUIAL3WPAUP; AWI CS2SMOS, https://www.meereisportal.de; sea ice motion vectors, https://doi.org/10.5067/INAWUWO7QH7B; OIB, https://doi.org/10.5067/G519SHCKWQV6 and https://doi.org/10.5067/GRIXZ91DE0L9; CRREL IMB, http://imb-crrel-dartmouth.org; PIOMAS, http://psc.apl.uw.edu/research/projects/arctic-sea-ice-volume-anomaly; Kang et al., 2021, https://doi.org/10.1029/2020MS002448.

**Appendix A**

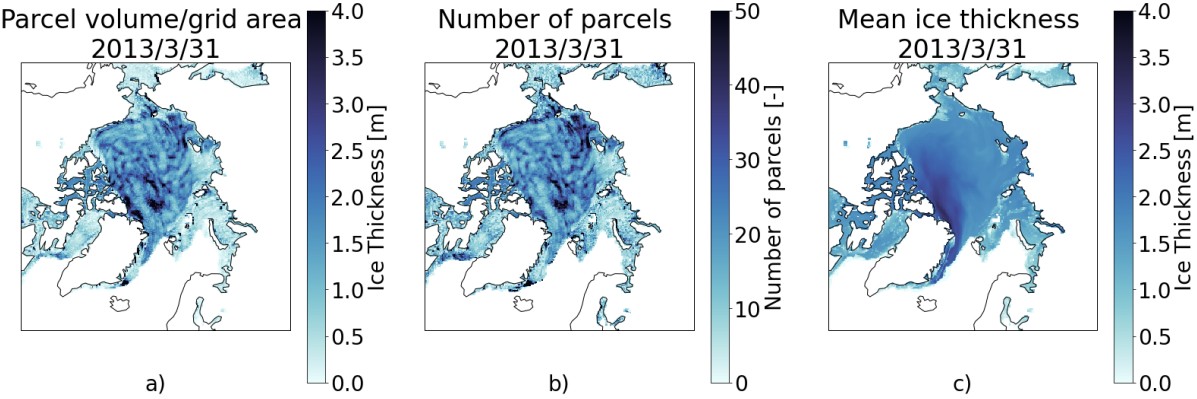

**Figure A1.** SLICE parcels on March 31 2013 (a) regridded using total parcel volume per grid cell divided by grid area, (b) counts within grid cell and (c) regridded mean parcel thickness within each grid cell. The volume per grid cell approach is unrealistic and dominated by erroneous convergence and divergence of parcels within grid cells.

*Author contributions.* All authors together conceived of the idea to use satellite retrieved snow–ice interface in a sea ice thickness satellite retrieval method. JA completed all analysis and wrote the first draft under guidance from YL and JK.

*Competing interests.* The authors declare that they have no conflict of interest.

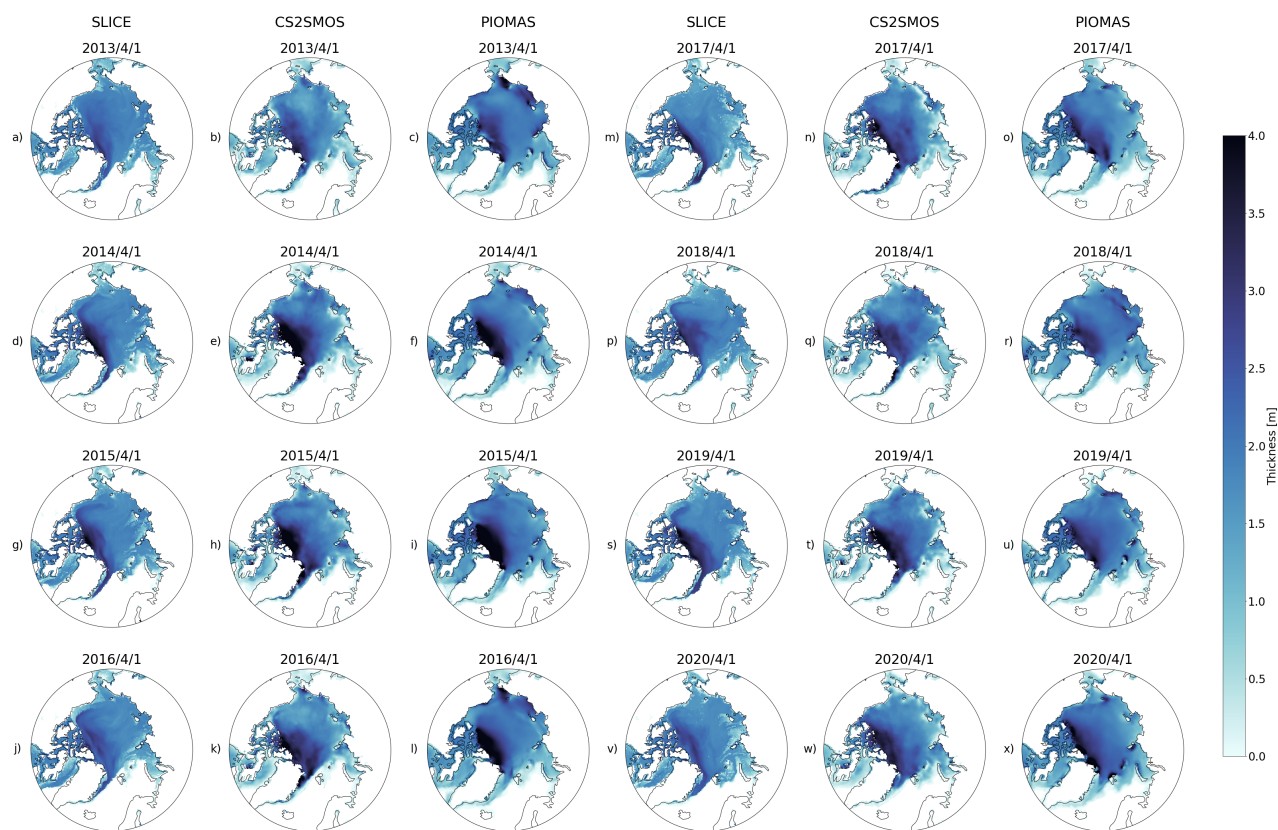

**Figure A2.** For the sea ice growth seasons ending in a–c) 2013, d–f) 2014, g–i) 2015, j–l) 2016, m–o) 2017, p–r) 2018, s–u) 2019 and v–x) 2020, a, d, g, j, m, p, s, v) SLICE sea ice thickness for 1 April, c, f, i, l, o, r, u, x) AWI CS2SMOS sea ice thickness from 1 April and b, e, h, k, n, q, t, w) PIOMAS sea ice thickness from 1 April.

*Acknowledgements.* This work was funded by the National Oceanic and Atmospheric Administration (NOAA) under grant no. NA20NES4320003. The views, opinions, and findings contained in this report are those of the author(s) and should not be construed as an official National Oceanic and Atmospheric Administration or U.S. Government position, policy, or decision. The merging of CryoSat-2 und SMOS data was funded by the ESA project SMOS & CryoSat-2 Sea Ice Data Product Processing and Dissemination Service and data from November 1st, 2012 to April 15th, 2021 were obtained from https://www.meereisportal.de (grant: REKLIM-2013-04).

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
