# Peer review of "A daily basin-wide sea ice thickness retrieval methodology: Stefan's Law Integrated Conducted Energy (SLICE)"

_The Cryosphere, 2021_

## Referee Comment (RC1)

**Review of: A daily basin-wide sea ice thickness retrieval methodology: Stefan's Law Integrated Conducted Energy (SLICE)**

**1 Synopsis**

The authors present a method of modelling thermodynamic sea ice growth based on the temperature of the snow-ice interface and assumptions involving the latent heat of fusion (SLICE). This was an original and interesting project, but I question whether it has reached the necessary level of completion to be published in The Cryosphere. In particular, I don't think the authors showed that their 'retrievals' (which are not made available) outperform a popular model (for which the data is publicly available): I believe this would affect the impact of the paper were it to be published.

I also take issue with the framing of SLICE as a 'retrieval', when I would argue it is more a model-output. Furthermore, I question the assertion that this exercise can be extended back in time in a useful sense, given that it requires initialisation with a separate product in a way that may not be practically possible pre-2000 as suggested. With regard to this, I also found it slightly strange that the authors stressed the near-term potential to improve the product by extending it back in time and including dynamical/advection based thickening, but didn't present either. Finally, I was concerned with the relatively undocumented application of a 5 K offset to the snow-ice interface temperatures, which are the fundamental data set underpinning the exercise.

Based on the above comments, I believe that the manuscript requires some major revisions and additions prior to any publication. Given the salience of the PIOMAS model within the paper, I strongly suggest the revisions include a comparison of the performance of SLICE against PIOMAS, explicitly evaluated against observations from a satellite product (or OIB). I believe the sea ice thickness community will only use this method where they can be clear about whether, when and where SLICE out-/under-performs PIOMAS. Should the editor move forward after receiving major revisions, I would like to review the manuscript again prior to publication.

**2 Significant Comments**

**2.1 Assessing the performance of a new sea ice thickness product**

Two key benchmarks for a new sea ice thickness product are (a) the degree to which the product outperforms a climatology, and (b) whether and in what ways the product outperforms comparable products - in this case probably PIOMAS.

To address point (a), I think the authors could do some fairly straightforward things. The first is to take the first two columns of Table 2 and correlate the anomalies. This would reveal whether SLICE is thicker when the sea ice is observed to be thicker. To put this another way - to what degree does SLICE capture the sign and size of volume anomalies from the mean state/ climatology. You could even break it down regionally into where it does & doesn't show skill.

To address (b) I think the users need to find a unique selling point over PIOMAS. To have impact, this product/method will need to be preferable in at least one way, either in availability or skill. The way I see it, PIOMAS has clear availability advantages: it is open-access so anybody can use it, and it stretches back to 1979. As for skill, it has the advantage of including both the dynamical and thermodynamical components of thickening. On the other hand, SLICE is based on fairly direct observations of the snow-ice interface temperature, whereas PIOMAS has to work this out by first modelling the snow and then calculating the temperature gradient across the snow and ice. To show that this translates into an actual skill advantage (and so to get people to use the method/product), I think the authors need to show that SLICE outperforms PIOMAS in some way, place or time. This is particularly the case because SLICE needs to be intialised

by a sea ice thickness data set anyway, which may well be PIOMAS. The best way to assess skill would be by benchmarking against (i.e. assuming as truth) some satellite-altimeter derived ice thickness product, or by combining the work they've done with other in-situ products like Operation Ice Bridge or ULS buoys. Without doing this, I don't think the SLICE-derived ice thickness will be greatly used by the community over PIOMAS.

**2.2 Retrieval vs modelling**

The term 'retrieval' touches on an emerging issue in the sea ice community concerning what properties we model, and what properties we observe/retrieve. For variables such as sea ice height, there is clearly a spectrum from direct observations (e.g. spot heights from a satellite-mounted laser altimeter) to highly-modelled (e.g. sea ice height output from a CMIP6-class model). Other quantities (e.g. radar-derived sea ice thickness from CryoSat-2, as used in this paper) are synthesised from observations (the timing and waveforms of scattered radar energy) and simple models (hydrostatic equilibrium, radar pulse propagation through snow, etc.). The case in this paper is similarly subjective: on one hand the authors are using observations of brightness temperatures, and what I would say is an observation of the snow-ice temperature. But then they're using a highly-idealised model of latent-heat release and heat flow known as Stefan's Law (which is in many ways is not a law but a series of combined thermodynamic assumptions, which are arguably outdated - see below).

Although I am certain this is not the intention of the authors, I fear that describing these sea ice thickness data as 'retrievals' implies a degree of direct observation that is too strong. I think that this implication may, at worst, lead to users (i.e. those wishing to initialise models) thinking that these data are more certain than they are, and more directly observed than they are. We regularly see this phenomenon with PIOMAS data for instance, which is very much a model but is treated by some as if it were observed because it is fed by reanalysis products. I therefore suggest that the authors be more explicit that they are modelling ice growth, and accumulating the results of that modelling exercise to model total ice thickness. On this basis I also urge them to remove the term 'retrieval' from their title.

**2.3 Long Term Applications**

The authors state that this method could be deployed several decades into the past. For instance, they do this in both their abstract and penultimate sentence. Their justification for this is that the snow-ice temperature is retrievable back to 1987, but I think that reconstruction back to this date is not usefully possible because initialisation is not available. It seems to me that the only way of doing this would be to initialise the product with an already existing and probably more accurate pan-Arctic sea ice thickness product. If this already exists, what would be the benefit of having this product, that would be dependent on (i.e. initialised by) the superior product? As a side point, I also fear that the authors' 5 K bias correction may not be relevant pre-2000, given than the roughness and snow depth of sea ice has declined, among other geophysical changes.

**2.4 5 K bias Correction**

On L82 the authors mention that they have performed a 5K bias correction on the Snow-Ice temperature data to make it match the buoy data. It's possible that they didn't do this themselves, but took it from a paper - but if so they should cite it. They certainly need to say whether they've added or subtracted the value. But this seems to be a pretty critical point that is not explored nearly enough. How did they get to this number? How sensitive is it to the data from individual buoys? How much did it improve the match between S-I temperature and the buoys? It also concerns me that they say they've done this 'to produce the best sea ice thickness retrievals'. Evaluated against what? If SIT data at the buoys has been used to tune or train the method, it casts doubt on the whole buoy-based evaluation exercise. The veracity and role of this correction must be quantified prior to publication, and its impact on the validity of the evaluation must be assessed.

**2.5   Ice-Ocean Boundary Conditions**

Seawater is not always in local thermal equilibrium with the sea ice interface (Schmidt et al., 2004; Mcphee, 2016). I'm not an expert on this, but it's relevant because this paper assumes equilibrium. For instance (as reported in McPhee), Maykut et al. (1971) found that without a steady basal flux of about $2\mathrm{Wm^{-1}}$, ice continued to grow unrealistically large in their model. Indeed Parkinson and Washington (1979) had to use a flux of an order of magnitude higher than this in their model. McPhee reports that observations from Sheba and Aidjex back these model fluxes up. This is clearly something the authors should address, perhaps with a sensitivity analysis to ocean-ice heat flux (which they say they've set to zero). If the 5 K offset discussed earlier was deployed to reduce modelled ice growth, perhaps the authors should consider that it is not the snow-ice temperatures being too low that are causing it, but an underestimation of ocean-ice heat flux?

**2.6   Sea ice is a mushy layer**

Sea ice is a mushy layer (Feltham et al., 2006) and this should be addressed when discussing heat flow through sea ice and accretion of new ice. Recently formed sea ice has brine inclusions, the phase equilibrium of which alters the bulk thermodynamic properties of the ice even well below the freezing temperature of seawater. Just stating what values you're using for the sea ice geophysical properties (L219) is insufficient. At minimum the values should be cited, and ideally they should be justified based on other previous modelling applications of the values. The constancy (as a function of temperature) of these values should also be considered. I'm not suggesting a multi-phase model of ice as I see that would make the whole situation very complicated and probably non-analytically soluble - the strength of SLICE is its simplicity. However, when presenting a model for ice growth based on heat flow through and phase change in ice near the freezing point, the mushy, mixed-phase characteristics of sea ice should be at least mentioned, and probably discussed.

**2.7   Data and code availability**

I was disappointed that the code and data used in this project were not made available to either the reviewers or the sea ice community. This is particularly the case given how much the authors have used other open data such as the CS2-SMOS and PIOMAS sea ice thickness data sets. To support this view, It's perhaps useful to refer to the data policy of this journal:

> The output of research is not only journal articles but also data sets, model code, samples, etc. Only the entire network of interconnected information can guarantee integrity, transparency, reuse, and reproducibility of scientific findings. Moreover, all of these resources provide great additional value in their own right. Hence, it is particularly important that data and other information underpinning the research findings are "findable, accessible, interoperable, and reusable" (FAIR) not only for humans but also for machines.

I would recommend that upon resubmission they make their code available on a site such as GitHub, and produce a persistent identifier such as a DOI. I also suggest they place their data product in a persistent archive such as that run by Zenodo, for which they will receive a DOI and the opportunity to reversion the data upon article acceptance. In taking the above steps, I believe the authors will significantly increase the impact of their research.

**3   Other Comments**

L2: 'Coupling'. I feel that 'coupled' systems/equations generally exchange information with and influence each other. However it seems that in this case you're feeding satellite information on the snow-ice interface temperature to an equation which tells you the growth rate. The satellite algorithm is not dependent on Eq. 7. So I think you should avoid portraying this as a coupled system; perhaps something like 'linking', or 'feeding'?

L34: I think "is also effective" is subjective and should be changed. Perhaps "is also popular"?.

L46: I think the word 'promising' is subjective and should be removed.

L63: Should be polarization, not polarity I think?

L113: "Obvious dynamic effects" - what does this mean? I think you need to be clearer in this paper between dynamic thickening in a Lagrangian sense (i.e. convergence driven ridging and rafting of ice to make it thicker), and dynamical thickening in an Eulerian sense (advection of thicker ice into and thinner ice out of a grid cell).

L127: The snow loading is used before the hydrostatic conversion, in the calculation of the height of the ice surface above the waterline to account for the delay in radar propagation through the snow (e.g. Mallett et al., 2020).

L129 CPOM is not affiliated with ESA

L142: Complementing, not complimenting

L150: It's noticeable that the grid on which data are supplied and applied is consistently described up until the PIOMAS description. This is perhaps the most important data set for which to mention this, because the native grid is very unusual. Worth describing or not describing the grids consistently.

L360: Antarctic sea ice floes often have negative freeboards so you probably won't retrieve get the snow-ice interface temperature. Some floes have had them in the past leading to the formation of snow-ice, and ice lenses also exist in the snow, which I imagine will significantly complicate the retrieval of the snow-ice interface temperature. Indeed the potential for negative freeboards in the Arctic (Merkouriadi et al., 2020) should perhaps be mentioned at some point.

L374: It's my opinion that you'll only be 'retrieving' sea ice thickness when you do actually account for both thermodynamic and dynamic/advective contributions to sea ice thickness at a point. Right now I'd say you're modelling one part of it.

**3.1 Figures and Tables**

The map projections used in this paper were unusual and not well-suited to the data being displayed. They look a bit like a Near-Sided Perspective projection? In any case, I think a more traditional North-Polar-Stereographic or Lambert Azimuthal-Equal-Area projection would be better. It looks in this case like data nearer the pole is being over-represented in area, and it's concerning that Hudson and Baffin Bay are hidden and highly distorted respectively.

I also think a figure should be displayed complementing Table 1 (perhaps put in a supplement?) with the tracks of the buoys used to evaluate SLICE. This would give the reader a better sense of the geographic/spatial validity of the buoy-based evaluation presented.

Figure 1: The colorbar should be labelled with the variable (S-I Temp), and units (Kelvin) should be stated.

Figure 5: The blue/white plots aren't providing much narrative value here. They're similar in appearance and concept to Fig 4, and the panels often look very similar to each other; I would suggest putting them in a supplement and increasing the size of the difference plots, which are much more relevant and important.

Table 2: I think put this in a supplement and display the data as a timeseries. You could put the Vol. Growth in first two columns on the Y axis and the relative difference in % on a secondary Y axis. I'm not convinced the column with absolute differences adds much value. I think displaying this data as a graph

would give the reader a much better feel for what's going on.

Figure 6: Again, enlarge and focus on the difference plots and put the blue/white plots in a supplement.

Table 3: Same comment as Table 2, and you could probably merge the resulting figures.

**References**

Feltham, D. L., Untersteiner, N., Wettlaufer, J. S., and Worster, M. G.: Sea ice is a mushy layer, Geophysical Research Letters, 33, 4–7, https://doi.org/10.1029/2006GL026290, 2006.

Mallett, R. D. C., Lawrence, I. R., Stroeve, J. C., Landy, J. C., and Tsamados, M.: Brief communication: Conventional assumptions involving the speed of radar waves in snow introduce systematic underestimates to sea ice thickness and seasonal growth rate estimates, Cryosphere, 14, 251–260, https://doi.org/10.5194/tc-14-251-2020, 2020.

Maykut, G. A., Untersteiner, N., MAYKUT GA, and UNTERSTEINER N: Some results from a time- dependent thermodynamic model of sea ice, J Geophys Res, 76, 1550–1575, https://doi.org/10.1029/jc076i006p01550, 1971.

Mcphee, M. G.: The sea ice–ocean boundary layer, Sea Ice: Third Edition, pp. 138–159, https://doi.org/10.1002/9781118778371.CH5, 2016.

Merkouriadi, I., Liston, G. E., Graham, R. M., and Granskog, M. A.: Quantifying the Potential for Snow-Ice Formation in the Arctic Ocean, Geophysical Research Letters, 47, e2019GL085 020, https://doi.org/10.1029/2019GL085020, URL https://onlinelibrary.wiley.com/doi/abs/10.1029/2019GL085020, 2020.

Parkinson, C. L. and Washington, W. M.: A large-scale numerical model of sea ice, Journal of Geophysical Research: Oceans, 84, 311–337, https://doi.org/10.1029/JC084IC01P00311, 1979.

Schmidt, G. A., Bitz, C. M., Mikolajewicz, U., and Tremblay, L. B.: Ice–ocean boundary conditions for coupled models, Ocean Modelling, 7, 59–74, https://doi.org/10.1016/S1463-5003(03)00030-1, 2004.

---

## Referee Comment (RC2)

**A review of "A daily basin-wide sea ice thickness retrieval methodology: Stefan's Law Integrated Conducted Energy (SLICE)" by James Anheuser et al.**

**General Comments**

The authors introduced a new method for sea ice thickness estimation from satellite snow-ice interface temperature ($T_{si}$) by using idealized sea ice thermodynamic model. The key idea of their methodology is that thermodynamic sea ice growth rate can be calculated from upward conductive heat flux within the sea ice layer which balances with the latent heat of fusion. In their method, the conductive heat flux is a function of the $T_{si}$ under the linear temperature profile assumption. Therefore, sea ice thickness can be calculated from $T_{si}$ with appropriate initial ice thickness. Furthermore, the authors insist that the introduced method is self-correcting.

However, I have some major concerns about the introduced method. 1) There should be clarifications on the physical conditions (regions and seasons) that meet with the four assumptions they made. 2) More explanation is needed to insist that the method is "effectively" self-correcting. 3) The method seems to be a modeling approach rather than satellite retrieval. 4) Detailed procedure for the bias correction of the satellite $T_{si}$ must be provided.

Associated with the major concerns above, I think there should be significant improvements on the data and methodology before the manuscript is published to The Cryosphere. Therefore, my decision is to reconsider after the major revision. I would like to review the manuscript again after the revision.

**Major comments**

**1. Assumptions in the SLICE method**

From L202 to L208, the authors listed the four assumptions used in the SLICE retrieval method. I have concerns about the second and the third assumptions. In my opinion, the second assumption is equivalent to the statement that the temperature profile of sea ice is linear. But if you see the buoy measured temperature profiles, you will find this assumption is not always valid. Such linear profile assumption is generally valid during wintertime. Moreover, even during wintertime, sudden change in air temperature due to warm/cold advection or radiative forcing due to cloud cover can rapidly change surface temperature which makes curves in the temperature profile. The good thing is time-averaged temperature profile during wintertime is close to linear (Shi et al., 2020). The authors would consider shortening of retrieval period of the SLICE method.

The third assumption tells that there is no internal heat source associated with shortwave radiation. In other words, this assumption is valid for the regions where the solar zenith angle is maintained less than zero. The authors should check the validity of this assumption regarding the seasonal variability of the solar zenith angle. There can be sunlight in lower latitude regions during fall and spring. Otherwise, please consider the shortwave radiation effects or justify that the shortwave radiation effect is negligible for the lower latitude regions during the fall and spring seasons.

The other point is that the authors mentioned that the retrieval method should be applied in a Lagrangian sense in L224 but they neglected sea ice motion in the actual calculation (L262). What are the reasons for this? There must be justification for the neglect of sea ice motion. Each sea ice parcel should be tracked and matched with the nearest satellite $T_{si}$ because the equation used in this study is a time-dependent equation. Meanwhile, the neglect of sea ice motion is not the same as focusing on thermodynamic growth. Thermodynamical growth, sea ice motion, and dynamical growth (deformation due to convergence and divergence) should be addressed separately. Consideration of sea ice motion without dynamical growth is possible.

**2. Effectiveness of self-correcting characteristic**

It was interesting to read the statement in L225 regarding the self-correcting characteristic of the SLICE method. Thicker sea ice indeed grows slower than thinner sea ice with a given $T_{si}$ and vice versa according to equation (7). Therefore, the error in sea ice thickness can be relaxed by the modulation of sea ice growth speed.

However, the relaxation speed of error is important as well. If the speed of relaxation is slow, the effectiveness of self-correcting characteristics will be minor and the initial condition will be the major factor that determines the accuracy of sea ice thickness estimation. In L249-250 and Figure 2, the authors tried to show the effect of the self-correcting characteristic. Although it seems that 0.25 m deviations in the initial condition are decreasing with time, it will be better to specify the improvement quantitatively to know how fast the errors are relaxed. In addition, I suggest conducting a sensitivity test and including the result as an appendix.

I found some doubtful points on the self-correcting characteristic of the SLICE method. In my opinion, if the method is self-correcting, the retrieval result should fluctuate around the true state. Why is the SLICE retrieval (red solid line) the center of red shade instead of the buoy (blue solid line) which is the true state? In addition, I think the sentence "The bias grows with time as the SLICE profile moves away from its initialized thickness" makes a contradiction with the self-correcting characteristic of SLICE.

The significance of self-correcting characteristic is important for the algorithm extension to the past because such characteristic makes the retrieval method relatively independent from the accurate initial condition. If the self-correction is significant, SLICE sea ice thickness records initialized with PIOMAS can be constructed, and it will be more accurate than PIOMAS. To examine this, I suggest comparing the accuracy of the sea ice thickness from the PIOMAS and that from the SLICE initialized with the PIOMAS. There are some widely used independent

datasets for validation such as Operation IceBridge (OIB), buoy, upward-looking sonar (ULS), and submarine observations.

**3. Retrieval or modeling (significance of this study)**

In some sense, the SLICE retrieval method seems to be a thermodynamic sea ice model. The reason is that it simulates sea ice thickness evolution with time, and the result of SLICE retrieval is highly dependent on initial conditions rather than observed data. I think that the SLICE method is a simplified version of the thermodynamic sea ice model introduced by Maykut and Unterstiener (1971) or the PIOMAS. It will be nice for the authors to explain why the SLICE method is satellite retrieval.

Nonetheless, the novel point of this study is SLICE method is independent of the atmospheric reanalysis generally used as the forcing to sea ice model. The most relevant study to the SLICE method will be Kang et al. (2021), which simulates the physical state of a snow-ice system by using a thermodynamic equation set forced by atmospheric reanalysis and nudged by satellite $T_{si}$. This study has significance in terms of constructing an independent sea ice thickness record, while the physics of SLICE is very simplified compared to Kang et al. (2021) or other thermodynamic sea ice models. I recommend including an ice thickness comparison with the results of Kang et al. (2021). Their results are open to the public, and the authors can find the data repository in their paper. It is worth comparing the performance of the SLICE method with other sea ice models with more sophisticated physics and forced by reanalysis data.

**4. Bias correction for satellite $T_{si}$**

The authors mentioned that "The resultant snow-ice interface temperatures were found to require a bias correction of 5 K in order to match buoy snow-ice interface temperatures…"). I have read Lee and Sohn (2015) and remember that the snow-ice interface derived from AMSR-E 6.9 GHz brightness temperatures are validated with buoy measured temperature. The validation result showed that the bias was less than 1 K, which is a very different result from the 5 K bias in the manuscript. Lee and Sohn (2015) also neglected atmospheric/snow absorption.

Regarding this situation, first I thought that it is possibly due to the bias within AMSR-E and AMSR2 measurements. However, the authors stated that the AMSR2 data has been intercalibrated with the AMSR-E data so this may not be the issue. Then, may the version of L3 brightness temperature be a problem? Or simply authors failed to reproduce the $T_{si}$ retrieval algorithm.

This is a very critical issue because sea ice thickness is determined by $T_{si}$, which is the only real observation used for the sea ice thickness retrieval. The mentioned comparison result between buoy data and $T_{si}$ calculated by the authors showing 5 K bias must be presented (as an appendix) to justify the bias correction procedure. It will be worth reproducing figure 6 in Lee

and Sohn (2015).

**Minor comments**

L29-L37: Please provide more details for relevant studies on sea ice thickness retrieval in order to emphasize the novelty or necessity of SLICE. How are the satellite altimetry methods limited in spatial coverage and temporal resolution (I think the resolution of ICESat-2 is better than passive microwave sensors such as AMSR2 6.9 GHz)? What are the limitations of the other methods? How is this study related to the existing studies?

L63: horizontally and vertically polarized…

L215: Please define negative degree-days in the manuscript and provide what happens if the temperature is positive (melting?).

L221: It is hard to know which equation was used for sea ice thickness calculation. Equation (4) is too general. Did you use equation (8) which is an analytic solution for sea ice thickness, or equation (7) for change in sea ice thickness per unit time and accumulate the thickness changes?

L235-237: Why the retrieval method was initialized with such condition (the day when the 14 d rolling average sea ice growth exceeded 1mm d$^{-1}$)? Please provide the reason.

L400: I think uploading the data produced in this study to the public data repository more fits the data policy of TC journal.

**References**

Kang, E. J., Sohn, B. J., Tonboe, R. T., Dybkjaer, G., Holmlund, K., Kim, J. M., and Liu, C.: Implementation of a 1-D Thermodynamic Model for Simulating the Winter-Time Evolvement of Physical Properties of Snow and Ice Over the Arctic Ocean, J. Adv. Model. Earth Sy., 13, 3, https://doi.org/10.1029/2020ms002448, 2021.

Lee, S.-M and Sohn, B.-J.: Retrieving the refractive index, emissivity, and surface temperature of polar sea ice from 6.9 GHz microwave measurements: A theoretical development, J. Geophys. Res. Atmos., 120(6), 2293-2305, https://doi.org/10.1002/2014JD022481, 2015.

Maykut, G. A. and Untersteiner, N.: Some results from a time-dependent thermodynamic model of sea ice, J. Geophys. Res., 76, 1550–1575, https://doi.org/10.1029/JC076i006p01550, 1971.

Shi, H., Sohn, B.-J., Dybkjær, G., Tonboe, R. T., and Lee, S.-M.: Simultaneous estimation of wintertime sea ice thickness and snow depth from space-borne freeboard measurements, The Cryosphere, 14, 3761–3783, https://doi.org/10.5194/tc-14-3761-2020, 2020.

---

## Author Comment (AC1)

**Review of: A daily basin-wide sea ice thickness retrieval methodology: Stefan's Law Integrated Conducted Energy (SLICE)**

The authors wish to thank the reviewer for the very thoughtful and helpful review. We have responded in red font to individual comments below where necessary.

1 Synopsis

The authors present a method of modelling thermodynamic sea ice growth based on the temperature of the snow-ice interface and assumptions involving the latent heat of fusion (SLICE). This was an original and interesting project, but I question whether it has reached the necessary level of completion to be published in The Cryosphere. In particular, I don't think the authors showed that their 'retrievals' (which are not made available) outperform a popular model (for which the data is publicly available): I believe this would affect the impact of the paper were it to be published.

I also take issue with the framing of SLICE as a 'retrieval', when I would argue it is more a model-output. Furthermore, I question the assertion that this exercise can be extended back in time in a useful sense, given that it requires initialisation with a separate product in a way that may not be practically possible pre-2000 as suggested. With regard to this, I also found it slightly strange that the authors stressed the near-term potential to improve the product by extending it back in time and including dynamical/advection based thickening, but didn't present either. Finally, I was concerned with the relatively undocumented application of a 5 K offset to the snow-ice interface temperatures, which are the fundamental data set underpinning the exercise.

Based on the above comments, I believe that the manuscript requires some major revisions and additions prior to any publication. Given the salience of the PIOMAS model within the paper, I strongly suggest the revisions include a comparison of the performance of SLICE against PIOMAS, explicitly evaluated against observations from a satellite product (or OIB). I believe the sea ice thickness community will only use this method where they can be clear about whether, when and where SLICE out-/under-performs PIOMAS. Should the editor move forward after receiving major revisions, I would like to review the manuscript again prior to publication.

2 Significant Comments
2.1 Assessing the performance of a new sea ice thickness product

Two key benchmarks for a new sea ice thickness product are (a) the degree to which the product outperforms a climatology, and (b) whether and in what ways the product outperforms comparable products - in this case probably PIOMAS.

To address point (a), I think the authors could do some fairly straightforward things. The first is to take the first two columns of Table 2 and correlate the anomalies. This would reveal whether SLICE is thicker when the sea ice is observed to be thicker. To put this another way - to what degree does SLICE capture the sign and size of volume anomalies from the mean state/ climatology. You could even break it down regionally into where it does & doesn't show skill.

We have completed this analysis and found that the p-value of the correlation between the SLICE volume growth and the CryoSat-2/SMOS volume growth is too high to report the results as significant.

To address (b) I think the users need to find a unique selling point over PIOMAS. To have impact, this product/method will need to be preferable in at least one way, either in availability or skill. The way I see it, PIOMAS has clear availability advantages: it is open-access so anybody can use it, and it stretches back to 1979. As for skill, it has the advantage of including both the dynamical and thermodynamical components of thickening. On the other hand, SLICE is based on fairly direct observations of the snow-ice interface temperature, whereas PIOMAS has to work this out by first modelling the snow and then calculating the temperature gradient across the snow and ice. To show that this translates into an actual skill advantage (and so

to get people to use the method/product), I think the authors need to show that SLICE outperforms PIOMAS in some way, place or time. This is particularly the case because SLICE needs to be intialised by a sea ice thickness data set anyway, which may well be PIOMAS. The best way to assess skill would be by benchmarking against (i.e. assuming as truth) some satellite-altimeter derived ice thickness product, or by combining the work they've done with other in-situ products like Operation Ice Bridge or ULS buoys. Without doing this, I don't think the SLICE-derived ice thickness will be greatly used by the community over PIOMAS.

We will add a comparison to the paper of SLICE initialized with CS2SMOS, PIOMAS and the model described by Kang et al. 2021 (hereafter K21) to Operation Ice Bridge (OIB) data (Kurtz, 2015).

The SLICE basin-wide component now includes advection of sea ice parcels using the NSIDC Polar Pathfinder daily sea ice drift product (Tschudi et al., 2019). We had planned to include this element in a future study but at the request of Referee #2, we have chosen to include advection in this paper. SLICE is initialized with the CS2SMOS data or PIOMAS (interpolated to the 25 km EASE grid 2.0) from the first week of November and each 25 km x 25 km grid cell is divided into 5 km x 5km parcels, which are advected daily using the motion vectors interpolated to their position and who add sea ice thickness thermodynamically using the SLICE thermodynamic model. As before, new ice per a sea ice concentration product is initialized at 0.01 m. At any given time step, the parcels can be gridded back to the EASE grid 2.0 grid by taking the mean of parcels within each EASE grid. This process will be included in all basin-wide results shown in the revised manuscript.

OIB data from the month of March for the years 2013 through 2018 (including NSIDC OIB quick looks data) was first binned by SLICE grid cell and averaged across each bin to create an OIB dataset collocated with SLICE. Both PIOMAS and the Kang et al. 2021 data were also interpolated to the SLICE grid. Using only SLICE grid cells containing 100 or more individual OIB sea ice thickness data points, a comparison between SLICE initialized CS2SMOS, PIOMAS and K21 was created and shown in Figure 1.

[Figure]

*Figure 1: OIB thickness versus (a) SLICE intilialized with CryoSat-2/SMOS, (b) PIOMAS and (c) Kang et al., 2021 data including number of data points, linear correlations and bias (standard deviation). SLICE has the highest linear correlation though all three are virtually equal.*

The highest linear correlation value belongs to SLICE at 0.704, however the linear correlation for PIOMAS and K21 are very near that value at 0.700 and 0.699 respectively. The smallest mean (standard deviation) bias is exhibited by PIOMAS at -0.050 m (0.629 m) followed by SLICE with 0.171 m (0.628 m) and K21 with 0.307 m (0.647 m). This analysis will be included in the revised manuscript.

These statistics show that all three models have similar performance when modeling sea ice thickness, even without SLICE including a deformation component. The differences are related to complexity of the model and reliance upon model reanalysis data. Whereas both PIOMAS and K21 require snow information and must

calculate the temperature profile in the snow in order to determine the temperature profile in the ice from a reanalysis product, SLICE uses direct retrieval of the snow—ice interface temperature in order to calculate the heat flux through the ice and therefore thermodynamic sea ice growth. By assuming a linear temperature profile in the sea ice, SLICE also removes the requirement for multiple ice layers to be tracked by the model.

We don't believe SLICE to be a replacement for existing sea ice thickness retrievals, rather an additional independent dataset created using an observationally constrained very simple model that may be more applicable in certain situations.

**2.2 Retrieval vs modelling**

The term 'retrieval' touches on an emerging issue in the sea ice community concerning what properties we model, and what properties we observe/retrieve. For variables such as sea ice height, there is clearly a spectrum from direct observations (e.g. spot heights from a satellite-mounted laser altimeter) to highly- modelled (e.g. sea ice height output from a CMIP6-class model). Other quantities (e.g. radar-derived sea ice thickness from CryoSat-2, as used in this paper) are synthesised from observations (the timing and waveforms of scattered radar energy) and simple models (hydrostatic equilibrium, radar pulse propagation through snow, etc.). The case in this paper is similarly subjective: on one hand the authors are using observations of brightness temperatures, and what I would say is an observation of the snow-ice temperature. But then they're using a highly-idealised model of latent-heat release and heat flow known as Stefan's Law (which is in many ways is not a law but a series of combined thermodynamic assumptions, which are arguably outdated - see below).

Although I am certain this is not the intention of the authors, I fear that describing these sea ice thickness data as 'retrievals' implies a degree of direct observation that is too strong. I think that this implication may, at worst, lead to users (i.e. those wishing to initialise models) thinking that these data are more certain than they are, and more directly observed than they are. We regularly see this phenomenon with PIOMAS data for instance, which is very much a model but is treated by some as if it were observed because it is fed by reanalysis products. I therefore suggest that the authors be more explicit that they are modelling ice growth, and accumulating the results of that modelling exercise to model total ice thickness. On this basis I also urge them to remove the term 'retrieval' from their title.

The authors broadly agree with the points made here. Certainly, all retrievals rely on a model at some level. For instance, even the spot heights from a laser altimeter referred to by the reviewer are based on a very simple model converting signal response times to spot heights. We believe SLICE sits in between these types of methods that are accepted as retrievals and results from models such as PIOMAS and that this is indeed the strength of SLICE.

As stated by the reviewer, the most direct output from SLICE is a thermodynamic growth rate (and conducted flux through the ice). Much like many accepted retrievals, this output relies upon a priori information--sea ice thickness, freezing point temperature, etc. We believe this step of the process can be considered a retrieval based on a simple model. We concede that accumulating the sea ice growth into an absolute sea ice thickness is more of a modeling exercise, albeit one that is heavily observationally constrained. We will make this clear in the next revision.

We propose the following title:

"A simple model for daily basin-wide thermodynamic sea ice thickness growth retrieval"

**2.3 Long Term Applications**

The authors state that this method could be deployed several decades into the past. For instance, they do this in both their abstract and penultimate sentence. Their justification for this is that the snow-ice temperature is retrievable back to 1987, but I think that reconstruction back to this date is not usefully possible because initialisation is not available. It seems to me that the only way of doing this would be to initialise the product with an already existing and probably more accurate pan-Arctic sea ice thickness product. If this already exists,

what would be the benefit of having this product, that would be dependent on (i.e. initialised by) the superior product? As a side point, I also fear that the authors' 5 K bias correction may not be relevant pre-2000, given than the roughness and snow depth of sea ice has declined, among other geophysical changes.

We will remove allusions to the long term application of SLICE and instead leave that for future investigation.

Lee and Sohn (2015) investigated the impact of surface roughness on their snow—ice interface temperature retrieval. They report typical sea ice surface roughness figures of at maximum 2.5 mm. With this maximum in mind, they applied a Bragg scattering model to find the sensitivity of the snow—ice interface emissivity to surface roughness values of 0 to 3 mm. In the case of 3 mm surface roughness, the impact to horizontal polarization emissivity is 3.5% while the impact to vertical polarization emissivity is nearly zero. For long term applications, we will use the vertical emissivity and vertical brightness temperatures. Additionally, they note that snow in the Arctic region is transparent to the low frequency channels of the AMSR-E and AMSR2 instrument (Matthew, et al., 2009). Lastly, atmospheric changes over time will be accounted for with the inclusion of a radiative transfer model based replacement for the 5 K bias correction (more info in the following section).

2.4 5 K bias Correction

On L82 the authors mention that they have performed a 5K bias correction on the Snow-Ice temperature data to make it match the buoy data. It's possible that they didn't do this themselves, but took it from a paper - but if so they should cite it. They certainly need to say whether they've added or subtracted the value. But this seems to be a pretty critical point that is not explored nearly enough. How did they get to this number? How sensitive is it to the data from individual buoys? How much did it improve the match between S-I temperature and the buoys? It also concerns me that they say they've done this 'to produce the best sea ice thickness retrievals'. Evaluated against what? If SIT data at the buoys has been used to tune or train the method, it casts doubt on the whole buoy-based evaluation exercise. The veracity and role of this correction must be quantified prior to publication, and its impact on the validity of the evaluation must be assessed.

The "bias correction" is added to the satellite observed brightness temperature is due to the slight absorption of 6.9 GHz radiation by the polar atmosphere. In order to better account for this, we have chosen to use a radiation transfer model (RTTOV; Saunders et al., 2018) and pressure, temperature and humidity profiles along with skin temperature, surface pressure, 2 m temperature, 2 m humidity and 10 m winds from ECMWF ERA5 reanalysis monthly data (Hersbach et al., 2018) to model the effect of the atmosphere on the 6.9 GHz AMSR2 channels. For every month since 2003 and for the entirety of the Arctic basin, we have used the model to estimate the atmospheric transmission at 6.9 GHz and applied a location and time specific transmission factor to each AMSR2 radiance used in the calculation of snow—ice interface temperature. We will change equation 1 to reflect this by inserting a transmission $t$ to the right side of the equation and remove the statement that absorption at 6.9 GHz by the atmosphere is assumed negligible. The phrase "bias correction" will be removed from the manuscript as it doesn't accurately describe this methodology. Rather, we will add a description of the new methodology described here.

Whereas we previously had used a static 5 K correction, the resulting change to 6.9 GHz brightness temperatures affected by the modeled transmission term is very consistently near 5 K. Figure 2 shows mean and standard deviation atmospheric correction from atmospheric transmission to a 250 K brightness temperature during December, January and February (DJF) across the years 2003-2019. The Arctic basin shows a very spatially consistent roughly 4.5 K mean with standard deviations less than 0.1 K. These results are very similar to those reported by Burgard et al. (2020) who used a geophysical model to simulate 6.9 GHz brightness temperature at TOA using MPI-ESM output data. They report a difference of 4.49 K between the model ice surface temperature and the simulated 6.9 GHz brightness temperature at TOA for pixels with 99% or greater sea ice concentration during the summer season when accounting for columnar water vapor and columnar

cloud liquid water. Though we've reported our DJF results here, our summer results are very similar. These results will not be shown in the manuscript but are relevant to this review response.

[Figure]

*Figure 2: AMSR-E and AMSR2 6.9 GHz channels brightness temperature correction at 250 K in the 2003-2019 DJF (a) mean and (b) standard deviation calculated using a radiative transfer model and ERA5 reanalysis data. The correction is consistently near 4.5 K.*

**2.5 Ice-Ocean Boundary Conditions**

Seawater is not always in local thermal equilibrium with the sea ice interface (Schmidt et al., 2004; Mcphee, 2016). I'm not an expert on this, but it's relevant because this paper assumes equilibrium. For instance (as reported in McPhee), Maykut et al. (1971) found that without a steady basal flux of about 2Wm$^{-1}$, ice continued to grow unrealistically large in their model. Indeed Parkinson and Washington (1979) had to use a flux of an order of magnitude higher than this in their model. McPhee reports that observations from Sheba and Aidjex back these model fluxes up. This is clearly something the authors should address, perhaps with a sensitivity analysis to ocean-ice heat flux (which they say they've set to zero). If the 5 K offset discussed earlier was deployed to reduce modelled ice growth, perhaps the authors should consider that it is not the snow-ice temperatures being too low that are causing it, but an underestimation of ocean-ice heat flux?

First, we will remove the assumption of no flux from the liquid sea water to the solid sea ice and add a term for this flux ($F_w$) to equation 7:

$$\frac{\partial H}{\partial t} = \frac{\kappa_i}{\rho_i L H}(T_f - T_{si}) - \frac{F_w}{\rho_i L}$$

The addition of this term makes equation 7 difficult to solve analytically. However, the reduction in thickness growth due to flux from the liquid sea water is not dependent upon thickness and constant for a given flux value:

$$\frac{\partial H}{\partial t} = -\frac{F_w}{\rho_i L}$$

Multiplying this by a time interval will yield the total effect of the flux from the liquid sea water over that time interval. Equation 8 can be updated to include this effect:

$$H_t = \sqrt{H_{t-1}^2 + a^2 S} - \Delta t \frac{F_w}{\rho_i L}$$

We like the suggestion to apply a sensitivity analysis to the ocean—ice heat flux. With the new form of equation 8, it is straight forward to determine how much a change in basal flux will affect thickness growth. For a given snow—ice interface temperature, the reduction of sea ice thickness growth by inclusion of a basal flux is linearly related to the basal flux value by a factor of $1/\rho_i L$. Each 1 W m$^{-2}$ of basal flux decreases sea ice thickness growth by $2.84 \times 10^{-4}$ m d$^{-1}$. As such, the inclusion of 2 W m$^{-2}$ per Maykut and Untersteiner (1971) will reduce sea ice growth by $5.67 \times 10^{-4}$ m d$^{-1}$. We have updated SLICE to include a 2 W m$^{-2}$ flux from the liquid sea water.

2.6 Sea ice is a mushy layer

Sea ice is a mushy layer (Feltham et al., 2006) and this should be addressed when discussing heat flow through sea ice and accretion of new ice. Recently formed sea ice has brine inclusions, the phase equilibrium of which alters the bulk thermodynamic properties of the ice even well below the freezing temperature of seawater. Just stating what values you're using for the sea ice geophysical properties (L219) is insufficient. At minimum the values should be cited, and ideally they should be justified based on other previous modelling applications of the values. The constancy (as a function of temperature) of these values should also be considered. I'm not suggesting a multi-phase model of ice as I see that would make the whole situation very complicated and probably non-analytically soluble - the strength of SLICE is its simplicity. However, when presenting a model for ice growth based on heat flow through and phase change in ice near the freezing point, the mushy, mixed-phase characteristics of sea ice should be at least mentioned, and probably discussed.

We agree both that the strength of SLICE is indeed its simplicity and that further discussion of the choice of constants and their relationship to the multi-phase properties of sea ice is warranted.

We don't think that including the effects of temperature and phase change within internal brine pockets on the thermal conductivity of the sea ice is too complicated for SLICE and will include this effect. We begin by assigning an ocean salinity value of S = 33 ppt. We will assume that the ice is in thermal equilibrium relative to phase change within the ice between liquid brine and solid ice and calculate freezing point temperature (in °C) from S (in ppt) per Notz (2005) with the following polynomial fit to the liquidis curve for sea ice:

$$T_{f,NaCl} = -0.0592S - 9.37x10^{-6}S^2 - 5.33x10^{-7}S^3.$$

The latent heat of fusion of liquid to solid phase change the difference between the enthalpies of the two states if they are at the same temperature. In this case, we will use a latent heat of fusion as calculated by Notz (2005) using his empirical relationship for latent heat of fusion as a function of temperature (in °C):

$$L(T) = 333700 + 762.7T - 7.929T^2.$$

We then use equation 10 from Feltham (2006) to calculate effective conductivity at each time step:

$$k_{eff} = k_i - (k_i - k_b)(T_l(S_i) - T_l(S_{bulk}))/(T_l(S_i) - T),$$

with $k_i$ and $k_b$ from Batrak et al. (2018), Bailey (2010) and Schwertfeger (1963):

$$k_{bi} = k_i(2k_i + k_a - 2V_a(k_i - k_a))/(2k_i + k_a + 2V_a(k_i - k_a))$$

$$k_i = 1.162(1.905 - 8.66x10^{-3}T + 2.97x10^{-5}T^2)$$
$$k_b = 1.162(0.45 - 1.08x10^{-2}T + 5.04x10^{-5}T^2)$$
$$k_a = 0.03$$
$$V_a = 0.025.$$

The question becomes what temperature and ice salinity, $S_{bulk}$, would be used when calculating the effective conductivity since SLICE treats the sea ice layer as a single layer. Two options are to use snow—ice interface temperature and a surface salinity of 0 or to use a temperature mean over the depth of the ice and a salinity value from a parameterization such as Cox and Weeks (1974). We've tried both approaches and while the results are slightly different from one another, the difference is not significant. We will use effective conductivity calculated with surface conditions for SLICE which is similar to the approach adopted by Cox and Weeks (1988) who also used conductivity calculated from the surface to determine conductive flux through the ice layer.

Additionally, we will have further investigated our choices of density and latent heat of fusion. A first-year sea ice (FYI) density of 917 kg m$^{-3}$ and multi-year sea ice (MYI) density of 882 kg m$^{-3}$ was reported by Alexandrov et al. (2010) and these values have seen use in the sea ice thickness calculations from CryoSat-2 data (Laxon et al., 2013; Tilling et al., 2018; Hendricks and Ricker, 2016; Kwok and Cunningham, 2015). A sea ice density of 915 kg m$^{-3}$ is also in use with altimeter data (Kurtz et al., 2014; Petty et al., 2020) and a sea ice density of 925 kg m$^{-3}$ has been used with IceSat data (Kwok et al., 2009). Choice of sea ice density is a significant source of uncertainty in altimeter-based estimates of sea ice thickness. Kurtz et al. (2014) report that the range of densities from 882 kg m$^{-3}$ to 925 kg m$^{-3}$ yields a 1.1 m range in sea ice thickness estimates from a 60 cm snow—ice freeboard with 35 cm of snow. All ice formed by the SLICE model is FYI, leaving us with the choice of 915 kg m$^{-3}$, 917 kg m$^{-3}$ or 925 kg m$^{-3}$. We have moved forward with 917 kg m$^{-3}$. For a given snow—ice interface temperature, basal flux and sea ice thickness, a change to 915 kg m$^{-3}$ would increase sea ice thickness growth rate by only 0.2% and a change to 925 kg m$^{-3}$ would decrease sea ice thickness growth by only 0.8%.

2.7 Data and code availability

I was disappointed that the code and data used in this project were not made available to either the reviewers or the sea ice community. This is particularly the case given how much the authors have used other open data such as the CS2-SMOS and PIOMAS sea ice thickness data sets. To support this view, It's perhaps useful to refer to the data policy of this journal:

The output of research is not only journal articles but also data sets, model code, samples, etc. Only the entire network of interconnected information can guarantee integrity, transparency, reuse, and repro- ducibility of scientific findings. Moreover, all of these resources provide great additional value in their own right. Hence, it is particularly important that data and other information underpinning the research findings are "findable, accessible, interoperable, and reusable" (FAIR) not only for humans but also for machines.

I would recommend that upon resubmission they make their code available on a site such as GitHub, and produce a persistent identifier such as a DOI. I also suggest they place their data product in a persistent archive such as that run by Zenodo, for which they will receive a DOI and the opportunity to reversion the data upon article acceptance. In taking the above steps, I believe the authors will significantly increase the impact of their research.

While this step is not required for publication, we would like to increase the impact of this research in any way we can and will work to post both code and data on a publicly available repository. We will aim to provide more details along with the next revision of the article.

3 Other Comments

L2: 'Coupling'. I feel that 'coupled' systems/equations generally exchange information with and influence each other. However it seems that in this case you're feeding satellite information on the snow-ice interface temperature to an equation which tells you the growth rate. The satellite algorithm is not dependent on Eq. 7. So I think you should avoid portraying this as a coupled system; perhaps something like 'linking', or 'feeding' ?

We will use the word "linking".

L34: I think "is also effective" is subjective and should be changed. Perhaps "is also popular"?.

We will replace that sentence with the following:

"A global coupled ocean sea ice model with assimilated observational data is also commonly referenced."

L46: I think the word 'promising' is subjective and should be removed.

We will remove "promising".

L63: Should be polarization, not polarity I think?

You are correct, we will make this change.

L113: "Obvious dynamic effects" - what does this mean? I think you need to be clearer in this paper between dynamic thickening in a Lagrangian sense (i.e. convergence driven ridging and rafting of ice to make it thicker), and dynamical thickening in an Eulerian sense (advection of thicker ice into and thinner ice out of a grid cell).

We agree, we need to be clearer about advection vs. deformation. In this case, we mean "obvious deformed ice". We will also be sure to differentiate between the two effects throughout the paper.

L127: The snow loading is used before the hydrostatic conversion, in the calculation of the height of the ice surface above the waterline to account for the delay in radar propagation through the snow (e.g. Mallett et al., 2020).

We will update this description.

L129 CPOM is not affiliated with ESA

We will remove "ESA".

L142: Complementing, not complimenting

This will be updated.

L150: It's noticeable that the grid on which data are supplied and applied is consistently described up until the PIOMAS description. This is perhaps the most important data set for which to mention this, because the native grid is very unusual. Worth describing or not describing the grids consistently.

We will add a description of the PIOMAS grid, it is an important detail since the PIOMAS data is later interpolated to the EASE grid.

L360: Antarctic sea ice floes often have negative freeboards so you probably won't retrieve get the snow-ice interface temperature. Some floes have had them in the past leading to the formation of snow-ice, and ice lenses also exist in the snow, which I imagine will significantly complicate the retrieval of the snow-ice interface temperature. Indeed the potential for negative freeboards in the Arctic (Merkouriadi et al., 2020) should perhaps be mentioned at some point.

We will remove the allusion to Antarctic sea ice.

L374: It's my opinion that you'll only be 'retrieving' sea ice thickness when you do actually account for both thermodynamic and dynamic/advective contributions to sea ice thickness at a point. Right now I'd say you're modelling one part of it.

See comments after section 2.2.

3.1 Figures and Tables

The map projections used in this paper were unusual and not well-suited to the data being displayed. They look a bit like a Near-Sided Perspective projection? In any case, I think a more traditional North-Polar-Stereographic or Lambert Azimuthal-Equal-Area projection would be better. It looks in this case like data nearer the pole is being over-represented in area, and it's concerning that Hudson and Baffin Bay are hidden and highly distorted respectively.

We will update all applicable figures to the Stereographic projection.

I also think a figure should be displayed complementing Table 1 (perhaps put in a supplement?) with the tracks of the buoys used to evaluate SLICE. This would give the reader a better sense of the geo- graphic/spatial validity of the buoy-based evaluation presented.

We will add this, along with a plot of the OIB tracks.

Figure 1: The colorbar should be labelled with the variable (S-I Temp), and units (Kelvin) should be stated.

We will update this.

Figure 5: The blue/white plots aren't providing much narrative value here. They're similar in appear- ance and concept to Fig 4, and the panels often look very similar to each other; I would suggest putting them in a supplement and increasing the size of the difference plots, which are much more relevant and important.

Table 2: I think put this in a supplement and display the data as a timeseries. You could put the Vol. Growth in first two columns on the Y axis and the relative difference in % on a secondary Y axis. I'm not convinced the column with absolute differences adds much value. I think displaying this data as a graph

would give the reader a much better feel for what's going on.
Figure 6: Again, enlarge and focus on the difference plots and put the blue/white plots in a supplement. Table 3: Same comment as Table 2, and you could probably merge the resulting figures.

We agree that these suggestions will improve the narrative of the manuscript will update accordingly.

References

Feltham, D. L., Untersteiner, N., Wettlaufer, J. S., and Worster, M. G.: Sea ice is a mushy layer, Geophysical Research Letters, 33, 4–7, https://doi.org/10.1029/2006GL026290, 2006.

Mallett, R. D. C., Lawrence, I. R., Stroeve, J. C., Landy, J. C., and Tsamados, M.: Brief communication: Conventional assumptions involving the speed of radar waves in snow introduce systematic underesti- mates to sea ice thickness and seasonal growth rate estimates, Cryosphere, 14, 251–260, https://doi.org/ 10.5194/tc-14-251-2020, 2020.

Maykut, G. A., Untersteiner, N., MAYKUT GA, and UNTERSTEINER N: Some results from a time- dependent thermodynamic model of sea ice, J Geophys Res, 76, 1550–1575, https://doi.org/ 10.1029/jc076i006p01550, 1971.

Mcphee, M. G.: The sea ice–ocean boundary layer, Sea Ice: Third Edition, pp. 138–159, https://doi.org/ 10.1002/9781118778371.CH5, 2016.

Merkouriadi, I., Liston, G. E., Graham, R. M., and Granskog, M. A.: Quantifying the Potential for Snow- Ice Formation in the Arctic Ocean, Geophysical Research Letters, 47, e2019GL085 020, https://doi.org/ 10.1029/2019GL085020, URL https://onlinelibrary.wiley.com/doi/abs/10.1029/2019GL085020, 2020.

Parkinson, C. L. and Washington, W. M.: A large-scale numerical model of sea ice, Journal of Geophysical Research: Oceans, 84, 311–337, https://doi.org/10.1029/JC084IC01P00311, 1979.

Schmidt, G. A., Bitz, C. M., Mikolajewicz, U., and Tremblay, L. B.: Ice–ocean boundary conditions for coupled models, Ocean Modelling, 7, 59–74, https://doi.org/10.1016/S1463-5003(03)00030-1, 2004.

Alexandrov, V., Sandven, S., Wahlin, J., and Johannessen, O. M.: The relation between sea ice thickness and freeboard in the Arctic, Cryosphere, 4, 373–380, https://doi.org/10.5194/tc-4-373-2010, 2010.

Bailey, E., Feltham, D. L., and Sammonds, P. R.: A model for the consolidation of rafted sea ice, J. Geophys. Res.-Oceans, 115, c04015, https://doi.org/10.1029/2008JC005103, 2010.

Batrak, Y., Kourzeneva, E., and Homleid, M.: Implementation of a simple thermodynamic sea ice scheme, SICE version 1.0-38h1, within the ALADIN–HIRLAM numerical weather prediction system version 38h1, Geosci. Model Dev., 11, 3347–3368, https://doi.org/10.5194/gmd-11-3347-2018, 2018.

Burgard, C., Notz, D., Pedersen, L. T., and Tonboe, R. T.: The Arctic Ocean Observation Operator for 6.9 GHz (ARC3O) – Part 2: Development and evaluation, Cryosphere, 14, 2387–2407, https://doi.org/10.5194/tc-14-2387-2020, 2020.

Cox, G., and Weeks, W.: Salinity Variations in Sea Ice, J. Glaciol., 13(67), 109-120. https://doi.org/10.3189/S0022143000023418, 1974.

Cox, G., and Weeks, W.: Numerical simulations of the profile properties of undeformed first-year sea ice during the growth season. J. Geophys. Res. 93: 12,449-12,460., 1988.

Hersbach, H., Bell, B., Berrisford, P., Biavati, G., Horányi, A., Muñoz Sabater, J., Nicolas, J., Peubey, C., Radu, R., Rozum, I., Schepers, D., Simmons, A., Soci, C., Dee, D., Thépaut, J-N.: ERA5 hourly data on pressure levels from 1979 to present. Copernicus Climate Change Service (C3S) Climate Data Store (CDS). (Accessed on < 20-OCT-2020 >), 10.24381/cds.bd0915c6, 2018.

Kang, E.-J., Sohn, B.-J., Tonboe, R.T., Dybkjær, G., Holmlund, K., Kim, J.-M., and Liu, C.: Implementation of a 1-D thermodynamic model for simulating the winter-time evolvement of physical properties of snow and ice over the Arctic Ocean, Journal of Advances in Modeling Earth Systems, 13, e2020MS002448. https://doi.org/10.1029/2020MS002448, 2021

Kurtz, N. T., Galin, N., and Studinger, M.: An improved CryoSat-2 sea ice freeboard retrieval algorithm through the use of waveform fitting, Cryosphere, 8, 1217–1237, https://doi.org/10.5194/tc-8-1217-2014, 2014.

Kurtz, N., M. Studinger, J. Harbeck, V. Onana, and D. Yi.: IceBridge L4 Sea Ice Freeboard, Snow Depth, and Thickness, Version 1. [2013/3/20-2013/3/27; 2014/3/12-2014/3/31; 2015/3/19-2015/3/30; 2017/3/9-2017/3/24; 2018/3/22]. Boulder, Colorado USA. NASA National Snow and Ice Data Center Distributed Active Archive Center. doi: https://doi.org/10.5067/G519SHCKWQV6. [2/2022]., 2015

Kwok, R. and Rothrock, D. A.: Decline in Arctic sea ice thickness from submarine and ICESat records: 1958–2008, Geophys. Res. Lett., 36, L15501, https://doi.org/10.1029/2009GL039035, 2009.

Laxon, S. W., Giles, K. A., Ridout, A. L., Wingham, D. J., Willatt, R., Cullen, R., Kwok, R., Schweiger, A., Zhang, J., Haas, C., Hendricks, S., Krishfield, R., Kurtz, N., Farrell, S., and Davidson, M.: CryoSat-2 estimates of Arctic sea ice thickness and volume, Geophys. Res. Lett., 40, 732–737, https://doi.org/10.1002/grl.50193, 2013.

Mathew, N., Heygster, G., and Melsheimer, C.: Surface Emissivity of the Arctic Sea Ice at AMSR-E Frequencies, IEEE Transactions on Geoscience and Remote Sensing, 47, 4115-4124, https://doi.org/10.1109/TGRS.2009.2023667., 2009

Notz, D.: Thermodynamic and fluid-dynamical processes in sea ice. Cambridge: University of Cambridge, 2005.

Petty, A. A., Kurtz, N. T., Kwok, R., Markus, T., and Neumann, T. A.: Winter Arctic sea ice thickness from ICESat-2 freeboards, Journal of Geophysical Research: Oceans, 125, e2019JC015764. https://doi.org/10.1029/2019JC015764, 2020.

Saunders, R., Hocking, J., Turner, E., Rayer, P., Rundle, D., Brunel, P., Vidot, J., Roquet, P., Matricardi, M., Geer, A., Bormann, N., and Lupu, C.: An update on the RTTOV fast radiative transfer model (currently at version 12), Geosci. Model Dev., 11, 2717–2737, https://doi.org/10.5194/gmd-11-2717-2018, 2018.

Schwerdtfeger, P.: The thermal properties of sea ice, J. Glaciol., 4(36), 789– 807., 1963.

Tilling, R. L., Ridout, A., and Shepherd, A.: Estimating Arctic sea ice thickness and volume using CryoSat-2 radar altimeter data, Adv. Space Res., 62, 1203–1225, https://doi.org/10.1016/j.asr.2017.10.051, 2018.

Tschudi, M., Meier, W.N. , Stewart, J. S.,  Fowler, C., and Maslanik, J.: Polar Pathfinder Daily 25 km EASE-Grid Sea Ice Motion Vectors, Version 4. [Indicate subset used]. Boulder, Colorado USA. NASA National Snow and Ice Data Center Distributed Active Archive Center. https://doi.org/10.5067/INAWUWO7QH7B. [3/2021]., 2019

---

## Author Comment (AC2)

**A review of "A daily basin-wide sea ice thickness retrieval methodology: Stefan's Law Integrated Conducted Energy (SLICE)" by James Anheuser et al.**

The authors wish to thank the reviewer for the constructive review. We have responded in red font to individual comments below where necessary.

General Comments

The authors introduced a new method for sea ice thickness estimation from satellite snow- ice interface temperature (Tsi) by using idealized sea ice thermodynamic model. The key idea of their methodology is that thermodynamic sea ice growth rate can be calculated from upward conductive heat flux within the sea ice layer which balances with the latent heat of fusion. In their method, the conductive heat flux is a function of the Tsi under the linear temperature profile assumption. Therefore, sea ice thickness can be calculated from Tsi with appropriate initial ice thickness. Furthermore, the authors insist that the introduced method is self- correcting.

However, I have some major concerns about the introduced method. 1) There should be clarifications on the physical conditions (regions and seasons) that meet with the four assumptions they made. 2) More explanation is needed to insist that the method is "effectively" self-correcting. 3) The method seems to be a modeling approach rather than satellite retrieval. 4) Detailed procedure for the bias correction of the satellite Tsi must be provided.

Associated with the major concerns above, I think there should be significant improvements on the data and methodology before the manuscript is published to The Cryosphere. Therefore, my decision is to reconsider after the major revision. I would like to review the manuscript again after the revision.

Major comments
1. Assumptions in the SLICE method

From L202 to L208, the authors listed the four assumptions used in the SLICE retrieval method. I have concerns about the second and the third assumptions. In my opinion, the second assumption is equivalent to the statement that the temperature profile of sea ice is linear. But if you see the buoy measured temperature profiles, you will find this assumption is not always valid. Such linear profile assumption is generally valid during wintertime. Moreover, even during wintertime, sudden change in air temperature due to warm/cold advection or radiative forcing due to cloud cover can rapidly change surface temperature which makes curves in the temperature profile. The good thing is time-averaged temperature profile during wintertime is close to linear (Shi et al., 2020). The authors would consider shortening of retrieval period of the SLICE method.

The third assumption tells that there is no internal heat source associated with shortwave radiation. In other words, this assumption is valid for the regions where the solar zenith angle is maintained less than zero. The authors should check the validity of this assumption regarding the seasonal variability of the solar zenith angle. There can be sunlight in lower latitude regions during fall and spring. Otherwise, please consider the shortwave radiation effects or justify that the shortwave radiation effect is negligible for the lower latitude regions during the fall and spring seasons.

We agree that the assumptions of linear temperature profile and negligible shortwave radiation are valid during the winter time only and that SLICE is not valid outside of the sea ice growth season. As such, we will shorten the SLICE one-dimensional and basin-wide outputs to be from November 1 to March 31 only.

The other point is that the authors mentioned that the retrieval method should be applied in a Lagrangian sense in L224 but they neglected sea ice motion in the actual calculation (L262). What are the reasons for this? There must be justification for the neglect of sea ice motion. Each sea ice parcel should be tracked and matched with

the nearest satellite $T_{si}$ because the equation used in this study is a time-dependent equation. Meanwhile, the neglect of sea ice motion is not the same as focusing on thermodynamic growth. Thermodynamical growth, sea ice motion, and dynamical growth (deformation due to convergence and divergence) should be addressed separately. Consideration of sea ice motion without dynamical growth is possible.

The authors agree that sea ice motion and deformed sea ice due to convergence or divergence should be treated separately. While we are unable to include the effects of deformed ice, we will add a sea ice motion component to the basin-wide SLICE results.

The SLICE basin-wide component now includes advection of sea ice parcels using the NSIDC Polar Pathfinder daily sea ice drift product (Tschudi et al., 2019). We had planned to include this element in a future study but at the request of RC#2, we have chosen to include advection in this paper. SLICE is initialized with the CS2SMOS data or PIOMAS (interpolated to the 25 km EASE grid 2.0) from the first week of November and each 25 km x 25 km grid cell is divided into 5 km x 5km parcels, which are advected daily using the motion vectors interpolated to their position and who add sea ice thickness thermodynamically using the SLICE thermodynamic model. As before, new ice per a sea ice concentration product is initialized at 0.01 m. At any given time step, the parcels can be gridded back to the EASE grid 2.0 grid by taking the mean of parcels within each EASE grid. This process will be included in all basin-wide results shown in the revised manuscript.

In order to investigate whether SLICE can accurately capture deformed ice, we also attempted re-griding the parcels by taking the sum of parcel volume within grid cell and dividing by area. This process yielded unphysical results. Figure 1 shows an example of SLICE on March 31 2013 initialized with CryoSat-2/ SMOS on November 1 2012. The total volume of sea ice parcels within grid cell divided by grid cell area is as shown in Figure 1a. Those results are unphysical and are dominated by unrealistic convergence and divergence of parcels as shown by the total number of parcels per grid cell shown Figure 1b. The mean ice thickness of the parcels within each grid cell is shown in Figure 1c. and are the best results. The mean thickness within grid cell does not, however, capture deformed ice. Perhaps and improved sea ice motion product would allow the inclusion of deformed ice into SLICE.

[Figure]

*Figure 1: SLICE parcels on March 31 2013 (a) regridded using total parcel volume per grid cell divided by grid area, (b) counts within grid cell and (c) regridded mean parcel thickness within each grid cell. The volume per grid cell approach is unrealistic and dominated by erroneous convergence and divergence of parcels within grid cells.*

2. Effectiveness of self-correcting characteristic

It was interesting to read the statement in L225 regarding the self-correcting characteristic of the SLICE method. Thicker sea ice indeed grows slower than thinner sea ice with a given $T_{si}$ and vice versa according to equation (7). Therefore, the error in sea ice thickness can be relaxed by the modulation of sea ice growth speed.

However, the relaxation speed of error is important as well. If the speed of relaxation is slow, the effectiveness of self-correcting characteristics will be minor and the initial condition will be the major factor that determines the accuracy of sea ice thickness estimation. In L249-250 and Figure 2, the authors tried to show the effect of the self-correcting characteristic. Although it seems that 0.25 m deviations in the initial condition are decreasing with time, it will be better to specify the improvement quantitatively to know how fast the errors are relaxed. In addition, I suggest conducting a sensitivity test and including the result as an appendix.

Equation 7 shows that the conducted heat flux from basal sea ice growth is inversely related to sea ice thickness. All other factors being held equal, a change to sea ice thickness will be reflected by the inverse of that change to sea ice thickness growth rate. For example, a sea ice parcel that is twice as thick as a separate parcel will grow half as fast as that other parcel. We will add a +/- 0.5 m set of lines to Figure 3 and also include a quantitative assessment of how the 0.25 m and 0.5 m perturbations change of the course of the growth season.

I found some doubtful points on the self-correcting characteristic of the SLICE method. In my opinion, if the method is self-correcting, the retrieval result should fluctuate around the true state. Why is the SLICE retrieval (red solid line) the center of red shade instead of the buoy (blue solid line) which is the true state? In addition, I think the sentence "The bias grows with time as the SLICE profile moves away from its initialized thickness" makes a contradiction with the self-correcting characteristic of SLICE.

Theoretically, in the absence of any effects other than thermodynamic growth and if the SLICE assumptions are valid, initial condition errors will reduce over time. Because there are indeed other factos other than thermodynamic growth, this will not necessarily be reflected in the SLICE and buoy profiles. We will remove the sentence quoted by the reviewer regarding bias growth over time.

The significance of self-correcting characteristic is important for the algorithm extension to the past because such characteristic makes the retrieval method relatively independent from the accurate initial condition. If the self-correction is significant, SLICE sea ice thickness records initialized with PIOMAS can be constructed, and it will be more accurate than PIOMAS. To examine this, I suggest comparing the accuracy of the sea ice thickness from the PIOMAS and that from the SLICE initialized with the PIOMAS. There are some widely used independent datasets for validation such as Operation IceBridge (OIB), buoy, upward-looking sonar (ULS), and submarine observations.

The self-correcting characteristics of SLICE will not be significant enough to remove any dependence upon its initial condition. An accurate initial condition is important for SLICE's results. With regard to long term studies, we will remove allusions to the long-term application of SLICE and instead leave that for future investigation.

There are other advantages of SLICE over PIOMAS that are more significant than the theoretical error reduction discussed here. SLICE is thermodynamically forced by satellite observations of snow-ice interface temperature rather than an atmospheric reanalysis and is a much simpler model.

3. Retrieval or modeling (significance of this study)

In some sense, the SLICE retrieval method seems to be a thermodynamic sea ice model. The reason is that it simulates sea ice thickness evolution with time, and the result of SLICE retrieval is highly dependent on initial conditions rather than observed data. I think that the SLICE method is a simplified version of the

thermodynamic sea ice model introduced by Maykut and Unterstiener (1971) or the PIOMAS. It will be nice for the authors to explain why the SLICE method is satellite retrieval.

The most direct output from SLICE is a thermodynamic growth rate (and conducted flux through the ice). Much like many accepted retrievals, this output relies upon a priori information--sea ice thickness, freezing point temperature, etc. We believe this step of the process can be considered a retrieval based on a simple model. We concede that accumulating the sea ice growth into an absolute sea ice thickness is more of a modeling exercise, albeit one that is heavily observationally constrained. We will make this clear in the next revision.

Nonetheless, the novel point of this study is SLICE method is independent of the atmospheric reanalysis generally used as the forcing to sea ice model. The most relevant study to the SLICE method will be Kang et al. (2021), which simulates the physical state of a snow- ice system by using a thermodynamic equation set forced by atmospheric reanalysis and nudged by satellite $T_{si}$. This study has significance in terms of constructing an independent sea ice thickness record, while the physics of SLICE is very simplified compared to Kang et al. (2021) or other thermodynamic sea ice models. I recommend including an ice thickness comparison with the results of Kang et al. (2021). Their results are open to the public, and the authors can find the data repository in their paper. It is worth comparing the performance of the SLICE method with other sea ice models with more sophisticated physics and forced by reanalysis data.

We will add a comparison to the paper of SLICE initialized with CS2SMOS, PIOMAS and the model described by Kang et al. 2021 (hereafter K21) to Operation Ice Bridge (OIB) data (Kurtz, 2015).

OIB data from the month of March for the years 2013 through 2018 (including NSIDC OIB quick looks data) was first binned by SLICE grid cell and averaged across each bin to create collocated SLICE (initialized with CryoSat-2/SMOS) and OIB data. Both PIOMAS and the Kang et al. 2021 data were also interpolated to the SLICE grid. Using only SLICE grid cells with 100 or more individual OIB sea ice thickness data points within their bounds, a comparison between the datasets was created and shown below.

[Figure]

Figure 2: OIB thickness versus (a) SLICE intilialized with CryoSat-2/SMOS, (b) PIOMAS and (c) Kang et al., 2021 data including number of data points, linear correlations and bias with standar deviation. SLICE has the highest linear correlation though all three are nearly equal.

The highest linear correlation value belongs to SLICE at 0.704, however the linear correlation for PIOMAS and K21 are very near that value at 0.700 and 0.699 respectively. The smallest mean (standard deviation) bias is

exhibited by PIOMAS at -0.050 m (0.629 m) followed by SLICE with 0.171 m (0.628 m) and K21 with 0.307 m (0.647 m). This analysis will be included in the revised manuscript.

These statistics show that all three models have similar performance when modeling sea ice thickness, even without SLICE including a deformation component. The differences are related to complexity of the model and reliance upon model reanalysis data. Whereas both PIOMAS and K21 require snow information and must calculate the temperature profile in the snow in order to determine the temperature profile in the ice from a reanalysis product, SLICE uses direct retrieval of the snow—ice interface temperature in order to calculate the heat flux through the ice and therefore thermodynamic sea ice growth. By assuming a linear temperature profile in the sea ice, SLICE also removes the requirement for multiple ice layers to be tracked by the model.

We don't believe SLICE to be a replacement for existing sea ice thickness retrievals, rather an additional independent dataset created using an observationally constrained very simple model that may be more applicable in certain situations.

4. Bias correction for satellite $T_{si}$

The authors mentioned that "The resultant snow-ice interface temperatures were found to require a bias correction of 5 K in order to match buoy snow-ice interface temperatures...". I have read Lee and Sohn (2015) and remember that the snow-ice interface derived from AMSR- E 6.9 GHz brightness temperatures are validated with buoy measured temperature. The validation result showed that the bias was less than 1 K, which is a very different result from the 5 K bias in the manuscript. Lee and Sohn (2015) also neglected atmospheric/snow absorption.

Regarding this situation, first I thought that it is possibly due to the bias within AMSR-E and AMSR2 measurements. However, the authors stated that the AMSR2 data has been intercalibrated with the AMSR-E data so this may not be the issue. Then, may the version of L3 brightness temperature be a problem? Or simply authors failed to reproduce the $T_{si}$ retrieval algorithm.

It is unclear why the results from Lee and Sohn (2015) seem to have not required a correction for atmospheric absorption. The physics described in that paper are valid at the surface but the brightness temperatures viewed by the satellite at 6.9 GHz will be affected by the atmosphere, which we are accounting for.

This is a very critical issue because sea ice thickness is determined by $T_{si}$, which is the only real observation used for the sea ice thickness retrieval. The mentioned comparison result between buoy data and $T_{si}$ calculated by the authors showing 5 K bias must be presented (as an appendix) to justify the bias correction procedure. It will be worth reproducing figure 6 in Lee and Sohn (2015).

The "bias correction" is due to the slight absorption of 6.9 GHz radiation by the polar atmosphere. In order to better account for this, we have chosen to use a radiation transfer model (RTTOV; Saunders, et al., 2018) and pressure, temperature and humidity profiles along with skin temperature, surface pressure, 2 m temperature and humidity and 10 m winds from ECMWF ERA5 reanalysis data (Hersbach, et al., 2018) to model the effect of the atmosphere on the 6.9 GHz AMSR2 channels. For every day since 2003 and for the entirety of the Arctic basin, we have used the model to estimate the atmospheric transmission at 6.9 GHz and applied a location and time specific transmission factor to each AMSR2 radiance used in the calculation of snow—ice interface temperature. We will change equation 1 to reflect this by inserting a transmission $t$ to the right side of the equation and remove the statement that absorption at 6.9 GHz by the atmosphere is assumed negligible. The phrase "bias correction" will be removed from the manuscript as it doesn't accurately describe this methodology. Rather, we will add a description of the new methodology described here.

Whereas we previously had used a static 5 K correction, the resulting change to 6.9 GHz brightness temperatures affected by the modeled transmission term is very consistently near 5 K. The below figure shows mean and standard deviation atmospheric correction from atmospheric transmission to a 250 K brightness temperature during December, January and February (DJF) across the years 2003-2019. The Arctic basin shows a very spatially consistent roughly 4.5 K mean with standard deviations less than 0.1 K. These results are very similar to those reported by Burgard, et al., 2020 who used a geophysical model to simulate 6.9 GHz brightness temperature at TOA using MPI-ESM output data. They report a difference of 4.49 K between the model ice surface temperature and the simulated 6.9 GHz brightness temperature at TOA for pixels with 99% or greater sea ice concentration during the summer season when accounting for columnar water vapor and columnar cloud liquid water. Though we've reported our DJF results here, our summer results are very similar. These results will not be shown in the manuscript but are relevant to this review response.

[Figure]

*Figure 3: AMSR-E and AMSR2 6.9 GHz channels brightness temperature correction at 250 K in the 2003-2019 DJF (a) mean and (b) standard deviation calculated using a radiative transfer model and ERA5 reanalysis data. The correction is consistently near 4.5 K.*

Minor comments

L29-L37: Please provide more details for relevant studies on sea ice thickness retrieval in order to emphasize the novelty or necessity of SLICE. How are the satellite altimetry methods limited in spatial coverage and temporal resolution (I think the resolution of ICESat-2 is better than passive microwave sensors such as AMSR2 6.9 GHz)? What are the limitations of the other methods? How is this study related to the existing studies?

We will add more quantitative information to this passage. In any case, while the ICESat-2 and CryoSat-2 spatial resolutions may be better than microwave instruments (which we did not and do not dispute), the orbit details, spatial coverage and temporal resolutions combine in such a way that both satellite sensors take much longer to cover the entire Arctic than AMSR2 and AMSR-E.

L63: horizontally and vertically polarized…

We will make this change.

L215: Please define negative degree-days in the manuscript and provide what happens if the temperature is positive (melting?).

The negative degree-days term is defined in L216. SLICE is not capable of capturing melt.

L221: It is hard to know which equation was used for sea ice thickness calculation. Equation (4) is too general. Did you use equation (8) which is an analytic solution for sea ice thickness, or equation (7) for change in sea ice thickness per unit time and accumulate the thickness changes?

The equations will be changed slightly in order to account for heat flux from the liquid ocean to the solid sea ice per a recommendation from RC#1. We will be sure to be more clear about which equations are used in the algorithm.

L235-237: Why the retrieval method was initialized with such condition (the day when the 14 d rolling average sea ice growth exceeded 1mm $d^{-1}$)? Please provide the reason.

We have updated the one-dimensional results to begin with the buoy initial condition on November 1 rather than the previous definition of a start time based on ice growth exceeding a threshold. This also reflects how the basin-wide methodology works.

L400: I think uploading the data produced in this study to the public data repository more fits the data policy of TC journal.

While this step is not required for publication, we would like to increase the impact of this research in any way we can and will work to post both code and data on a publicly available repository. We will aim to provide more details along with the next revision of the article.

References

Kang, E. J., Sohn, B. J., Tonboe, R. T., Dybkjaer, G., Holmlund, K., Kim, J. M., and Liu, C.: Implementation of a 1-D Thermodynamic Model for Simulating the Winter-Time Evolvement of Physical Properties of Snow and Ice Over the Arctic Ocean, J. Adv. Model. Earth Sy., 13, 3, https://doi.org/10.1029/2020ms002448, 2021.

Lee, S.-M and Sohn, B.-J.: Retrieving the refractive index, emissivity, and surface temperature of polar sea ice from 6.9 GHz microwave measurements: A theoretical development, J. Geophys. Res. Atmos., 120(6), 2293-2305, https://doi.org/10.1002/2014JD022481, 2015.

Maykut, G. A. and Untersteiner, N.: Some results from a time-dependent thermodynamic model of sea ice, J. Geophys. Res., 76, 1550–1575, https://doi.org/10.1029/JC076i006p01550, 1971.

Shi, H., Sohn, B.-J., Dybkjær, G., Tonboe, R. T., and Lee, S.-M.: Simultaneous estimation of wintertime sea ice thickness and snow depth from space-borne freeboard measurements, The Cryosphere, 14, 3761–3783, https://doi.org/10.5194/tc-14-3761-2020, 2020.

Burgard, C., Notz, D., Pedersen, L. T., and Tonboe, R. T.: The Arctic Ocean Observation Operator for 6.9 GHz (ARC3O) – Part 2: Development and evaluation, Cryosphere, 14, 2387–2407, https://doi.org/10.5194/tc-14-2387-2020, 2020.

Hersbach, H., Bell, B., Berrisford, P., Biavati, G., Horányi, A., Muñoz Sabater, J., Nicolas, J., Peubey, C., Radu, R., Rozum, I., Schepers, D., Simmons, A., Soci, C., Dee, D., Thépaut, J-N.: ERA5 hourly data on pressure levels from 1979 to present. Copernicus Climate Change Service (C3S) Climate Data Store (CDS). (Accessed on < 20-OCT-2020 >), 10.24381/cds.bd0915c6, 2018.

Kurtz, N., M. Studinger, J. Harbeck, V. Onana, and D. Yi.: IceBridge L4 Sea Ice Freeboard, Snow Depth, and Thickness, Version 1. [2013/3/20-2013/3/27; 2014/3/12-2014/3/31; 2015/3/19-2015/3/30; 2017/3/9-2017/3/24; 2018/3/22]. Boulder, Colorado USA. NASA National Snow and Ice Data Center Distributed Active Archive Center. doi: https://doi.org/10.5067/G519SHCKWQV6. [2/2022]., 2015

Saunders, R., Hocking, J., Turner, E., Rayer, P., Rundle, D., Brunel, P., Vidot, J., Roquet, P., Matricardi, M., Geer, A., Bormann, N., and Lupu, C.: An update on the RTTOV fast radiative transfer model (currently at version 12), Geosci. Model Dev., 11, 2717–2737, https://doi.org/10.5194/gmd-11-2717-2018, 2018.

Tschudi, M., Meier, W.N. , Stewart, J. S.,  Fowler, C., and Maslanik, J.: Polar Pathfinder Daily 25 km EASE-Grid Sea Ice Motion Vectors, Version 4. [Indicate subset used]. Boulder, Colorado USA. NASA National Snow and Ice Data Center Distributed Active Archive Center. https://doi.org/10.5067/INAWUWO7QH7B. [3/2021]., 2019

---

## Author Response (AR1)

**Author Response**

Anheuser et al., 2021

tc-2021-333

RC 1

1.    Assessing the performance of a new sea ice thickness product

Two key benchmarks for a new sea ice thickness product are (a) the degree to which the product outperforms a climatology, and (b) whether and in what ways the product outperforms comparable products - in this case probably PIOMAS.

To address point (a), I think the authors could do some fairly straightforward things. The first is to take the first two columns of Table 2 and correlate the anomalies. This would reveal whether SLICE is thicker when the sea ice is observed to be thicker. To put this another way - to what degree does SLICE capture the sign and size of volume anomalies from the mean state/ climatology. You could even break it down regionally into where it does & doesn't show skill.

We have completed this analysis and found that the p-value of the correlation between the SLICE volume growth and the CryoSat-2/SMOS volume growth is too high to report the results as significant. We have added Figure 8 which shows volume growth of SLICE compared to PIOMAS and AWI CS2SMOS.

To address (b) I think the users need to find a unique selling point over PIOMAS. To have impact, this product/method will need to be preferable in at least one way, either in availability or skill. The way I see it, PIOMAS has clear availability advantages: it is open-access so anybody can use it, and it stretches back to 1979. As for skill, it has the advantage of including both the dynamical and thermodynamical components of thickening. On the other hand, SLICE is based on fairly direct observations of the snow-ice interface temperature, whereas PIOMAS has to work this out by first modelling the snow and then calculating the temperature gradient across the snow and ice. To show that this translates into an actual skill advantage (and so to get people to use the method/product), I think the authors need to show that SLICE outperforms PIOMAS in some way, place or time. This is particularly the case because SLICE needs to be intialised by a sea ice thickness data set anyway, which may well be PIOMAS. The best way to assess skill would be by benchmarking against (i.e. assuming as truth) some satellite-altimeter derived ice thickness product, or by combining the work they've done with other in-situ products like Operation Ice Bridge or ULS buoys. Without doing this, I don't think the SLICE-derived ice thickness will be greatly used by the community over PIOMAS.

We have added a comparison of SLICE initialized with CS2SMOS, PIOMAS and the model described by Kang et al. 2021 (hereafter K21) to Operation Ice Bridge (OIB) data (Kurtz, 2015). This comparison has been added to Sect. 4.2 (Ln 377-384). Additionally, we've added a comparison between SLICE and PIOMAS by comparing both to AWI CS2SMOS (Ln 385-403).

These comparisons show that all three models have similar performance when modeling sea ice thickness, even without SLICE including a deformation component and including a number of assumptions that allow SLICE to retrieve instantaneous thermodynamic thickness growth rate. This has become a prominent point of the paper (Ln 9-11, 438-447, 477-478).

2.    Retrieval vs modelling

The term 'retrieval' touches on an emerging issue in the sea ice community concerning what properties we model, and what properties we observe/retrieve. For variables such as sea ice height, there is clearly a spectrum from direct observations (e.g. spot heights from a satellite-mounted laser altimeter) to highly-modelled (e.g. sea ice height output from a CMIP6-class model). Other quantities (e.g. radar-derived sea ice thickness from CryoSat-2, as used in this paper) are synthesised from observations (the timing and waveforms of scattered radar energy) and simple models (hydrostatic equilibrium, radar pulse

propagation through snow, etc.). The case in this paper is similarly subjective: on one hand the authors are using observations of brightness temperatures, and what I would say is an observation of the snow-ice temperature. But then they're using a highly-idealised model of latent-heat release and heat flow known as Stefan's Law (which is in many ways is not a law but a series of combined thermodynamic assumptions, which are arguably outdated - see below).

Although I am certain this is not the intention of the authors, I fear that describing these sea ice thickness data as 'retrievals' implies a degree of direct observation that is too strong. I think that this implication may, at worst, lead to users (i.e. those wishing to initialise models) thinking that these data are more certain than they are, and more directly observed than they are. We regularly see this phenomenon with PIOMAS data for instance, which is very much a model but is treated by some as if it were observed because it is fed by reanalysis products. I therefore suggest that the authors be more explicit that they are modelling ice growth, and accumulating the results of that modelling exercise to model total ice thickness. On this basis I also urge them to remove the term 'retrieval' from their title.

The authors broadly agree with the points made here. Certainly, all retrievals rely on a model at some level. For instance, even the spot heights from a laser altimeter referred to by the reviewer are based on a very simple model converting signal response times to spot heights. We believe SLICE sits in between these types of methods that are accepted as retrievals and results from models such as PIOMAS and that this is indeed the strength of SLICE.

As stated by the reviewer, the most direct output from SLICE is a thermodynamic growth rate (and conducted flux through the ice). Much like many accepted retrievals, this output relies upon a priori information--sea ice thickness, freezing point temperature, etc. We believe this step of the process can be considered a retrieval based on a simple model. We concede that accumulating the sea ice growth into an absolute sea ice thickness is more of a modeling exercise, albeit one that is heavily observationally constrained. We will make this clear in the next revision.

We have re-framed SLICE as a thermodynamic sea ice growth retrieval rather than a sea ice thickness product and propose the following title:

*A simple model for daily basin-wide thermodynamic sea ice thickness growth retrieval*

3.    Long Term Applications

The authors state that this method could be deployed several decades into the past. For instance, they do this in both their abstract and penultimate sentence. Their justification for this is that the snow-ice temperature is retrievable back to 1987, but I think that reconstruction back to this date is not usefully possible because initialisation is not available. It seems to me that the only way of doing this would be to initialise the product with an already existing and probably more accurate pan-Arctic sea ice thickness product. If this already exists, what would be the benefit of having this product, that would be dependent on (i.e. initialised by) the superior product? As a side point, I also fear that the authors' 5 K bias correction may not be relevant pre-2000, given than the roughness and snow depth of sea ice has declined, among other geophysical changes.

We will remove allusions to the long term application of SLICE and instead leave that for future investigation.

4.    5 K bias Correction

On L82 the authors mention that they have performed a 5K bias correction on the Snow-Ice temperature data to make it match the buoy data. It's possible that they didn't do this themselves, but took it from a paper - but if so they should cite it. They certainly need to say whether they've added or subtracted the value. But this seems to be a pretty critical point that is not explored nearly enough. How did they get to this number? How sensitive is it to the data from individual buoys? How much did it improve the match between S-I temperature and the buoys? It also concerns me that they say they've done this 'to produce the best sea ice thickness retrievals'. Evaluated against what? If SIT data at the buoys has been used to tune or train the method, it casts doubt on the whole buoy-based evaluation exercise. The veracity and

role of this correction must be quantified prior to publication, and its impact on the validity of the evaluation must be assessed.

The "bias correction" is added to the satellite observed brightness temperature and is due to the slight absorption of 6.9 GHz radiation by the polar atmosphere. In order to better account for this, we have chosen to use a radiative transfer model. Details can be found in new section Sect. 3.1.1.

Whereas we previously had used a static 5 K correction, the resulting change to 6.9 GHz brightness temperatures affected by the modeled transmission term is very consistently near 5 K. Figure 2 shows mean and standard deviation atmospheric correction from atmospheric transmission to a 250 K brightness temperature during December, January and February (DJF) across the years 2003-2019. The Arctic basin shows a very spatially consistent roughly 4.5 K mean with standard deviations less than 0.1 K. These results are very similar to those reported by Burgard et al. (2020) who used a geophysical model to simulate 6.9 GHz brightness temperature at TOA using MPI-ESM output data. They report a difference of 4.49 K between the model ice surface temperature and the simulated 6.9 GHz brightness temperature at TOA for pixels with 99% or greater sea ice concentration during the summer season when accounting for columnar water vapor and columnar cloud liquid water. Though we've reported our DJF results here, our summer results are very similar. These results will not be shown in the manuscript but are relevant to this review response.

[Figure]

*Figure 1: AMSR-E and AMSR2 6.9 GHz channels brightness temperature correction at 250 K in the 2003-2019 DJF (a) mean and (b) standard deviation calculated using a radiative transfer model and ERA5 reanalysis data.The correction is consistently near 4.5 K.*

5. Ice-Ocean Boundary Conditions

Seawater is not always in local thermal equilibrium with the sea ice interface (Schmidt et al., 2004; Mcphee, 2016). I'm not an expert on this, but it's relevant because this paper assumes equilibrium. For instance (as reported in McPhee), Maykut et al. (1971) found that without a steady basal flux of about 2Wm$^{-1}$, ice continued to grow unrealistically large in their model. Indeed Parkinson and Washington (1979) had to use a flux of an order of magnitude higher than this in their model. McPhee reports that observations from Sheba and Aidjex back these model fluxes up. This is clearly something the authors should address, perhaps with a sensitivity analysis to ocean-ice heat flux (which they say they've set to zero). If the 5 K offset discussed earlier was deployed to reduce modelled ice growth, perhaps the

authors should consider that it is not the snow-ice temperatures being too low that are causing it, but an underestimation of ocean-ice heat flux?

We have removed the assumption of no flux from the liquid sea water to the solid sea ice and added a term for this flux ($F_w$) to the SLICE model (Eqs. (8), (9), and (10)). A detailed investigation of basal flux and its implications has been added with new section Sect. 3.2.2.

6.  Sea ice is a mushy layer

Sea ice is a mushy layer (Feltham et al., 2006) and this should be addressed when discussing heat flow through sea ice and accretion of new ice. Recently formed sea ice has brine inclusions, the phase equilibrium of which alters the bulk thermodynamic properties of the ice even well below the freezing temperature of seawater. Just stating what values you're using for the sea ice geophysical properties (L219) is insufficient. At minimum the values should be cited, and ideally they should be justified based on other previous modelling applications of the values. The constancy (as a function of temperature) of these values should also be considered. I'm not suggesting a multi-phase model of ice as I see that would make the whole situation very complicated and probably non-analytically soluble - the strength of SLICE is its simplicity. However, when presenting a model for ice growth based on heat flow through and phase change in ice near the freezing point, the mushy, mixed-phase characteristics of sea ice should be at least mentioned, and probably discussed.

We agree both that the strength of SLICE is indeed its simplicity and that further discussion of the choice of constants and their relationship to the multi-phase properties of sea ice is warranted. We have added a parameterization of the multi-phase effects on thermal conductivity per Feltham, et l. (2006). New section Sect. 3.2.3. describes this parametrization in detail and includes a discussion of sea ice density.

7.  Data and code availability

I was disappointed that the code and data used in this project were not made available to either the reviewers or the sea ice community. This is particularly the case given how much the authors have used other open data such as the CS2-SMOS and PIOMAS sea ice thickness data sets. To support this view, It's perhaps useful to refer to the data policy of this journal:

The output of research is not only journal articles but also data sets, model code, samples, etc. Only the entire network of interconnected information can guarantee integrity, transparency, reuse, and repro-ducibility of scientific findings. Moreover, all of these resources provide great additional value in their own right. Hence, it is particularly important that data and other information underpinning the research findings are "findable, accessible, interoperable, and reusable" (FAIR) not only for humans but also for machines.

I would recommend that upon resubmission they make their code available on a site such as GitHub, and produce a persistent identifier such as a DOI. I also suggest they place their data product in a persistent archive such as that run by Zenodo, for which they will receive a DOI and the opportunity to reversion the data upon article acceptance. In taking the above steps, I believe the authors will significantly increase the impact of their research.

Thank you for this suggestion. Data used in creation of all figures is now available at https://doi.org/10.5281/zenodo.6554832. Code for creation of data and figures is available at https://doi.org/10.5281/zenodo.6561431 and https://github.com/janheuser/SLICE/releases/tag/1.0.0.

8.  Other Comments

L2: 'Coupling'. I feel that 'coupled' systems/equations generally exchange information with and influence each other. However it seems that in this case you're feeding satellite information on the snow-ice interface temperature to an equation which tells you the growth rate. The satellite algorithm is not dependent on Eq. 7. So I think you should avoid portraying this as a coupled system; perhaps something like 'linking', or 'feeding' ?

We changed "coupled" to "linking".

L34: I think "is also effective" is subjective and should be changed. Perhaps "is also popular"?.

We have replaced that sentence with the following:

*A coupled ocean--sea ice model with assimilated observational data is also commonly referenced.* (Ln 30)

L46: I think the word 'promising' is subjective and should be removed.

We have removed "promising".

L63: Should be polarization, not polarity I think?

You are correct, we have made this change. (Ln 68)

L113: "Obvious dynamic effects" - what does this mean? I think you need to be clearer in this paper between dynamic thickening in a Lagrangian sense (i.e. convergence driven ridging and rafting of ice to make it thicker), and dynamical thickening in an Eulerian sense (advection of thicker ice into and thinner ice out of a grid cell).

We agree, we need to be clearer about advection vs. deformation. In this case, we mean "obvious deformed ice". We have added a distinction between deformation and advection throughout the paper (e.g., 173; 336).

L127: The snow loading is used before the hydrostatic conversion, in the calculation of the height of the ice surface above the waterline to account for the delay in radar propagation through the snow (e.g. Mallett et al., 2020).

We have updated this description. (Ln 91-92)

L129 CPOM is not affiliated with ESA

We have removed "ESA". (Ln 94)

L142: Complementing, not complimenting

This has been updated. (Ln 107)

L150: It's noticeable that the grid on which data are supplied and applied is consistently described up until the PIOMAS description. This is perhaps the most important data set for which to mention this, because the native grid is very unusual. Worth describing or not describing the grids consistently.

We have added a description of the PIOMAS grid. (Ln 150-154)

L360: Antarctic sea ice floes often have negative freeboards so you probably won't retrieve get the snow-ice interface temperature. Some floes have had them in the past leading to the formation of snow-ice, and ice lenses also exist in the snow, which I imagine will significantly complicate the retrieval of the snow-ice interface temperature. Indeed the potential for negative freeboards in the Arctic (Merkouriadi et al., 2020) should perhaps be mentioned at some point.

We have removed the allusion to Antarctic sea ice.

L374: It's my opinion that you'll only be 'retrieving' sea ice thickness when you do actually account for both thermodynamic and dynamic/advective contributions to sea ice thickness at a point. Right now I'd say you're modelling one part of it.

See comments above.

9.  Figures and Tables

The map projections used in this paper were unusual and not well-suited to the data being displayed. They look a bit like a Near-Sided Perspective projection? In any case, I think a more traditional North-Polar- Stereographic or Lambert Azimuthal-Equal-Area projection would be better. It looks in this case like data nearer the pole is being over-represented in area, and it's concerning that Hudson and Baffin Bay are hidden and highly distorted respectively.

We have updated all applicable figures to the Stereographic projection.

I also think a figure should be displayed complementing Table 1 (perhaps put in a supplement?) with the tracks of the buoys used to evaluate SLICE. This would give the reader a better sense of the geographic/spatial validity of the buoy-based evaluation presented.

We have added this, along with a plot of the OIB tracks.

Figure 1: The colorbar should be labelled with the variable (S-I Temp), and units (Kelvin) should be stated.

We have updated this.

Figure 5: The blue/white plots aren't providing much narrative value here. They're similar in appear- ance and concept to Fig 4, and the panels often look very similar to each other; I would suggest putting them in a supplement and increasing the size of the difference plots, which are much more relevant and important.

Table 2: I think put this in a supplement and display the data as a timeseries. You could put the Vol. Growth in first two columns on the Y axis and the relative difference in % on a secondary Y axis. I'm not convinced the column with absolute differences adds much value. I think displaying this data as a graph would give the reader a much better feel for what's going on.

Figure 6: Again, enlarge and focus on the difference plots and put the blue/white plots in a supplement. Table 3: Same comment as Table 2, and you could probably merge the resulting figures.

We have removed SLICE initialized with PIOMAS from the manuscript and instead focused on comparing SLICE (initialized with AWI CS2SMOS) to PIOMAS, per comments from both referees. As such, we have removed previous Fig. 6 and previous Table 3. We have separated out end of season results from end of season difference plots as well. In an updated version of previous Fig. 5, new Fig. 6 shows end of season SLICE minus AWI CS2SMOS and end of season PIOMAS minus AWI CS2SMOS in order to compare SLICE and PIOMAS using AWI CS2SMOS as the reference dataset. End of season results for SLICE, PIOMAS and AWI CS2SMOS are shown in new Fig. A1. Additionally, previous Table 3 has been replaced by new Fig. 8 showing volume growth for SLICE, PIOMAS and CS2SMOS in graphic form rather than tabular form.

References

Feltham, D. L., Untersteiner, N., Wettlaufer, J. S., and Worster, M. G.: Sea ice is a mushy layer, Geophysical Research Letters, 33, 4–7, https://doi.org/10.1029/2006GL026290, 2006.

Mallett, R. D. C., Lawrence, I. R., Stroeve, J. C., Landy, J. C., and Tsamados, M.: Brief communication: Conventional assumptions involving the speed of radar waves in snow introduce systematic underesti- mates to sea ice thickness and seasonal growth rate estimates, Cryosphere, 14, 251–260, https://doi.org/ 10.5194/tc-14-251-2020, 2020.

Maykut, G. A., Untersteiner, N., MAYKUT GA, and UNTERSTEINER N: Some results from a time-dependent thermodynamic model of sea ice, J Geophys Res, 76, 1550–1575, https://doi.org/ 10.1029/jc076i006p01550, 1971.

Mcphee, M. G.: The sea ice–ocean boundary layer, Sea Ice: Third Edition, pp. 138–159, https://doi.org/ 10.1002/9781118778371.CH5, 2016.

Merkouriadi, I., Liston, G. E., Graham, R. M., and Granskog, M. A.: Quantifying the Potential for Snow-Ice Formation in the Arctic Ocean, Geophysical Research Letters, 47, e2019GL085 020, https://doi.org/ 10.1029/2019GL085020, URL https://onlinelibrary.wiley.com/doi/abs/10.1029/2019GL085020, 2020.

Parkinson, C. L. and Washington, W. M.: A large-scale numerical model of sea ice, Journal of Geophysical Research: Oceans, 84, 311–337, https://doi.org/10.1029/JC084IC01P00311, 1979.

Schmidt, G. A., Bitz, C. M., Mikolajewicz, U., and Tremblay, L. B.: Ice–ocean boundary conditions for coupled models, Ocean Modelling, 7, 59–74, https://doi.org/10.1016/S1463-5003(03)00030-1, 2004.

Burgard, C., Notz, D., Pedersen, L. T., and Tonboe, R. T.: The Arctic Ocean Observation Operator for 6.9 GHz (ARC3O) – Part 2: Development and evaluation, Cryosphere, 14, 2387–2407, https://doi.org/10.5194/tc-14-2387-2020, 2020.

Kang, E.-J., Sohn, B.-J., Tonboe, R.T., Dybkjær, G., Holmlund, K., Kim, J.-M., and Liu, C.: Implementation of a 1-D thermodynamic model for simulating the winter-time evolevent of physical properties of snow and ice over the Arctic Ocean, Journal of Advances in Modeling Earth Systems, 13, e2020MS002448. https://doi. org/10.1029/2020MS002448, 2021

Kurtz, N., M. Studinger, J. Harbeck, V. Onana, and D. Yi.: IceBridge L4 Sea Ice Freeboard, Snow Depth, and Thickness, Version 1. [2013/3/20-2013/3/27; 2014/3/12-2014/3/31; 2015/3/19-2015/3/30; 2017/3/9-2017/3/24; 2018/3/22]. Boulder, Colorado USA. NASA National Snow and Ice Data Center Distributed Active Archive Center. doi: https://doi.org/10.5067/G519SHCKWQV6. [2/2022]., 2015

RC 2

Major comments

1.  Assumptions in the SLICE method

From L202 to L208, the authors listed the four assumptions used in the SLICE retrieval method. I have concerns about the second and the third assumptions. In my opinion, the second assumption is equivalent to the statement that the temperature profile of sea ice is linear. But if you see the buoy measured temperature profiles, you will find this assumption is not always valid. Such linear profile assumption is generally valid during wintertime. Moreover, even during wintertime, sudden change in air temperature due to warm/cold advection or radiative forcing due to cloud cover can rapidly change surface temperature which makes curves in the temperature profile. The good thing is time-averaged temperature profile during wintertime is close to linear (Shi et al., 2020). The authors would consider shortening of retrieval period of the SLICE method.

The third assumption tells that there is no internal heat source associated with shortwave radiation. In other words, this assumption is valid for the regions where the solar zenith angle is maintained less than zero. The authors should check the validity of this assumption regarding the seasonal variability of the solar zenith angle. There can be sunlight in lower latitude regions during fall and spring. Otherwise, please consider the shortwave radiation effects or justify that the shortwave radiation effect is negligible for the lower latitude regions during the fall and spring seasons.

We agree that the assumptions of linear temperature profile and negligible shortwave radiation are valid during the winter time only and that SLICE is not valid outside of the sea ice growth season. As such, we have shortened the SLICE one-dimensional and basin-wide outputs to be from November 1 to March 31 only.

The other point is that the authors mentioned that the retrieval method should be applied in a Lagrangian sense in L224 but they neglected sea ice motion in the actual calculation (L262). What are the reasons for this? There must be justification for the neglect of sea ice motion. Each sea ice parcel should be tracked and matched with the nearest satellite Tsi because the equation used in this study is a time-dependent equation. Meanwhile, the neglect of sea ice motion is not the same as focusing on thermodynamic growth. Thermodynamical growth, sea ice motion, and dynamical growth (deformation due to convergence and divergence) should be addressed separately. Consideration of sea ice motion without dynamical growth is possible.

The authors agree that sea ice motion and deformed sea ice due to convergence or divergence should be treated separately. While we are unable to include the effects of deformed ice, we have added a sea ice motion component to the basin-wide SLICE results. New section Sect. 3.3 describes this component.

2.  Effectiveness of self-correcting characteristic

It was interesting to read the statement in L225 regarding the self-correcting characteristic of the SLICE method. Thicker sea ice indeed grows slower than thinner sea ice with a given Tsi and vice versa

according to equation (7). Therefore, the error in sea ice thickness can be relaxed by the modulation of sea ice growth speed.

However, the relaxation speed of error is important as well. If the speed of relaxation is slow, the effectiveness of self-correcting characteristics will be minor and the initial condition will be the major factor that determines the accuracy of sea ice thickness estimation. In L249-250 and Figure 2, the authors tried to show the effect of the self-correcting characteristic. Although it seems that 0.25 m deviations in the initial condition are decreasing with time, it will be better to specify the improvement quantitatively to know how fast the errors are relaxed. In addition, I suggest conducting a sensitivity test and including the result as an appendix.

I found some doubtful points on the self-correcting characteristic of the SLICE method. In my opinion, if the method is self-correcting, the retrieval result should fluctuate around the true state. Why is the SLICE retrieval (red solid line) the center of red shade instead of the buoy (blue solid line) which is the true state? In addition, I think the sentence "The bias grows with time as the SLICE profile moves away from its initialized thickness" makes a contradiction with the self-correcting characteristic of SLICE.

The significance of self-correcting characteristic is important for the algorithm extension to the past because such characteristic makes the retrieval method relatively independent from the accurate initial condition. If the self-correction is significant, SLICE sea ice thickness records initialized with PIOMAS can be constructed, and it will be more accurate than PIOMAS. To examine this, I suggest comparing the accuracy of the sea ice thickness from the PIOMAS and that from the SLICE initialized with the PIOMAS. There are some widely used independent datasets for validation such as Operation IceBridge (OIB), buoy, upward-looking sonar (ULS), and submarine observations.

The self-correcting characteristics of SLICE will not be significant enough to remove any dependence upon its initial condition. An accurate initial condition is important for SLICE's results. With regard to long term studies, we will remove allusions to the long-term application of SLICE and instead leave that for future investigation. There are other advantages of SLICE over PIOMAS that are more significant than the theoretical error reduction discussed here. SLICE is thermodynamically forced by satellite observations of snow-ice interface temperature rather than an atmospheric reanalysis and is a much simpler model. Additionally, it is likely that PIOMAS exhibits similar behavior in that thinner ice grows faster than thicker. We have removed any discussion of self-correction other than:

In Eq. 9, thicker sea ice grows slower than thinner sea ice and thinner sea ice grows faster than thicker ice with a given snow--ice interface temperature. This means that in the presence of only thermodynamic effects, sea ice that is too thick or too thin will correct towards the unbiased SLICE thickness profile. (Ln 268-271)

and

Additionally, SLICE has a self-correcting quality by nature of Eq. 9 whereby sea ice thicknesses that are biased in either direction approach the unbiased SLICE sea ice thickness over time. (Ln 419-420)

3. Retrieval or modeling (significance of this study)

In some sense, the SLICE retrieval method seems to be a thermodynamic sea ice model. The reason is that it simulates sea ice thickness evolution with time, and the result of SLICE retrieval is highly dependent on initial conditions rather than observed data. I think that the SLICE method is a simplified version of the thermodynamic sea ice model introduced by Maykut and Unterstiener (1971) or the PIOMAS. It will be nice for the authors to explain why the SLICE method is satellite retrieval.

The most direct output from SLICE is a thermodynamic growth rate (and conducted flux through the ice). Much like many accepted retrievals, this output relies upon a priori information--sea ice thickness, freezing point temperature, etc. We believe this step of the process can be considered a retrieval based on a simple model. We concede that accumulating the sea ice growth into an absolute sea ice thickness is more of a modeling exercise, albeit one that is heavily observationally constrained.

We have re-framed SLICE as a thermodynamic sea ice growth retrieval rather than a sea ice thickness product and propose the following title:

*A simple model for daily basin-wide thermodynamic sea ice thickness growth retrieval*

Nonetheless, the novel point of this study is SLICE method is independent of the atmospheric reanalysis generally used as the forcing to sea ice model. The most relevant study to the SLICE method will be Kang et al. (2021), which simulates the physical state of a snow- ice system by using a thermodynamic equation set forced by atmospheric reanalysis and nudged by satellite Tsi. This study has significance in terms of constructing an independent sea ice thickness record, while the physics of SLICE is very simplified compared to Kang et al. (2021) or other thermodynamic sea ice models. I recommend including an ice thickness comparison with the results of Kang et al. (2021). Their results are open to the public, and the authors can find the data repository in their paper. It is worth comparing the performance of the SLICE method with other sea ice models with more sophisticated physics and forced by reanalysis data.

We have added a comparison of SLICE initialized with CS2SMOS, PIOMAS and the model described by Kang et al. 2021 (hereafter K21) to Operation Ice Bridge (OIB) data (Kurtz, 2015). This comparison has been added to Sect. 4.2 (Ln 377-384). Additionally, we've added a comparison between SLICE and PIOMAS by comparing both to AWI CS2SMOS (Ln 385-403).

These comparisons show that all three models have similar performance when modeling sea ice thickness, even without SLICE including a deformation component and including a number of assumptions that allow SLICE to retrieve instantaneous thermodynamic thickness growth rate. This has become a prominent point of the paper (Ln 9-11, 438-447, 477-478).

4. Bias correction for satellite Tsi

The authors mentioned that "The resultant snow-ice interface temperatures were found to require a bias correction of 5 K in order to match buoy snow-ice interface temperatures..."). I have read Lee and Sohn (2015) and remember that the snow-ice interface derived from AMSR- E 6.9 GHz brightness temperatures are validated with buoy measured temperature. The validation result showed that the bias was less than 1 K, which is a very different result from the 5 K bias in the manuscript. Lee and Sohn (2015) also neglected atmospheric/snow absorption.

Regarding this situation, first I thought that it is possibly due to the bias within AMSR-E and AMSR2 measurements. However, the authors stated that the AMSR2 data has been intercalibrated with the AMSR-E data so this may not be the issue. Then, may the version of L3 brightness temperature be a problem? Or simply authors failed to reproduce the $T_{si}$ retrieval algorithm.

It is unclear why the results from Lee and Sohn (2015) seem to have not required a correction for atmospheric absorption. The physics described in that paper are valid at the surface but the brightness temperatures viewed by the satellite at 6.9 GHz will be affected by the atmosphere, which we are accounting for.

This is a very critical issue because sea ice thickness is determined by Tsi, which is the only real observation used for the sea ice thickness retrieval. The mentioned comparison result between buoy data and Tsi calculated by the authors showing 5 K bias must be presented (as an appendix) to justify the bias correction procedure. It will be worth reproducing figure 6 in Lee and Sohn (2015).

The "bias correction" is added to the satellite observed brightness temperature and is due to the slight absorption of 6.9 GHz radiation by the polar atmosphere. In order to better account for this, we have chosen to use a radiative transfer model. Details can be found in new section Sect. 3.1.1.

Whereas we previously had used a static 5 K correction, the resulting change to 6.9 GHz brightness temperatures affected by the modeled transmission term is very consistently near 5 K. The below figure shows mean and standard deviation atmospheric correction from atmospheric transmission to a 250 K brightness temperature during December, January and February (DJF) across the years 2003-2019. The Arctic basin shows a very spatially consistent roughly 4.5 K mean with standard deviations less than 0.1 K. These results are very similar to those reported by Burgard, et al., 2020 who used a geophysical model to simulate 6.9 GHz brightness temperature at TOA using MPI-ESM output data. They report a difference of 4.49 K between the model ice surface temperature and the simulated 6.9 GHz brightness temperature at TOA for pixels with 99% or greater sea ice concentration during the summer season when accounting

for columnar water vapor and columnar cloud liquid water. Though we've reported our DJF results here, our summer results are very similar. These results will not be shown in the manuscript but are relevant to this review response.

[Figure]

*Figure 3: AMSR-E and AMSR2 6.9 GHz channels brightness temperature correction at 250 K in the 2003-2019 DJF (a) mean and (b) standard deviation calculated using a radiative transfer model and ERA5 reanalysis data.The correction is consistently near 4.5 K.*

**5. Minor comments**

L29-L37: Please provide more details for relevant studies on sea ice thickness retrieval in order to emphasize the novelty or necessity of SLICE. How are the satellite altimetry methods limited in spatial coverage and temporal resolution (I think the resolution of ICESat-2 is better than passive microwave sensors such as AMSR2 6.9 GHz)? What are the limitations of the other methods? How is this study related to the existing studies?

We have added more quantitative information to this passage:

*Though the instruments aboard these satellites have relatively high spatial resolutions, it takes 28 days and 91 days, respectively, for CryoSat-2 and ICESat-2 to cover the entire Arctic due to their relatively low spatial coverage* (Ln 26-28)

L63: horizontally and vertically polarized...

You are correct, we have made this change. (Ln 68)

L215: Please define negative degree-days in the manuscript and provide what happens if the temperature is positive (melting?).

The negative degree-days term is defined in L216. SLICE is not capable of capturing melt.

L221: It is hard to know which equation was used for sea ice thickness calculation. Equation (4) is too general. Did you use equation (8) which is an analytic solution for sea ice thickness, or equation (7) for change in sea ice thickness per unit time and accumulate the thickness changes?

We have added Ln 265-266, clearly stating which equation is used.

L235-237: Why the retrieval method was initialized with such condition (the day when the 14 d rolling average sea ice growth exceeded 1mm d-1)? Please provide the reason.

We have updated the one-dimensional results to begin with the buoy initial condition on November 1 rather than the previous definition of a start time based on ice growth exceeding a threshold. This also reflects how the basin-wide methodology works. (Ln 370)

L400: I think uploading the data produced in this study to the public data repository more fits the data policy of TC journal.

Thank you for this suggestion. Data used in creation of all figures is now available at https://doi.org/10.5281/zenodo.6554832. Code for creation of data and figures is available at https://doi.org/10.5281/zenodo.6561431 and https://github.com/janheuser/SLICE/releases/tag/1.0.0.

References

Kang, E. J., Sohn, B. J., Tonboe, R. T., Dybkjaer, G., Holmlund, K., Kim, J. M., and Liu, C.: Implementation of a 1-D Thermodynamic Model for Simulating the Winter-Time Evolvement of Physical Properties of Snow and Ice Over the Arctic Ocean, J. Adv. Model. Earth Sy., 13, 3, https://doi.org/10.1029/2020ms002448, 2021.

Lee, S.-M and Sohn, B.-J.: Retrieving the refractive index, emissivity, and surface temperature of polar sea ice from 6.9 GHz microwave measurements: A theoretical development, J. Geophys. Res. Atmos., 120(6), 2293-2305, https://doi.org/10.1002/2014JD022481, 2015.

Maykut, G. A. and Untersteiner, N.: Some results from a time-dependent thermodynamic model of sea ice, J. Geophys. Res., 76, 1550–1575, https://doi.org/10.1029/JC076i006p01550, 1971.

Shi, H., Sohn, B.-J., Dybkjær, G., Tonboe, R. T., and Lee, S.-M.: Simultaneous estimation of wintertime sea ice thickness and snow depth from space-borne freeboard measurements, The Cryosphere, 14, 3761–3783, https://doi.org/10.5194/tc-14-3761-2020, 2020.

Burgard, C., Notz, D., Pedersen, L. T., and Tonboe, R. T.: The Arctic Ocean Observation Operator for 6.9 GHz (ARC3O) – Part 2: Development and evaluation, Cryosphere, 14, 2387–2407, https://doi.org/10.5194/tc-14-2387-2020, 2020.

Kurtz, N., M. Studinger, J. Harbeck, V. Onana, and D. Yi.: IceBridge L4 Sea Ice Freeboard, Snow Depth, and Thickness, Version 1. [2013/3/20-2013/3/27; 2014/3/12-2014/3/31; 2015/3/19-2015/3/30; 2017/3/9-2017/3/24; 2018/3/22]. Boulder, Colorado USA. NASA National Snow and Ice Data Center Distributed Active Archive Center. doi: https://doi.org/10.5067/G519SHCKWQV6. [2/2022]., 2015

---

## Referee Report (RR1)

**Referee report for "A simple model for daily basin-wide thermodynamic sea ice thickness growth retrieval" by James Anheuser et al.**

The authors have made a significant improvement to the manuscript addressing my comments. Especially, I appreciate the authors' hard work on the inclusion of sea ice drift in their methodology. The presented methodology and results in this manuscript are a novel and valuable addition to the sea ice community, but the manuscript still needs some clarifications. I, therefore, recommend the paper for publication following minor revisions.

**Comments**

Title: Although the key result of this study is the daily basin-wide sea ice thickness record, the new title does not include anything about sea ice thickness. I think it is better to show that this paper is about sea ice thickness in the title. How about something like "A simple thermodynamic model for simulating daily basin-wide sea ice thickness using satellite passive microwave measurements"?

L2: I think it is not appropriate to indicate the algorithm of Lee and Sohn (2015), which was developed 7 years ago, as a "recently" developed algorithm.

L11: The word "equally" overstates the result of this paper. I suggest using "comparable" instead. Moreover, the authors should state clearly which quantities are comparable between SLICE and PIOMAS.

L24-L33: I think the literature review is still weak compared to other sections. The novelty and significance of this study can be highlighted based on the solid literature review. There are various sea ice thickness retrieval methods besides the methods using space-based altimetry only. For example, there is an algorithm for thin sea ice, simultaneous estimation of snow and sea ice thickness by combining satellite altimeter and radiometer measurements (Zhou et al., 2018; Shi et al., 2020), and the simultaneous estimation using two satellite altimeters at different frequencies (Kwok et al., 2020). Or if this study focuses on retrieving the sea ice growth rate, then relevant studies should be introduced.

L49-L50: The methodology also requires good initial guesses for sea ice thickness as well as passive microwave observations.

L65: It *as* an → It *is* an?

L87-L106: I think these two paragraphs can be shortened and moved into the introduction section.

L112: Where did you get the CS2SMOS data? Please provide data availability information.

L123: Also please provide data availability information for the QuickLook product.

L129: Two acoustic rangefinder sounders positioned above and below the ice can measure sea ice thickness if there is no snow on sea ice. How can sea ice thickness be measured with snow presence?

L185: I think the sentence "In Lee et al. (2018) … in 1987" is not necessary.

L205: Explain why there are no upwelling and surface-reflected downwelling atmospheric TBs

in equations (2) and (3). I also want to confirm that the bias correction for estimated $T_{si}$ is not applied in the revised manuscript.

L209: horizontal, vertical → horizontally, vertically

Figure 2: I found Figure 2 is not mentioned in the manuscript. Please mention it at an appropriate place.

L332: What density value is used for MYI? I can see value for FYI only. Authors may refer to the most recent research by Jutila et al. (2021) and Lee et al. (2021) on the sea ice density issue.

L347: It is a little bit strange that Figure A1 appears after Figure A2.

L365: I see that SLICE sea ice thickness is generally greater than buoy sea ice thickness and the difference between them increases with time. It is better to make some discussion/explanation on these results.

Figure 3: Please explain the meaning of color (red and blue) in the figure caption. Y-axis has only two ticks, which is not quite informative. It would be better to make the figure more informative.

L395: The word "improved" may not be a good choice because CS2SMOS is not a ground truth measurement (snow depth, sea ice densities, etc. are assumed). It should be better to say such as "shows better consistency". Besides, I think the reason why SLICE is closer to CS2SMOS than PIOMAS is that SLICE uses CS2SMOS sea ice thickness value for its initial condition. Therefore, it becomes logical circulation if more consistency with CS2SMOS means "improvement".

Figure 5: I suggest exchanging the x- and y-axis. It is generally easier to read plots with the reference variable on the x-axis.

L405: I suggest removing the word "new".

L419-422: I don't think these sentences agree with the result shown in Figure 3. The difference between SLICE and buoy sea ice thickness increases with time even though the initial value is the same.

L427: These assumptions are reasonable because of what?

Figures 6 and 7: Is there a reason that K21 is not included in Figures 6 and 7, disturbing the consistency of the paper?

Figure 8: How did you calculate sea ice volume? Did you multiply sea ice concentration, grid area, and sea ice thickness? Is the 95% sea ice concentration criterion also applied to SC2SMOS and PIOMAS sea ice volume? Were three volumes calculated and compared based on the same area/criteria?

L443-444: Sea ice motion product used in the SLICE method also includes near-surface wind vectors from atmospheric reanalysis.

L479-481: Again, this conclusion can only be made upon a solid literature review. Please look for more state of art sea ice thickness observation methods.

**References**

Jutila, A., Hendricks, S., Ricker, R., von Albedyll, L., Krumpen, T., and Haas, C.: Retrieval and parameterisation of sea-ice bulk density from airborne multi-sensor measurements. The Cryosphere, 16(1), 259-276, 2022.

Kang, E. J., Sohn, B. J., Tonboe, R. T., Dybkjaer, G., Holmlund, K., Kim, J. M., and Liu, C.: Implementation of a 1-D Thermodynamic Model for Simulating the Winter-Time Evolvement of Physical Properties of Snow and Ice Over the Arctic Ocean, J. Adv. Model. Earth Sy., 13, 3, https://doi.org/10.1029/2020ms002448, 2021.

Kwok, R., Kacimi, S., Webster, M. A., Kurtz, N. T., and Petty, A. A.: Arctic snow depth and sea ice thickness from ICESat-2 and CryoSat-2 freeboards: A first examination. *J. Geophys. Res. Oceans*, 125, e2019JC016008, https://doi.org/10.1029/2019JC016008, 2020.

Lee, S.-M and Sohn, B.-J.: Retrieving the refractive index, emissivity, and surface temperature of polar sea ice from 6.9 GHz microwave measurements: A theoretical development, J. Geophys. Res. Atmos., 120(6), 2293-2305, https://doi.org/10.1002/2014JD022481, 2015.

Lee, S.-M., Shi, H., Sohn, B.-J., Gasiewski, A. J., Meier, W. N., and Dybkjær, G.: Winter snow depth on Arctic sea ice from satellite radiometer measurements (2003-2020): Regional patterns and trends. *Geophys. Res. Lett.*, 48(15), e2021GL094541, 2021.

Shi, H., Sohn, B.-J., Dybkjær, G., Tonboe, R. T., and Lee, S.-M.: Simultaneous estimation of wintertime sea ice thickness and snow depth from space-borne freeboard measurements, The Cryosphere, 14, 3761–3783, https://doi.org/10.5194/tc-14-3761-2020, 2020.

Zhou, L., Xu, S., Liu, J., and Wang, B.: On the retrieval of sea ice thickness and snow depth using concurrent laser altimetry and L-band remote sensing data, The Cryosphere, 12, 993–1012, https://doi.org/10.5194/tc-12-993-2018, 2018.

---

## Referee Report (RR2)

**Review of: A simple model for daily basin-wide thermodynamic sea ice thickness growth retrieval**

**1 Synopsis**

Firstly, congratulations to the authors on a strong set of responses to my last review. A lot of effort has clearly gone into this, and it has visibly improved the manuscript. I believe that the write up to this project can and should ultimately be published; driving a model with the snow-ice interface temperature is an original and interesting idea, and a lot of work has gone in to implementing the method.

However I have some remaining concerns about the paper presented here. These chiefly involve the design of the line-plot figures and the consideration of the contribution of dynamical processes to sea ice thickness. I'd also like to see an additional investigation involving the snow-ice temperatures actually measured at the buoys.

I would like to review the revised manuscript again before publication.

**2 Significant Comments**

**2.1 Figure 3**

This plot is not publication quality due to its design. The subplots do not succeed in fully displaying their data and conveying the information within. The subplots are too narrow, and the lines are so similar that they are almost indistinguishable in some cases. You also don't need to put 'Date' under every x axis. You could also probably move the panel-letter annotations so that the plots are less letterbox-like.

The plot also raises the question of how skillful your model actually is. It looks like the SLICE output is pretty linear - from this it's hard to see how much the slope actually responds to atmospheric conditions?

I think you should also critically consider the use of the r-value here. Specifically, if your model output just linearly and monotonically increased in value but had completely the wrong gradient, and your buoy-data also monotonically and linearly increased, the r value would be 1. Even though the rate of growth in your model is completely wrong. I think you need to come up with a better or additional statistic for evaluating your model. For instance 2013F shows a pretty clear divergence in the growth rates, but r=0.99. Whereas 2012L seems to show a very similar divergence, but it occurs later and delivers r=0.58. What justifies the r value being so much lower for these two model-runs when they're so visually similar?

It's also clear that your r-value is lower when the buoy-data exhibits thinning. This is confusing, as it often happens in the cold-season when I doubt that the ice is thermodynamically melting or thinning? Or perhaps it is? I think you should offer a (perhaps speculative) explanation about what's driving this thinning as it's clearly not captured by your model.

Finally your x-axis ticks presumably indicate the first of the month. If so, say this in the caption.

**2.2 Retrieving the Thermodynamic Component of the Thickness**

The authors now explicitly claim to be 'retrieving' the thermodynamic component of sea ice thickness (so do not consider the contribution of dynamical processes). I think it is theoretically possible to do this, but the validation of such a retrieval beyond 1D is nuanced. Specifically, actually observing the thermodynamic component of mean sea ice thickness in an area is difficult and this quantity is not necessarily represented by the mean thickness of a CS2/PIOMAS grid cell.

The sea ice cover in a specified area has a well-defined mean thickness, part of which is the result of thermodynamic growth (e.g. congelation/accretion), and part of which is the result of dynamical processes (thin ice formation in leads, ridging, rafting etc). We can conceptually break these components up into two

additive parts (see von Albedyll et al. (2022)). But actually observing these components separately in reality is hard work: The 2022 paper by von Albedyll et al. using airborne data indicates that dynamical processes contributed 30% to the mean ice thickness during MOSAiC, with Koo et al. (2021) getting similar numbers for the same period using ICESat-2. Work previous to that (von Albedyll et al., 2021) indicates 50 %. Kwok and Cunningham (2016) have similarly high numbers for the contribution of dynamics to sea ice thickness.

So for a given area (like a pixel of CS2SMOS or PIOMAS), the thermodynamic component of the sea ice thickness is potentially quite variable, and is potentially much less than the total thickness. The authors should therefore not expect their 'retrievals' of thermodynamic-SIT to match these total-SIT products. If they do match, then the authors are implying that *there is no dynamical contribution* to the sea ice thickness in the area of comparison. This must be addressed before publication.

However, for the 1D case of the ice-mass-balance buoys, I think the validation is immediately useful. The buoys measure thermodynamic growth at a point, which is what is being modelled by SLICE. I'm not sure what the situation would be for OIB, but this also needs to be considered in these respects. I think the case remains to be made about why SLICE would reflect the basin-wide sea ice volume evolution (Figure 8 & L448), given that it doesn't capture the dynamical contributions to sea ice thickness. If the authors are of the mind that it matches well because the dynamical effects are negligible, then they should say so. This should specifically be done with reference to the teardrop rheology/ mechanical redistribution effects that are incorporated within PIOMAS (which I am not an expert in).

**2.3 Use buoy snow-ice temperature to separate error in model from error in forcing**

In exploring when/why the buoy data diverges from your modelled SIT, it would make sense to check if some of the divergence is caused by inaccurate retrievals of the snow-ice temperature by the radiometers. It seems that this could be done very straightforwardly as the buoys presumably give you the whole temperature profile of the ice and snow. I think doing this would provide a valuable insight into the true performance of the model itself (rather than the integrated performance of the model and its forcing). It may also provide valuable support for your model's efficacy, if you can shift some of 'the blame' (as it were) onto your forcing.

**3 Specific Comments**

L26: I don't think it's right that ICESat-2 takes 91 days to 'cover the entire Arctic' - I think its orbit repeats every month or so? Have you just taken the ICESat-column from Table 1 of Wang et al. (2016)? But I'm also not really sure what it means for a laser altimeter (which generates spot heights) to 'cover' the Arctic Ocean. I guess you'd have to define some time for every location to fall within some max-distance of a spot height - but I don't think Wang et al. do this. That's not even getting into the pole-hole issues. My advice is to avoid getting into this territory, and to ditch the specific numbers and reword. The point is a good one - altimeters have small footprints and only measure a line of points/footprints (depending on whether they're lidar/radar) underneath them, so they provide incomplete sampling.

L26: 'relatively high' relative to what? A lot of people out there would say that CS2 has a 'relatively poor' spatial resolution, relative to the typical length-scales of sea ice floes/ridges/leads, or a SAR-image pixel, or the resolution of a visible image. I guess you mean it's good relative to radiometers - so say this.

L30: I clicked on the Mecklenburg ref and it didn't take me to any literature that supports our ability to estimate SIT? It just says there's an L3 product in development...

L31: Say that this is PIOMAS, as it will come up again later and you're introducing it now.

L48: Retrieve daily rates, or retrieve *the* daily rate.

L50: I would add 'provided initialisation data is available'

L65: 'as an' should be 'is an'

L75: Lagrangian should be capitalised here and elsewhere

L115: My understanding is that OIB had its final flight in 2019

L119: On L89 you describe the freeboard as the "thickness of the sea ice above sea level". But here you describe the ATM as retrieving sea ice freeboard, when it retrieves the 'total freeboard' (ice + snow). Needs a bit of clarification.

L120-123: I think you're using the GSFC-NK product as described by Kwok et al. (2017). This is the quicklook product available from NSIDC (do correct me if I'm wrong). It's pretty clear from Fig. 4 of that paper (Kwok et al., 2017) that the GSFC-NK method underestimates the BROMEX snow depths, probably due to the sidelobe issues explained in Kwok and Haas (2015). I think you need to point out that you're probably using the most unreliable OIB snow depth product in terms of sidelobes, and that may introduce biases in your snow depths and SIT. See also Webster et al. (2014) who state:

> *However, the IceBridge product appears to underestimate thin snow depths in comparison to the in situ averages, and a clear discrepancy can be seen around 50 m along Transect 2 (Figure 4); this discrepancy is also apparent in the scatter plot (Figure 5) where the IceBridge quick-look and standard products estimate a snow thickness of ~5-8 cm while the in situ mean is ~23 cm.*

Table 1: I'd like to see the deployment date and the date of final data acquisition, so we can have some idea of the lifetime of the instrument.

Figure 1a: I think the size of the triangle potentially obscures some of the track here. Can you make the marker smaller or outline it in black or something?

Figure 1b: Parts of the OIB tracks are off the map edge here: make sure all your data are displayed in the figure!

Sect 2.4: At some point here you need to make the link to dynamical thickening, and the fact that buoys cannot observe it. As soon as their ice begins to thicken dynamically (something that can contribute a significant amount of thickness!) the buoy dies.

L173: Thorndike citation should not be in parentheses

L175: So are you using a parcel-tracking approach to get the second term on the right? If so, say so. If not, explain why not.

L234: I'd probably ditch the voltage/pressure analogy here - too colloquial in tone. And although the flows are described by the same maths, the physics is of course different. You couldn't explain why heat flows by appealing to electrons or fluid flow.

L269: This sentence seems repetitive? Or am I missing something?

L270: Better define 'too thick' and 'too thin'. What's it's too thick with respect to?

Figures 5 & 7: I suggest you plot the y=x line on these subplots to help the reader interpret the data.

271: I think the Bitz and Roe citation is warranted and useful, but 'thin ice grows slower' is not really *the phenomenon described* by B&R. The phenomenon they describe is how thicker ice thins faster in response to a climate perturbation. And they explain that by appealing to the 'thick ice grows slower' mechanism.

L274: Basal *heat* flux

L285: I think you should point out somewhere in this para that you're talking about *sensible* heat flux. There's also heat flux that occurs at the interface from release of latent heat, but I guess you're not addressing that?

L295: This is a great para, very useful to see this info set out like this.

L298: Medium is singular of media

L338: I think 'leading' is a non-standard term. I did a search for it and couldn't find any uses in the literature as you're using it. Perhaps consider 'lead-forming'.

L350: initialised at 0.05 m *thickness*

Figure 6: Like Figure 3, this is poorly arranged leading to the subplots themselves being too small. One thing that would have significantly improved the size of potential subplots would be to put things like [SLICE-CS2SMOS] at the top of the column, and not repeat it every row (since it's the same). It's also noticeable that the colorbar labels often are uncomfortably close to these labels. Again, subplot letter annotation can be moved so as to allow more vertical space into which subplots can be expanded. You can probably get away with one colorbar. Since you're only displaying data for first April, no need to include this in every subplot. It's also unclear why there's a gap between the middle two columns when the others are so closely squeezed together that colorbar labels impinge on the row labels?

L451/452: I disagree about how dynamic processes do not directly change sea ice volume. Within a satellite footprint or model grid cell, it is easily possible to have a lead open through local convergence which is not reflected by the ice motion vectors. This lead will freeze, but then often be 'closed' very quickly by ice dynamics because a refrozen lead is rheologically weak. This accordion-like action can happen quite frequently, and it can rapidly increase the mean sea ice thickness in an area (see literature in bibliography). Of course sea ice only forms by freezing seawater, and in this sense all ice is thermodynamically grown and the 'dynamical contribution to thickness' does not really exist. But in the sense of the literature, dynamical processes can produce ice very quickly in quite a small area, relative to congelation growth of already established ice.

**References**

Koo, Y. H., Lei, R., Cheng, Y., Cheng, B., Xie, H., Hoppmann, M., Kurtz, N. T., Ackley, S. F., and Mestas-Nuñez, A. M.: Estimation of thermodynamic and dynamic contributions to sea ice growth in the Central Arctic using ICESat-2 and MOSAiC SIMBA buoy data, Remote Sensing of Environment, 267, https://doi.org/10.1016/j.rse.2021.112730, 2021.

Kwok, R. and Cunningham, G. F.: Contributions of growth and deformation to monthly variability in sea ice thickness north of the coasts of Greenland and the Canadian Arctic Archipelago, Geophysical Research Letters, 43, 8097–8105, https://doi.org/10.1002/2016GL069333, 2016.

Kwok, R. and Haas, C.: Effects of radar side-lobes on snow depth retrievals from Operation IceBridge, Journal of Glaciology, 61, https://doi.org/10.3189/2015JoG14J229, 2015.

Kwok, R., Kurtz, N. T., Brucker, L., Ivanoff, A., Newman, T., Farrell, S. L., King, J., Howell, S., Webster, M. A., Paden, J., Leuschen, C., MacGregor, J. A., Richter-Menge, J., Harbeck, J., and Tschudi, M.: Intercomparison of snow depth retrievals over Arctic sea ice from radar data acquired by Operation IceBridge, Cryosphere, 11, 2571–2593, https://doi.org/10.5194/tc-11-2571-2017, URL https://www.the-cryosphere.net/11/2571/2017/, 2017.

von Albedyll, L., Haas, C., and Dierking, W.: Linking sea ice deformation to ice thickness redistribution

using high-resolution satellite and airborne observations, Cryosphere, 15, 2167–2186, https://doi.org/
10.5194/TC-15-2167-2021, 2021.

von Albedyll, L., Hendricks, S., Grodofzig, R., Krumpen, T., Arndt, S., Belter, H. J., Birnbaum, G., Cheng,
B., Hoppmann, M., Hutchings, J., Itkin, P., Lei, R., Nicolaus, M., Ricker, R., Rohde, J., Suhrhoff, M.,
Timofeeva, A., Watkins, D., Webster, M., and Haas, C.: Thermodynamic and dynamic contributions
to seasonal Arctic sea ice thickness distributions from airborne observations, Elementa: Science of the
Anthropocene, 10, https://doi.org/10.1525/ELEMENTA.2021.00074, 2022.

Wang, X., Key, J., Kwok, R., and Zhang, J.: Comparison of Arctic sea ice thickness from satellites, aircraft,
and PIOMAS data, Remote Sensing, 8, 1–17, https://doi.org/10.3390/rs8090713, 2016.

Webster, M. A., Rigor, I. G., Nghiem, S. V., Kurtz, N. T., Farrell, S. L., Perovich, D. K., Sturm, M., Webster,
M. A., Rigor, I. G., Nghiem, S. V., Kurtz, N. T., Farrell, S. L., Perovich, D. K., and Sturm, M.: Interdecadal
changes in snow depth on Arctic sea ice, Journal of Geophysical Research : Oceans, 119, 5395–5406,
https://doi.org/10.1002/2014JC009985.Received, URL `http://doi.wiley.com/10.1002/2014JC009985`,
2014.

---

## Referee Report (RR3)

**Review of: A simple model for daily basin-wide thermodynamic sea ice thickness growth retrieval**

**1  Synopsis**

Congratulations again to the authors for another round of good responses. I should apologize for the substantially delayed review on my end. I'm pleased to say I'm very satisfied with the responses and revised manuscript. I think the paper now represents a rigorous and interesting contribution to the literature on sea ice thickness - in particular I think it has a strong element of literature review, and it is pleasantly surprising how well the product does given its relatively low complexity and number of data inputs.

I have a couple of remaining comments. If the editor is satisfied that the authors have suitably responded to these, then I am happy for the manuscript to be accepted without further review.

**2  Comments**

Fig. 2: The label on the colorbar should be padded away from the tick labels to improve readability. This looks like a python plot - if so, this can be done with the *labelpad* keyword.

Fig 3: This is a good figure, but needs a bit of cleaning up. I don't think the colorbar label needs '[-]' in it. Furthermore, a line break would help improve readability for the right-hand panel's y-axis label, and the subplots could do with being moved apart a bit (fig.subplots_adjust(wspace)). You have lots of horizontal space to do this, so it should be straightforward. You also need (a) & (b) annotations to match the caption. Finally, the text annotations in both panels need proper right-hand-justifying, if you're using python you can do this with the *ha* keyword passed to ax.annotate().

The same points from Fig. 3 also apply to Fig. 6 & 8. Colorbars could also do with a tick-label at 0.

L458: 'instantaneous'

L501: That's a nice point about not being dependent on atmospheric reanalysis, consider putting this in your abstract.

---

## Author Response (AR2)

Response to Referee #1 "Review of: A simple model for daily basin-wide thermodynamic sea ice thickness growth retrieval"

By Anheuser, et al.

Author responses in red.

**1 Synopsis**

Firstly, congratulations to the authors on a strong set of responses to my last review. A lot of effort has clearly gone into this, and it has visibly improved the manuscript. I believe that the write up to this project can and should ultimately be published; driving a model with the snow-ice interface temperature is an original and interesting idea, and a lot of work has gone in to implementing the method.

However I have some remaining concerns about the paper presented here. These chiefly involve the design of the line-plot figures and the consideration of the contribution of dynamical processes to sea ice thickness. I'd also like to see an additional investigation involving the snow-ice temperatures actually measured at the buoys.

I would like to review the revised manuscript again before publication.

**2 Significant Comments**

**2.1 Figure 3**

This plot is not publication quality due to its design. The subplots do not succeed in fully displaying their data and conveying the information within. The subplots are too narrow, and the lines are so similar that they are almost indistinguishable in some cases. You also don't need to put 'Date' under every x axis. You could also probably move the panel-letter annotations so that the plots are less letterbox-like.

The plot also raises the question of how skillful your model actually is. It looks like the SLICE output is pretty linear - from this it's hard to see how much the slope actually responds to atmospheric conditions?

We have redesigned the figure to allow clearer viewing of its content. The panel-letter annotations have been moved to the side of each plot and the y axis range allowed to vary from plot to plot. Previous versions of this figure locked the y axis range between each so that vertical distance equated to the same thickness from plot to plot but by now allowing the range to vary, variations in the profiles over time are clearer. Variations in slope along both the SLICE and buoy profiles are more clear.

I think you should also critically consider the use of the r-value here. Specifically, if your model output just linearly and monotonically increased in value but had completely the wrong gradient, and your buoy- data also monotonically and linearly increased, the r value would be 1. Even though the rate of growth in your model is completely wrong. I think you need to come up with a better or additional statistic for evaluating your model. For instance 2013F shows a pretty clear divergence in the growth rates, but r=0.99. Whereas 2012L seems to show a very similar divergence, but it occurs later and delivers r=0.58. What justifies the r value being so much lower for these two model-runs when they're so visually similar?

This is a valid point. In order to more clearly validate SLICE's ability to retrieve thermodynamic sea ice thickness growth, we have added new Fig. 3, shown below. This figure compares the input retrieved snow—ice interface temperature to buoy snow ice interface temperature and SLICE thermodynamic sea ice thickness growth rate (computed using buoy thickness) to buoy thermodynamic sea ice thickness growth rate.

[Figure]

Figure 1: An instantaneous comparison of (a) AMSR-E/AMSR2 retrieved snow–ice interface temperature to buoy snow–ice interface tem perature and (b) SLICE retrieved thermodynamic growth rate using buoy thickness to buoy thermodynamic growth rate. Both exhibit a linear correlation of 0.71.

It's also clear that your r-value is lower when the buoy-data exhibits thinning. This is confusing, as it often happens in the cold-season when I doubt that the ice is thermodynamically melting or thinning? Or perhaps it is? I think you should offer a (perhaps speculative) explanation about what's driving this thinning as it's clearly not captured by your model.

We have added the following passage to paragraph #2 of the discussion:

*Buoys 2006C, 2012L and 2013G show a SLICE profile that produces greater sea ice thickness than the buoys. There are likely two mechanisms causing this error. For buoys 2006C and 2012L, the initial thicknesses are the two highest of the set and are near 3 m. In these cases, the cold atmospheric temperatures of the growth season have not yet reached the base of the ice, which must be below the freezing point in order for thickness to increase. In other words, the heat stored in the ice from summer has not yet escaped due to the higher thickness and greater heat storing capacity. SLICE assumes a linear temperature profile with the sea ice base at the freezing points---a condition that is not met in reality by buoys 2006C and 2012L until after 1 November. In the case of 2013G, a melt event, which SLICE is unable to capture, occurs in December. Both of these phenomena cause SLICE to overestimate sea ice thickness.*

Finally your x-axis ticks presumably indicate the first of the month. If so, say this in the caption.

We have updated the tick labels in this figure to show the first of the month.

**2.2 Retrieving the Thermodynamic Component of the Thickness**

The authors now explicitly claim to be 'retrieving' the thermodynamic component of sea ice thickness (so do not consider the contribution of dynamical processes). I think it is theoretically possible to do this, but the validation of such a retrieval beyond 1D is nuanced. Specifically, actually observing the thermodynamic component of mean sea ice thickness in an area is difficult and this quantity is not necessarily represented by the mean thickness of a CS2/PIOMAS grid cell.

The sea ice cover in a specified area has a well-defined mean thickness, part of which is the result of thermodynamic growth (e.g. congelation/accretion), and part of which is the result of dynamical processes (thin ice formation in leads, ridging, rafting etc). We can conceptually break these components up into two additive parts (see von Albedyll et al. (2022)). But actually observing these components separately in reality is hard work: The 2022 paper by von Albedyll et al. using airborne data indicates that dynamical processes contributed 30% to the mean ice

thickness during MOSAiC, with Koo et al. (2021) getting similar numbers for the same period using ICESat-2. Work previous to that (von Albedyll et al., 2021) indicates 50 %. Kwok and Cunningham (2016) have similarly high numbers for the contribution of dynamics to sea ice thickness.

So for a given area (like a pixel of CS2SMOS or PIOMAS), the thermodynamic component of the sea ice thickness is potentially quite variable, and is potentially much less than the total thickness. The authors should therefore not expect their 'retrievals' of thermodynamic-SIT to match these total-SIT products. If they do match, then the authors are implying that there is no dynamical contribution to the sea ice thickness in the area of comparison. This must be addressed before publication.

The authors certainly agree with these points. Average sea ice thickness across a satellite footprint or L3 grid cell likely does not yield a perfectly accurate characterization of mean thermodynamic sea ice growth across that footprint. Additionally, leaving out deformation processes like ridging and lead formation does not capture all process affecting sea ice thickness. A perfect match of SLICE to CS2SMOS is not necessarily the goal. Even still, we believe it is appropriate to show the basin-wide results as it lends credence to the efficacy of SLICE. SLICE is a novel, alternative method for modelling sea ice thickness and can be applied on a basin-wide basis, even if we don't yet fully understand the implications of the results. We've added the following passage to paragraph #5 of the discussion:

*Indeed, thermodynamic growth rate calculated using average thickness over a 25 km x 25 km grid cell likely does not completely accurately describe thermodynamic growth rate over the entirety that grid cell. Additionally, SLICE does not include the effects of deformation processes, which recently were shown to have contributed roughly 30% of total thickness growth during the Multidisciplinary drifting Observatory for the Study of Arctic Climate (MOSAiC; Nicolaus et al., 2022) field campaign (von Albedyll et al., 2022; Koo et al., 2021).*

However, for the 1D case of the ice-mass-balance buoys, I think the validation is immediately useful. The buoys measure thermodynamic growth at a point, which is what is being modelled by SLICE. I'm not sure what the situation would be for OIB, but this also needs to be considered in these respects. I think the case remains to be made about why SLICE would reflect the basin-wide sea ice volume evolution (Figure 8 & L448), given that it doesn't capture the dynamical contributions to sea ice thickness. If the authors are of the mind that it matches well because the dynamical effects are negligible, then they should say so. This should specifically be done with reference to the teardrop rheology/ mechanical redistribution effects that are incorporated within PIOMAS (which I am not an expert in).

Even with a lack of deformation processes, which we regard as a source of uncertainty rather than a disqualifying characteristic, SLICE compares similarly to CS2SMOS as PIOMAS. By apparently capturing thermodynamic growth better than PIOMAS, SLICE makes up for this source of uncertainty.

**2.3 Use buoy snow-ice temperature to separate error in model from error in forcing**

In exploring when/why the buoy data diverges from your modelled SIT, it would make sense to check if some of the divergence is caused by inaccurate retrievals of the snow-ice temperature by the radiometers. It seems that this could be done very straightforwardly as the buoys presumably give you the whole temperature profile of the ice and snow. I think doing this would provide a valuable insight into the true performance of the model itself (rather than the integrated performance of the model and its forcing). It may also provide valuable support for your model's efficacy, if you can shift some of 'the blame' (as it were) onto your forcing.

This is a worthwhile exercise. We have added new Fig. 3, mentioned in Section 2.1 results.

**3 Specific Comments**

L26: I don't think it's right that ICESat-2 takes 91 days to 'cover the entire Arctic' - I think its orbit repeats every month or so? Have you just taken the ICESat-column from Table 1 of Wang et al. (2016)? But I'm also not really

sure what it means for a laser altimeter (which generates spot heights) to 'cover' the Arctic Ocean. I guess you'd have to define some time for every location to fall within some max-distance of a spot height - but I don't think Wang et al. do this. That's not even getting into the pole-hole issues. My advice is to avoid getting into this territory, and to ditch the specific numbers and reword. The point is a good one - altimeters have small footprints and only measure a line of points/footprints (depending on whether they're lidar/radar) underneath them, so they provide incomplete sampling.

We agree that quantifying this aspect is not necessary, however referee #2 had asked for more detail to this passage. In an effort to satisfy both requests, we have added additional citations and background regarding existing sea ice thickness measurement and removed the quantitative discussion of the altimeter characteristics.

L26: 'relatively high' relative to what? A lot of people out there would say that CS2 has a 'relatively poor' spatial resolution, relative to the typical length-scales of sea ice floes/ridges/leads, or a SAR-image pixel, or the resolution of a visible image. I guess you mean it's good relative to radiometers - so say this.

You make a good point. We've removed this phrase and just replaced it with "However,".

L30: I clicked on the Mecklenburg ref and it didn't take me to any literature that supports our ability to estimate SIT? It just says there's an L3 product in development...

We have updated this to a more appropriate citation of a reference that describes the algorithm and shows an initial validation.

L31: Say that this is PIOMAS, as it will come up again later and you're introducing it now.

We have added this.

L48: Retrieve daily rates, or retrieve the daily rate.

This has been updated.

L50: I would add 'provided initialisation data is available'

We have updated the sentence to:

"By using the satellite retrieved snow–ice interface temperature and an assumed initial ice thickness value in this relationship,…."

L65: 'as an' should be 'is an'

This has been updated.

L75: Lagrangian should be capitalised here and elsewhere

This has been updated.

L115: My understanding is that OIB had its final flight in 2019

It appears there was an additional OIB flight in 2020. We have updated the manuscript accordingly.

L119: On L89 you describe the freeboard as the "thickness of the sea ice above sea level". But here you describe the ATM as retrieving sea ice freeboard, when it retrieves the 'total freeboard' (ice + snow). Needs a bit of clarification.

We have updated the sentence to:

"The ATM is a laser altimeter whose return signal is used along with an aerial photography based sea ice lead (fracture) discrimination algorithm to retrieve sea ice total freeboard height---i.e., freeboard plus snow depth---at 40 m spatial resolution."

L120-123: I think you're using the GSFC-NK product as described by Kwok et al. (2017). This is the quicklook product available from NSIDC (do correct me if I'm wrong). It's pretty clear from Fig. 4 of that paper (Kwok et al., 2017) that the GSFC-NK method underestimates the BROMEX snow depths, probably due to the sidelobe issues explained in Kwok and Haas (2015). I think you need to point out that you're probably using the most unreliable OIB snow depth product in terms of sidelobes, and that may introduce biases in your snow depths and SIT. See also Webster et al. (2014) who state:

However, the IceBridge product appears to underestimate thin snow depths in comparison to the in situ averages, and a clear discrepancy can be seen around 50 m along Transect 2 (Figure 4); this discrepancy is also apparent in the scatter plot (Figure 5) where the IceBridge quick-look and standard products estimate a snow thickness of ~5-8 cm while the in situ mean is ~23 cm.

The Quick Look data is indeed as described in Kwok et al. (2017). We have added the following sentence to this passage:

"The data from 2014-2019, covering all of the data used here except those form 2013, are only available in the Quick Look format and may have increased uncertainties due to a roughly 5 cm underestimation of snow depths by the snow radar waveform retracking method Kwok et al. (2017)."

Table 1: I'd like to see the deployment date and the date of final data acquisition, so we can have some idea of the lifetime of the instrument.

We have added a final acquisition data column.

Figure 1a: I think the size of the triangle potentially obscures some of the track here. Can you make the marker smaller or outline it in black or something?

We have replaced the triangle with a plus symbol which seems to be less intrusive to the path lines.

Figure 1b: Parts of the OIB tracks are off the map edge here: make sure all your data are displayed in the figure!

We have expanded the extent of this figure allowing for all paths to be viewed.

Sect 2.4: At some point here you need to make the link to dynamical thickening, and the fact that buoys cannot observe it. As soon as their ice begins to thicken dynamically (something that can contribute a significant amount of thickness!) the buoy dies.

We have updated this passage to be as follows:

"All buoys from the years 2003 to 2016 showing an entire season of sea ice thickness growth were used for comparison with the exception of buoys installed in landfast ice and those that show obvious ice deformation effects, which often lead to the end of data acquisition. As such, sea ice thickness growth observed by the buoys is taken to be caused strictly by thermodynamic processes."

L173: Thorndike citation should not be in parentheses

These has been updated.

L175: So are you using a parcel-tracking approach to get the second term on the right? If so, say so. If not, explain why not.

We have updated the last sentence in this paragraph to the following:

"Through the following methodology called Stefan's Law Integrated Conducted Energy (SLICE), we retrieve the first term on the right hand side of Eq. 1 and use a parcel tracking approach to approximate the sea ice advection component of the second term on the right in order to model basin-wide thickness."

L234: I'd probably ditch the voltage/pressure analogy here - too colloquial in tone. And although the flows are described by the same maths, the physics is of course different. You couldn't explain why heat flows by appealing to electrons or fluid flow.

This sentence has been removed.

L269: This sentence seems repetitive? Or am I missing something?

This sentence could easily be seen as repetitive—we have removed the repetitive wording.

L270: Better define 'too thick' and 'too thin'. What's it's too thick with respect to?

We have updated this sentence to the following:

"This means that in the presence of only thermodynamic effects, a SLICE sea ice thickness profile that is biased relative to ground truth will correct towards the unbiased SLICE thickness profile."

Figures 5 & 7: I suggest you plot the y=x line on these subplots to help the reader interpret the data.

We have added the 1:1 line on these plots.

271: I think the Bitz and Roe citation is warranted and useful, but 'thin ice grows slower' is not really the phenomenon described by B&R. The phenomenon they describe is how thicker ice thins faster in response to a climate perturbation. And they explain that by appealing to the 'thick ice grows slower' mechanism.

We have updated this sentence to the following:

"This relationship replicates the phenomenon described in Bitz and Roe (2004), whereby thick ice recovers from climate related perturbations slower than thin ice and has experienced greater thinning on a decadal time scale."

L274: Basal heat flux

Thanks, this has been updated.

L285: I think you should point out somewhere in this para that you're talking about sensible heat flux. There's also heat flux that occurs at the interface from release of latent heat, but I guess you're not address- ing that?

Indeed we are working with sensible heat here. We have added the term "sensible" throughout.

L295: This is a great para, very useful to see this info set out like this.

Thank you.

L298: Medium is singular of media

We have updated this term to medium.

L338: I think 'leading' is a non-standard term. I did a search for it and couldn't find any uses in the literature as you're using it. Perhaps consider 'lead-forming'.

We have updated this term to "lead formation" in the two locations where "leading" was used.

L350: initialised at 0.05 m thickness

New ice parcels are initialized at a thickness of 0.05 m.

Figure 6: Like Figure 3, this is poorly arranged leading to the subplots themselves being too small. One thing that would have significantly improved the size of potential subplots would be to put things like [SLICE- CS2SMOS] at the top of the column, and not repeat it every row (since it's the same). It's also noticeable that the colorbar labels often are uncomfortably close to these labels. Again, subplot letter annotation can be moved so as to allow more vertical space into which subplots can be expanded. You can probably get away with one colorbar. Since you're only displaying data for first April, no need to include this in every subplot. It's also unclear why there's a gap between the middle two columns when the others are so closely squeezed together that colorbar labels impinge on the row labels?

The gap in the middle was to signify that this is a two column that has been split into two halves which are placed side by side. The titles also signify this so we have removed the gap. We have also taken your other suggestions, all of which have improved the figure greatly. We have also applied all of these changes to Fig. A2.

L451/452: I disagree about how dynamic processes do not directly change sea ice volume. Within a satellite footprint or model grid cell, it is easily possible to have a lead open through local convergence which is not reflected by the ice motion vectors. This lead will freeze, but then often be 'closed' very quickly by ice dy- namics because a refrozen lead is rheologically weak. This accordion-like action can happen quite frequently, and it can rapidly increase the mean sea ice thickness in an area (see literature in bibliography). Of course sea ice only forms by freezing seawater, and in this sense all ice is thermodynamically grown and the 'dynamical contribution to thickness' does not really exist. But in the sense of the literature, dynamical processes can produce ice very quickly in quite a small area, relative to congelation growth of already established ice.

This effect is indeed what we intended to describe by the following statement:

*Though dynamic processes do not directly change sea ice volume, their changing of the thickness of ice at a given location does impact thermodynamic processes by virtue of f being a function of thickness, H, in Eq. (1). Inspection of Eq. (7) indeed shows that H impacts $\partial H/\partial t$ .*

By this we mean that within a given time step, thermodynamic and dynamic process are independent . As you describe and as we hoped to capture, dynamics do change the overall thickness field which effects the thermodynamic growth that increases volume. Indeed, we don't capture those types of processes. We will add the phrase "within a given time step" to the above passage.

**References**

Koo, Y. H., Lei, R., Cheng, Y., Cheng, B., Xie, H., Hoppmann, M., Kurtz, N. T., Ackley, S. F., and Mestas-Nuñez, A. M.: Estimation of thermodynamic and dynamic contributions to sea ice growth in the Central Arctic using ICESat-2 and MOSAiC SIMBA buoy data, Remote Sensing of Environment, 267, https://doi.org/10.1016/j.rse.2021.112730, 2021.

Kwok, R. and Cunningham, G. F.: Contributions of growth and deformation to monthly variability in sea ice thickness north of the coasts of Greenland and the Canadian Arctic Archipelago, Geophysical Research Letters, 43, 8097–8105, https://doi.org/10.1002/2016GL069333, 2016.

Kwok, R. and Haas, C.: Effects of radar side-lobes on snow depth retrievals from Operation IceBridge, Journal of Glaciology, 61, https://doi.org/10.3189/2015JoG14J229, 2015.

Kwok, R., Kurtz, N. T., Brucker, L., Ivanoff, A., Newman, T., Farrell, S. L., King, J., Howell, S., Webster, M. A., Paden, J., Leuschen, C., MacGregor, J. A., Richter-Menge, J., Harbeck, J., and Tschudi, M.: Intercomparison of snow depth retrievals over Arctic sea ice from radar data acquired by Operation IceBridge, Cryosphere, 11, 2571–2593, https://doi.org/10.5194/tc-11-2571-2017, URL https://www.the-cryosphere.net/11/2571/2017/, 2017.

von Albedyll, L., Haas, C., and Dierking, W.: Linking sea ice deformation to ice thickness redistribution using high-resolution satellite and airborne observations, Cryosphere, 15, 2167–2186, https://doi.org/ 10.5194/TC-15-2167-2021, 2021.

von Albedyll, L., Hendricks, S., Grodofzig, R., Krumpen, T., Arndt, S., Belter, H. J., Birnbaum, G., Cheng, B., Hoppmann, M., Hutchings, J., Itkin, P., Lei, R., Nicolaus, M., Ricker, R., Rohde, J., Suhrhoff, M., Timofeeva, A., Watkins, D., Webster, M., and Haas, C.: Thermodynamic and dynamic contributions to seasonal Arctic sea ice thickness distributions from airborne observations, Elementa: Science of the Anthropocene, 10, https://doi.org/10.1525/ELEMENTA.2021.00074, 2022.

Wang, X., Key, J., Kwok, R., and Zhang, J.: Comparison of Arctic sea ice thickness from satellites, aircraft, and PIOMAS data, Remote Sensing, 8, 1–17, https://doi.org/10.3390/rs8090713, 2016.

Webster, M. A., Rigor, I. G., Nghiem, S. V., Kurtz, N. T., Farrell, S. L., Perovich, D. K., Sturm, M., Webster, M. A., Rigor, I. G., Nghiem, S. V., Kurtz, N. T., Farrell, S. L., Perovich, D. K., and Sturm, M.: Interdecadal changes in snow depth on Arctic sea ice, Journal of Geophysical Research : Oceans, 119, 5395–5406, https://doi.org/10.1002/2014JC009985.Received, URL http://doi.wiley.com/10.1002/2014JC009985, 2014.

**Response to Referee #2 "Referee report for "A simple model for daily basin-wide thermodynamic sea ice thickness growth retrieval" by James Anheuser et al."**

**By Anheuser, et al.**

Author responses in red.

The authors have made a significant improvement to the manuscript addressing my comments. Especially, I appreciate the authors' hard work on the inclusion of sea ice drift in their methodology. The presented methodology and results in this manuscript are a novel and valuable addition to the sea ice community, but the manuscript still needs some clarifications. I, therefore, recommend the paper for publication following minor revisions.

**Comments**

Title: Although the key result of this study is the daily basin-wide sea ice thickness record, the new title does not include anything about sea ice thickness. I think it is better to show that this paper is about sea ice thickness in the title. How about something like "A simple thermodynamic model for simulating daily basin-wide sea ice thickness using satellite passive microwave measurements"?

We think the current title more accurately captures the content of the work. It is indeed a basin-wide sea ice thickness record, but only thermodynamic effects are captured. The title change from the first revision was a request from referee #1.

L2: I think it is not appropriate to indicate the algorithm of Lee and Sohn (2015), which was developed 7 years ago, as a "recently" developed algorithm.

We agree, "recently" has been removed.

L11: The word "equally" overstates the result of this paper. I suggest using "comparable" instead. Moreover, the authors should state clearly which quantities are comparable between SLICE and PIOMAS.

We have changed this passage to:

"basin-wide SLICE has equal linear correlation values to PIOMAS of 0.77 and 0.67 when compared against CryoSat-2 and Operation IceBridge, respectively."

L24-L33: I think the literature review is still weak compared to other sections. The novelty and significance of this study can be highlighted based on the solid literature review. There are various sea ice thickness retrieval methods besides the methods using space-based altimetry only. For example, there is an algorithm for thin sea ice, simultaneous estimation of snow and sea ice thickness by combining satellite altimeter and radiometer measurements (Zhou et al., 2018; Shi et al., 2020), and the simultaneous estimation using two satellite altimeters at different frequencies (Kwok et al., 2020). Or if this study focuses on retrieving the sea ice growth rate, then relevant studies should be introduced.

We have added the studies recommended above to the introduction section.

L49-L50: The methodology also requires good initial guesses for sea ice thickness as well as passive microwave observations.

We have added "and an assumed initial ice thickness value".

L65: It *as* an→It *is* an?

Yes, this has been updated.

L87-L106: I think these two paragraphs can be shortened and moved into the introduction section.

We have slightly shortened the first of these paragraphs slightly but feel the information contained is important background regarding the instruments providing the data we use in the rest of the paper. We also feel it belongs in the data section for the same reason.

L112: Where did you get the CS2SMOS data? Please provide data availability information.

This information is in the "Data Availability" section.

L123: Also please provide data availability information for the QuickLook product.

We have added a citation for the QL data and added it to the "Data Availability" section.

L129: Two acoustic rangefinder sounders positioned above and below the ice can measure sea ice thickness if there is no snow on sea ice. How can sea ice thickness be measured with snow presence?

We have added the following sentence:

"Winter sea ice growth is derived from the under--ice sounder."

L185: I think the sentence "In Lee et al. (2018) ... in 1987" is not necessary.

We believe this sentence to be important as it describes the time period over which SLICE is viable.

L205: Explain why there are no upwelling and surface-reflected downwelling atmospheric TBs

in equations (2) and (3). I also want to confirm that the bias correction for estimated $T_{si}$ is not applied in the revised manuscript.

The exclusion of upwelling and surface-reflected downwelling atmospheric flux was an oversight in the last revision. After including these fluxes in radiative balance equation, we find that the ~5 K bias, present in revision 1, remains.

In an effort to utilize unbiased snow—ice interface temperatures, we have adopted a new methodology for retrieving them. We have adopted the methodology described by Kilic et al. (2019). This methodology is a multi-linear regression between buoy snow—ice interface temperatures and AMSR2 brightness temperatures and includes a snow depth term. The results of this algorithm are accurate and do not show a bias, improving the findings of our own work. Additionally, this method does not require reanalysis data. We have updated Section 3.1 to reflect this new snow—ice interface temperature methodology.

Though this change to our methodology may seem significant, the paper's objective of driving a simple thermodynamic sea ice model with satellite retrieved snow—ice interface temperatures in order to retrieve thermodynamic sea ice growth remains unchanged. Some of the basin-wide comparison results have changed relative to revision 2 but the overall finding of SLICE as an effective strategy for retrieving thermodynamic sea ice growth remains.

L209: horizontal, vertical→horizontally, vertically

This has been updated.

Figure 2: I found Figure 2 is not mentioned in the manuscript. Please mention it at an

appropriate place.

This must have been lost during revision, we have added the following:

Figure 2 shows snow-ice interface temperatures on November 1, 2013 derived from AMSR2 data.

L332: What density value is used for MYI? I can see value for FYI only. Authors may refer to the most recent research by Jutila et al. (2021) and Lee et al. (2021) on the sea ice density issue.

All ice added by SLICE is new ice that has not experience a melt season and therefore is FYI ice (even if it has been added to existing MYI ice thickness). As such, only FYI density is used by SLICE.

L347: It is a little bit strange that Figure A1 appears after Figure A2.

Yes, that is strange and has been updated.

L365: I see that SLICE sea ice thickness is generally greater than buoy sea ice thickness and the difference between them increases with time. It is better to make some discussion/explanation on these results.

We have added the following explanation to the discussion section:

*Buoys 2006C, 2012L and 2013G show a SLICE profile that produces greater sea ice thickness than the buoys. There are likely two mechanisms causing this error. For buoys 2006C and 2012L, the initial thicknesses are the two highest of the set and are near 3 m. In these cases, the cold atmospheric temperatures of the growth season have not yet reached the base of the ice, which must be below the freezing point in order for thickness to increase. In other words, the heat stored in the ice from summer has not yet escaped due to the higher thickness and greater heat storing capacity. SLICE assumes a linear temperature profile with the sea ice base at the freezing points---a condition that is not met in reality by buoys 2006C and 2012L until after 1 November. In the case of 2013G, a melt event, which SLICE is unable to capture, occurs in December. Both of these phenomena cause SLICE to overestimate sea ice thickness.*

Figure 3: Please explain the meaning of color (red and blue) in the figure caption. Y-axis has only two ticks, which is not quite informative. It would be better to make the figure more informative.

We have added the color description to the caption of and made changes to this figure to make it more clear. The y-axis range now varies from pane to pane and we have increased the number of ticks. We have also added grid lines for this purpose.

L395: The word "improved" may not be a good choice because CS2SMOS is not a ground truth measurement (snow depth, sea ice densities, etc. are assumed). It should be better to say such as "shows better consistency". Besides, I think the reason why SLICE is closer to CS2SMOS than PIOMAS is that SLICE uses CS2SMOS sea ice thickness value for its initial condition. Therefore, it becomes logical circulation if more consistency with CS2SMOS means "improvement".

Your critique is valid, we will replace "improved" with "lower".

Figure 5: I suggest exchanging the x- and y-axis. It is generally easier to read plots with the reference variable on the x-axis.

We have made this change.

L405: I suggest removing the word "new".

We have removed the word "new".

L419-422: I don't think these sentences agree with the result shown in Figure 3. The difference between SLICE and buoy sea ice thickness increases with time even though the initial value is the same.

For most of the buoys, the bias doesn't increase significantly over time and the SLICE profile follows well. Prior to these lines, we discuss reasons why some buoy results are not as good.

L427: These assumptions are reasonable because of what?

These are reasonably because they are typical in sea ice analysis.

Figures 6 and 7: Is there a reason that K21 is not included in Figures 6 and 7, disturbing the consistency of the paper?

The K21 dataset does not include all of the years that we have used in this comparison. Rather than separating out the years into two figures, we have left K21 out of the comparison. Additionally, K21 includes its own comparison to CryoSat-2 data within the publication.

Figure 8: How did you calculate sea ice volume? Did you multiply sea ice concentration, grid area, and sea ice thickness? Is the 95% sea ice concentration criterion also applied to SC2SMOS and PIOMAS sea ice volume? Were three volumes calculated and compared based on the same area/criteria?

Sea ice concentration did not factor into the calculation. Each of the three volumes are total volumes directly from the product. SLICE and CS2SMOS volumes are thickness of all cells with sea ice multiplied by grid area. PIOMAS volume is taken from PIOMAS output data available at the link listed in the data availability section. We have added the following statement to the discussion of this figure:

*The 95% or greater critera for SLICE may also contribute to differences, as the other datasets are not limited by this threshold.*

L443-444: Sea ice motion product used in the SLICE method also includes near-surface wind vectors from atmospheric reanalysis.

We of course do not dispute this, however the sentence states that it is referring to the thermodynamics of each of the three products.

L479-481: Again, this conclusion can only be made upon a solid literature review. Please look for more state of art sea ice thickness observation methods.

We have added the studies listed above in the introduction section comments.

**References**

Jutila, A., Hendricks, S., Ricker, R., von Albedyll, L., Krumpen, T., and Haas, C.: Retrieval and parameterisation of sea-ice bulk density from airborne multi-sensor measurements. The Cryosphere, 16(1), 259-276, 2022.

Kang, E. J., Sohn, B. J., Tonboe, R. T., Dybkjaer, G., Holmlund, K., Kim, J. M., and Liu, C.: Implementation of a 1-D Thermodynamic Model for Simulating the Winter-Time Evolvement of Physical Properties of Snow and Ice Over the Arctic Ocean, J. Adv. Model. Earth Sy., 13, 3, https://doi.org/10.1029/2020ms002448, 2021.

Kwok, R., Kacimi, S., Webster, M. A., Kurtz, N. T., and Petty, A. A.: Arctic snow depth and sea ice thickness from ICESat-2 and CryoSat-2 freeboards: A first examination. *J. Geophys. Res. Oceans*, 125, e2019JC016008, https://doi.org/10.1029/2019JC016008, 2020.

Lee, S.-M and Sohn, B.-J.: Retrieving the refractive index, emissivity, and surface temperature of polar sea ice from 6.9 GHz microwave measurements: A theoretical development, J. Geophys. Res. Atmos., 120(6), 2293-2305, https://doi.org/10.1002/2014JD022481, 2015.

Lee, S.-M., Shi, H., Sohn, B.-J., Gasiewski, A. J., Meier, W. N., and Dybkjær, G.: Winter snow depth on Arctic sea ice from satellite radiometer measurements (2003-2020): Regional patterns and trends. *Geophys. Res. Lett.*, 48(15), e2021GL094541, 2021.

Shi, H., Sohn, B.-J., Dybkjær, G., Tonboe, R. T., and Lee, S.-M.: Simultaneous estimation of wintertime sea ice thickness and snow depth from space-borne freeboard measurements, The Cryosphere, 14, 3761–3783, https://doi.org/10.5194/tc-14-3761-2020, 2020.

Zhou, L., Xu, S., Liu, J., and Wang, B.: On the retrieval of sea ice thickness and snow depth using concurrent laser altimetry and L-band remote sensing data, The Cryosphere, 12, 993– 1012, https://doi.org/10.5194/tc-12-993-2018, 2018.

Kilic, L., Tonboe, R. T., Prigent, C., and Heygster, G.: Estimating the snow depth, the snow–ice interface temperature, and the effective temperature of Arctic sea ice using Advanced Microwave Scanning Radiometer 2 and ice mass balance buoy data, Cryosphere, 13, 1283–1296, https://doi.org/10.5194/tc-13-1283-2019, 2019.

---

## Author Response (AR3)

**Response to Referee #1 "Review of: A simple model for daily basin-wide thermodynamic sea ice thickness growth retrieval"**

**By Anheuser, et al.**

Author responses in red.

1 Synopsis

Congratulations again to the authors for another round of good responses. I should apologize for the substantially delayed review on my end. I'm pleased to say I'm very satisfied with the responses and revised manuscript. I think the paper now represents a rigorous and interesting contribution to the literature on sea ice thickness - in particular I think it has a strong element of literature review, and it is pleasantly surprising how well the product does given its relatively low complexity and number of data inputs.

I have a couple of remaining comments. If the editor is satisfied that the authors have suitably responded to these, then I am happy for the manuscript to be accepted without further review.

2 Comments

Fig. 2: The label on the colorbar should be padded away from the tick labels to improve readability. This looks like a python plot - if so, this can be done with the labelpad keyword.

This change has been made.

Fig 3: This is a good figure, but needs a bit of cleaning up. I don't think the colorbar label needs '[-]' in it. Furthermore, a line break would help improve readability for the right-hand panel's y-axis label, and the subplots could do with being moved apart a bit (fig.subplots adjust(wspace)). You have lots of horizontal space to do this, so it should be straightforward. You also need (a) & (b) annotations to match the caption. Finally, the text annotations in both panels need proper right-hand-justifying, if you're using python you can do this with the ha keyword passed to ax.annotate().

The same points from Fig. 3 also apply to Fig. 6 & 8. Colorbars could also do with a tick-label at 0.

These updates have been made to Figs. 3, 6, and 8.

L458: 'instantaneous'

This spelling has been updated.

L501: That's a nice point about not being dependent on atmospheric reanalysis, consider putting this in your abstract.

Good suggestion, we have added this to the abstract:

*The advantages of the SLICE retrieval method include daily basin-wide coverage, lack of atmospheric reanalysis product input requirement, and a potential for use beginning in 1987.*

**Response to Referee #2 "Review of: A simple model for daily basin-wide thermodynamic sea ice thickness growth retrieval"**

**By Anheuser, et al.**

Author responses in red.

I appreciate the authors' efforts to respond to my questions and comments. Most of the responses are successful, thus I think this manuscript has improved to be published in the Cryosphere.

However, I found some errors in the revised manuscript (inconsistency between the authors' response and the revised manuscript). Those are:

1) Abstract: I cannot find the passage "basin-wide SLICE has equal linear correlation ... Operation IceBridge, respectively.", which authors wrote in the authors' response. I suppose that they changed the word "equally" to "nearly"? Please confirm...

The next revision will include the following as the last line of the abstract:

*Despite its simplifications and assumptions relative to models like the Pan-Arctic Ice–Ocean Modeling and Assimilation System (PIOMAS), basin-wide SLICE performs nearly as well as PIOMAS when compared against CryoSat-2 and Operation IceBridge using a linear correlation between collocated points.*

2) Figure 2: Inconsistency between the response and figure in the revised manuscript. The authors said they show snow-ice interface temperatures on November 1, 2013 (also in the text). However, the caption in the manuscript is about January 1, 2013.

We have updated the manuscript text to state the figure shows January 1, 2013 snow—ice interface temperatures, which is indeed to data being show. We discussed a few other captions and numbers within the text that were not updated from previous revisions:

L9: mean correlation of 0.90 was replaced by 0.89 and mean bias of 0.08 m was replaced by 0.06 m

Fig. 3 caption: "Both exhibit a linear correlation of 0.71" was replaced by "Linear correlations are 0.70 and 0.71, respectively".

Fig. 4 caption: mean correlation of 0.90 was replaced by 0.89 and mean bias of 0.08 m was replaced by 0.06 m

L358-L360: mean (standard deviation) correlation of 0.90 (0.19) was replaced by 0.89 (0.21) and mean bias of 0.08 m was replaced by 0.06 m

Fig. 6 caption: "SLICE has the highest linear correlation though all three are virtually equal" replaced with "All three have similar linear correlations".

L413: mean correlation of 0.90 was replaced by 0.89 and mean bias of 0.08 m was replaced by 0.06 m

Fig. 8 caption: "SLICE has as lower bias at 0.03 m than PIOMAS at 0.10 m and linear correlation values are equal" replaced with "SLICE and PIOMAS have nearly equal mean bias and linear correlation values."

L484: mean correlation of 0.90 was replaced by 0.89 and mean bias of 0.08 m was replaced by 0.06 m

3) I found that the word "improved" in L395 of the previous version of the manuscript is replaced with "similar". However, the authors responded that they replace it with the word "lower". Please verify... Is this response to figure 8?

We've changed this word to "lower".

These must be corrected or confirmed before publication.